# Flexible experimental designs for valid single-cell RNA-sequencing experiments allowing batch effects correction

Fangda Song [1], Ga Ming Angus Chan[1] & Yingying Wei [1✉]

Despite their widespread applications, single-cell RNA-sequencing (scRNA-seq) experiments are still plagued by batch effects and dropout events. Although the completely randomized experimental design has frequently been advocated to control for batch effects, it is rarely implemented in real applications due to time and budget constraints. Here, we mathematically prove that under two more flexible and realistic experimental designs—the reference panel and the chain-type designs—true biological variability can also be separated from batch effects. We develop Batch effects correction with Unknown Subtypes for scRNA-seq data (BUSseq), which is an interpretable Bayesian hierarchical model that closely follows the data-generating mechanism of scRNA-seq experiments. BUSseq can simultaneously correct batch effects, cluster cell types, impute missing data caused by dropout events, and detect differentially expressed genes without requiring a preliminary normalization step. We demonstrate that BUSseq outperforms existing methods with simulated and real data.

[1] Department of Statistics, The Chinese University of Hong Kong, Hong Kong SAR, China. ✉email: yweicuhk@gmail.com

 

Single-cell RNA-sequencing (scRNA-seq) technologies enable the measurement of the transcriptome of individual cells, which provides unprecedented opportunities to discover cell types and understand cellular heterogeneity[1]. However, like the other high-throughput technologies[2–4], scRNA-seq experiments can suffer from severe batch effects[5]. Moreover, compared with bulk RNA-seq data, scRNA-seq data can have an excessive number of zeros that result from either biological zeros—that is, a gene is not expressed in a given cell—or dropout events—that is, the expression of some genes are not detected even though they are actually expressed in the cell due to amplification failure prior to sequencing[6]. Consequently, despite the widespread adoption of scRNA-seq experiments, the design of a valid scRNA-seq experiment that allows the batch effects to be removed, the biological cell types to be discovered, and the missing data to be imputed remains an open problem.

One of the major tasks of scRNA-seq experiments is to identify cell types for a population of cells[1]. The cell type of each individual cell is unknown and is often the target of inference. Classic batch effects correction methods, such as ComBat[7] and SVA[8,9], are designed for bulk experiments and require knowledge of the subtype information of each sample a prior. For scRNA-seq data, this subtype information corresponds to the cell type of each individual cell. Clearly, these methods are thus infeasible for scRNA-seq data. Alternatively, if one has knowledge of a set of control genes whose expression levels are constant across cell types, then it is possible to apply RUV[10,11]. However, selecting control genes is still challenging for scRNA-seq experiments, and recently there has been active research on identifying stably expressed genes that are reproducible and conserved across species for single cells[12].

To identify unknown subtypes, MetaSparseKmeans[13] jointly clusters samples across batches. Unfortunately, MetaSparseKmeans requires all subtypes to be present in each batch. Suppose that we conduct scRNA-seq experiments for blood samples from a healthy individual and a leukemia patient, one person per batch. Although we can anticipate that the two batches will share T cells and B cells, we do not expect that the healthy individual will have cancer cells as the leukemia patient. Therefore, MetaSparseKmeans is too restrictive for many scRNA-seq experiments.

The mutual-nearest-neighbor (MNN) based approaches, including MNN[14] and Scanorama[15], allow each batch to contain some but not all cell types. However, these methods require batch effects to be almost orthogonal to the biological subspaces and much smaller than the biological variations between different cell types[14]. These are strong assumptions and cannot be validated at the design stage of the experiments. Seurat[16,17], LIGER[18], and scMerge[19] attempt to identify shared variations across batches by low-dimensional embeddings and treat them as shared cell types. However, they may mistake the technical artifacts as the biological variability of interest if some batches share certain technical noises, for example when each patient is measured by several batches. To handle severe batch effects for microarray data, Luo and Wei[20] developed BUS to simultaneously cluster samples across multiple batches and correct batch effects. However, none of the above methods considers features unique to scRNA-seq data, such as the count nature of the data, overdispersion[21], dropout events[6], or cell-specific size factors[22]. ZIFA[23] and ZINB-WaVE[24] incorporate dropout events into the factor model, whereas scVI[25] and SAVER-X[26] couple the modeling of dropout events with neural networks. However, as is the case with the other state-of-the-art methods, these papers do not discuss the designs of scRNA-seq experiments under which their methods are applicable.

Nevertheless, it is crucial to understand the conditions under which biological variability can be separated from technical artifacts. Obviously, for completely confounded designs—for example one in which batch 1 measures cell type 1 and 2, whereas batch 2 measures cell type 3 and 4—no method is applicable.

Here, we propose Batch effects correction with Unknown Subtypes for scRNA-seq data (BUSseq), an interpretable hierarchical model that simultaneously corrects batch effects, clusters cell types, and takes care of the count data nature, the overdispersion, the dropout events, and the cell-specific size factors of scRNA-seq data. We mathematically prove that it is legitimate to conduct scRNA-seq experiments under not only the commonly advocated completely randomized design[1,5,27,28], in which each batch measures all cell types, but also the reference panel design and the chain-type design, which allow some cell types to be missing from some batches. Furthermore, we demonstrate that BUSseq outperforms the existing approaches in both simulation data and real applications. The theoretical results answer the question about when we can integrate multiple scRNA-seq datasets and analyze them jointly. We envision that the proposed experimental designs will be able to guide biomedical researchers and help them to design better scRNA-seq experiments.

## Results

**BUSseq is an interpretable hierarchical model for scRNA-seq.** We develop a hierarchical model BUSseq that closely mimics the data generating procedure of scRNA-seq experiments (Fig. 1a, Supplementary Fig. 1 and Supplementary Note 1). Given that we have measured $B$ batches of cells each with a sample size of $n_b$, let us denote the underlying gene expression level of gene $g$ in cell $i$ of batch $b$ as $X_{big}$. $X_{big}$ follows a negative binomial distribution with mean expression level $\mu_{big}$ and a gene-specific and batch-specific overdispersion parameter $\phi_{bg}$. The mean expression level is determined by the cell type $W_{bi}$ with the cell type effect $\beta_{gk}$, the log-scale baseline expression level $\alpha_g$, the location batch effect $\nu_{bg}$, and the cell-specific size factor $\delta_{bi}$. The cell-specific size factor $\delta_{bi}$ characterizes the impact of cell size, library size and sequencing depth. It is of note that the cell type $W_{bi}$ of each individual cell is unknown and is our target of inference. Therefore, we assume that a cell on batch $b$ comes from cell type $k$ with probability $\Pr(W_b = k) = \pi_{bk}$ and the proportions of cell types $(\pi_{b1}, \cdots, \pi_{bK})$ vary among batches.

Unfortunately, it is not always possible to observe the expression level $X_{big}$. Without dropout ($Z_{big} = 0$), we can directly observe $Y_{big} = X_{big}$. However, if a dropout event occurs ($Z_{big} = 1$), then we observe $Y_{big} = 0$ instead of $X_{big}$. In other words, when we observe a zero read count $Y_{big} = 0$, there are two possibilities: a non-expressed gene—biological zeros—or a dropout event. When gene $g$ is not expressed in cell $i$ of batch $b$ ($X_{big} = 0$), we always have $Y_{big} = 0$; when gene $g$ is actually expressed in cell $i$ of batch $b$ ($X_{big} > 0$) but a dropout event occurs, we can only observe $Y_{big} = 0$, and hence $Z_{big} = 1$. It has been noted that highly expressed genes are less-likely to suffer from dropout events[6]. We thus model the dependence of the dropout rate $\Pr(Z_{big} = 1|X_{big})$ on the expression level using a logistic regression with batch-specific intercept $\gamma_{b0}$ and log-odds ratio $\gamma_{b1}$.

Noteworthy, BUSseq includes the negative binomial distribution without zero inflation as a special case. When all cells are from a single cell type and the cell-specific size factor $\delta_{bi}$ is estimated a priori according to spike-in genes, BUSseq can reduce to a form similar to BASiCS[21].

We only observe $Y_{big}$ for all cells in the $B$ batches and the total $G$ genes. We conduct statistical inference under the Bayesian framework and adopt the Metropolis-within-Gibbs algorithm[29] for the Markov chain Monte Carlo (MCMC) sampling[30]

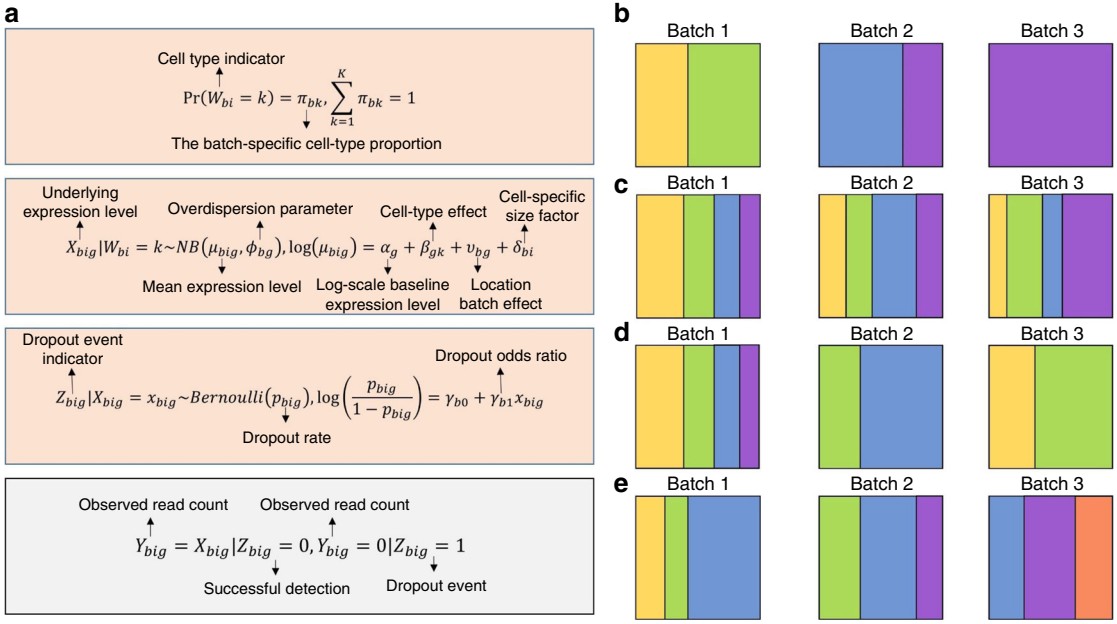

**Fig. 1 Illustration of the BUSseq model and various types of experimental designs. a** The hierarchical structure of the BUSseq model. Only $Y_{big}$ in the gray rectangle is observed. **b** A confounded design that contains three batches. Each polychrome rectangle represents one batch of scRNA-seq data with genes in rows and cells in columns; and each color indicates a cell type. Batch 1 assays cells from cell types 1 and 2; batch 2 profiles cells from cell types 3 and 4; and batch 3 only contains cells from cell type 4. **c** The complete setting design. Each batch assays cells from all of the four cell types, although the cellular compositions vary across batches. **d** The reference panel design. Batch 1 contains cells from all of the cell types, and all of the other batches have at least two cell types. **e** The chain-type design. Every two consecutive batches share two cell types. Batch 1 and Batch 2 share cell types 2 and 3; Batch 2 and Batch 3 share cell types 3 and 4 (see also Supplementary Figs. 1 and 2).

(Supplementary Note 2). Based on the parameter estimates, we can learn the cell type for each individual cell, impute the missing underlying expression levels $X_{big}$ for dropout events, and identify genes that are differentially expressed among cell types. Moreover, our algorithm can automatically detect the total number of cell types $K$ that exists in the dataset according to the Bayesian information criterion (BIC)[31]. BUSseq also provides a batch-effect corrected version of count data, which can be used for downstream analysis as if all of the data were measured in a single batch ("Methods").

**Valid experimental designs for scRNA-seq experiments**. If a study design is completely confounded, as shown in Fig. 1b, then no method can separate biological variability from technical artifacts, because different combinations of batch-effect and cell-type-effect values can lead to the same probabilistic distribution for the observed data, which in statistics is termed a non-identifiable model. Formally, a model is said to be identifiable if each probability distribution can arise from only one set of parameter values[32]. Statistical inference is impossible for non-identifiable models because two sets of distinct parameter values can give rise to the same probability distribution function. We prove that the BUSseq model is identifiable under conditions that are very easily met in reality. Thus, a wide range of designs of scRNA-seq experiments are valid as their batch effects can be adjusted at least by BUSseq.

For the complete setting, in which each batch measures all of the cell types (Fig. 1c and Theorem 1 in "Methods"), BUSseq is identifiable as long as: (I) the log-odds ratio $\gamma_{b1}$s in the logistic regressions for the dropout rates are negative for all of the batches, (II) every two cell types have more than one differentially expressed gene, and (III) the ratios of mean expression levels between two cell types $(\frac{\exp(\beta_{1k})}{\exp(\beta_{1\tilde{k}})}, \cdots, \frac{\exp(\beta_{Gk})}{\exp(\beta_{G\tilde{k}})})$ are different for each cell-type pair $(k, \tilde{k})$ (see Theorem 1 in "Methods"). Condition (I)

requires that the highly expressed genes are less likely to have dropout events, which is routinely observed for scRNA-seq data[6]. Condition (II) always holds in reality. Because scRNA-seq experiments measure the whole transcriptome of a cell, condition (III) is also always met in real data. For example, if there exists one gene $g$ such that for any two distinct cell-type pairs $(k_1, k_2)$ and $(k_3, k_4)$ their mean expression levels' ratios $\frac{\exp(\beta_{gk_1})}{\exp(\beta_{gk_2})}$ and $\frac{\exp(\beta_{gk_3})}{\exp(\beta_{gk_4})}$ are not the same, then condition (III) is already satisfied.

The commonly advocated completely randomized experimental design is a special case of the complete setting design. In a completely randomized design, cells are assigned to different batches completely at random. As a result, all of the batches have similar compositions of cell populations. In contrast, under the complete setting design, cells from different cell types can be distributed to different batches very unevenly. The requirement that each batch has similar cellular compositions is crucial for traditional batch effects correction methods developed for bulk experiments such as ComBat[7] to work well for scRNA-seq data. In contrast, BUSseq is not limited to this balanced design constraint and is applicable to not only the completely randomized design but also the general complete setting design.

Ideally, we would wish to adopt completely randomized experimental designs. However, in reality, it is always very challenging to implement complete randomization due to time and budget constraints. For example, when we recruit patients sequentially, we often have to conduct scRNA-seq experiments patient-by-patient rather than randomize the cells from all of the patients to each batch, and the patients may not have the same set of cell types. Fortunately, we can prove that BUSseq also applies to two sets of flexible experimental designs, which allow cell types to be measured in only some but not all of the batches.

Assuming that conditions (I)–(III) are satisfied, if there exists one batch that contains cells from all cell types and the other batches have at least two cell types (Fig. 1d), then BUSseq can

tease out the batch effects and identify the true biological variability (see Theorem 2 in "Methods"). We call this setting the reference panel design.

Sometimes, it can still be difficult to obtain a reference batch that collects all cell types. In this case, we can turn to the chain-type design, which requires every two consecutive batches to share two cell types (Fig. 1e). Under the chain-type design, given that conditions (I)–(III) hold, BUSseq is also identifiable and can estimate the parameters well (see Theorem 3 in "Methods").

A special case of the chain-type design is when two common cell types are shared by all of the batches, which is frequently encountered in real applications. For instance, when blood samples are assayed, even if we perform scRNA-seq experiment patient-by-patient with one patient per batch, we know a priori that each batch will contain at least both T cells and B cells, thus satisfying the requirement of the chain-type design.

The key insight is that despite batch effects, differences between cell types remain constant across batches. The differences between a pair of cell types allow us to distinguish batch effects from biological variability for those batches that measure both cell types. Therefore, BUSseq can separate batch effects from cell type effects under more general designs beyond the easily understood and commonly encountered reference panel design and chain-type design. If we regard each batch as a node in a graph and connect two nodes with an edge if the two batches share at least two cell types, then BUSseq is identifiable as long as the resulting graph is connected (Supplementary Fig. 2 and Theorem 4 in "Methods").

For scRNA-seq data, dropout rate depends on the underlying expression levels[6]. Such missing data mechanism is called missing not at random (MNAR) in statistics. It is very challenging to establish identifiability for MNAR. Miao et al.[33] showed that for many cases even when both the outcome distribution and the missing data mechanism has parametric forms, the model can be nonidentifiable. However, fortunately, despite the dropout events and the cell-specific size factors, we are able to prove Theorems 1–4 (Supplementary Note 3). The reference panel design, the chain-type design, and the connected design liberalize researchers from the ideal but often unrealistic requirement of the completely randomized design.

**BUSseq accurately learns the parameters and the missing data.** We first evaluated the performance of BUSseq via a simulation study. We simulated a dataset with four batches and a total of five cell types under the chain-type design (Fig. 2a–d and Theorem 3). Every two consecutive batches share at least two cell types, but none of the batches contains all of the cell types. The sample sizes for each batch are $(n_1, n_2, n_3, n_4) = (300, 300, 200, 200)$ (Supplementary Table 1), and there are a total of 3000 genes, out of which 500 genes are differentially expressed between cell types. The remaining 2500 genes have no biological differences between different cell types, so they are pure noises with only batch effects. In real datasets, batch effects are often much larger than the cell type effects (Fig. 3a) and not orthogonal to the cell type effects (Supplementary Fig. 3). In the simulation study, we choose the magnitude of the batch effects, cell type effects, the dropout rates, and the cell-specific size factors to mimic real data scenarios (Fig. 3a). The simulated observed data suffer from severe batch effects and dropout events (Figs. 2d, 3c). The dropout rates for the four batches are 26.79%, 24.53%, 28.36%, and 31.29%, with the corresponding total zero proportions given by 44.13%, 48.85%, 53.07%, and 61.38%.

BUSseq correctly identifies the presence of five cell types among the cells (Fig. 2e). Moreover, despite the dropout events, BUSseq accurately estimates the cell type effects $\beta_{gk}$s (Fig. 2a, f),

the batch effects $\nu_{bg}$s (Fig. 2b, g), and the cell-specific size factors $\delta_{bi}$s (Fig. 2j). In particular, BUSseq outperforms existing normalization methods, including DESeq normalization[34], trimmed mean of M-values (TMM) normalization[35], library size normalization, and the deconvolution normalization method[36], in estimating the cell-specific size factors $\delta_{bi}$s (Supplementary Fig. 4 and Supplementary Note 4). When controlling the Bayesian False Discovery Rate (FDR) at 5%[37,38], we identify all intrinsic genes that differentiate cell types with the true FDR being 2% ("Methods").

Figure 2h demonstrates that BUSseq can learn the underlying expression levels $X_{big}$s well based on the observed data $Y_{big}$s, which are subject to dropout events. This success arises because BUSseq uses an integrative model to borrow strengths both across genes and across cells from all batches. In comparison, we also benchmarked BUSseq with three state-of-the-art imputation methods for scRNA-seq data—SAVER[39], DrImpute[40], and scImpute[41]. Once again, BUSseq performs the best in identifying the true biological zeros and recovering the underlying expression levels $X_{big}$s for the dropout events (Supplementary Table 2 and Supplementary Note 5).

ComBat offers a version of data that have been adjusted for batch effects[7]. Here, we also provide batch-effects-corrected count data based on quantile matching ("Methods"). The adjusted count data no longer suffer from batch effects and dropout events, and they even do not need further cell-specific normalization (Fig. 2i). Therefore, they can be treated as if measured in a single batch for downstream analysis.

To evaluate the robustness of BUSseq, we conducted extensive sensitivity analyses, and they show that BUSseq is robust to the choice of hyperparameters, high zero rates, model misspecification and gene filtering (Supplementary Figs. 5–7, Supplementary Tables 3 and 4, and Supplementary Note 6).

**BUSseq outperforms existing methods in simulation study.** We benchmarked BUSseq with the state-of-the-art methods for batch effects correction for scRNA-seq data—LIGER[18], MNN[14], Scanorama[15], scVI[25], Seurat[17], and ZINB-WaVE[24]. The adjusted Rand index (ARI) measures the consistency between two clustering results and is between zero and one, a higher value indicating better consistency[42] (Supplementary Note 7). The ARI between the inferred cell types $\widehat{W}_{bi}$s by BUSseq and the true underlying cell types $W_{bi}$s is one. Thus, BUSseq can perfectly recover the true cell type of each cell. In comparison, we applied each of the compared methods to the dataset and then performed their own clustering approaches (Supplementary Note 8). The ARI is able to compare the consistency of two clustering results even if the numbers of clusters differ, therefore, we chose the number of cell types by the default approach of each method rather than set it to a common number. The resulting ARIs are 0.837 for LIGER, 0.654 for MNN, 0.521 for Scanorama, 0.480 for scVI, 0.632 for Seurat, and 0.571 for ZINB-WaVE. Moreover, the t-SNE plots (Fig. 3c, d) show that only BUSseq can perfectly cluster the cells by cell types rather than batches. We also calculated the silhouette score for each cell for each compared method (Supplementary Note 7). A high silhouette score indicates that the cell is well matched to its own cluster and separated from neighboring clusters. Figure 3b shows that BUSseq gives the best segregated clusters.

**BUSseq outperforms existing methods on hematopoietic data.** We re-analyzed the two hematopoietic datasets[43,44] previously studied by Haghverdi et al.[14] (Fig. 4a and Supplementary Fig. 8a, b). The two datasets shared at least three cell types, including the common myeloid progenitors (CMP), megakaryocyte-erythrocyte

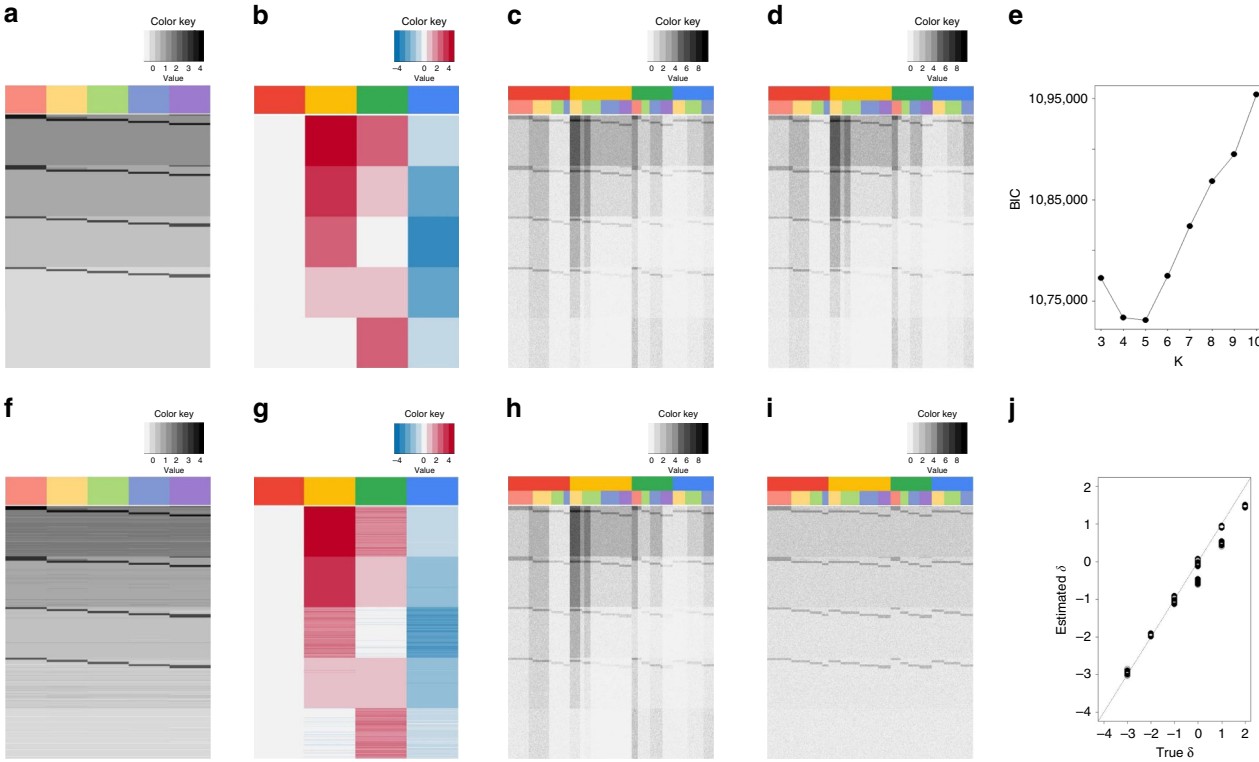

**Fig. 2 Patterns of the simulation study. a** True log-scale mean expression levels for each cell type $\alpha + \beta$. Each row represents a gene, and each column corresponds to a cell type. The intrinsic genes that are differentially expressed between cell types can have high, medium high, median low or low expression levels. **b** True batch effects. Each row represents a gene, and each column corresponds to a batch. **c** True underlying expression levels **X**. Each row represents a gene, and each column corresponds to a cell. The upper color bar indicates the batches, and the lower color bar represents the cell types. There are a total of 3000 genes. The sample sizes for each batch are 300, 300, 200, and 200, respectively. **d** The simulated observed data **Y**. The overall dropout rate is 27.3%, whereas the overall zero rate is 50.8%. **e** The BIC plot. The BIC attains the minimum at $K = 5$, identifying the true cell type number. **f** The estimated log-scale mean expression levels for each cell type $\widehat{\alpha} + \widehat{\beta}$. **g** Estimated batch effects. **h** Imputed expression levels $\widehat{\mathbf{X}}$. **i** Corrected count data $\widetilde{\mathbf{X}}$ grouped by batches. **j** Scatter plot of the estimated versus the true cell-specific size factors.

progenitors (MEP) and granulocyte-monocyte progenitors (GMP), thus they follow the chain-type design.

BUSseq fits the zero rates (Table 1 and Supplementary Note 9) and the mean-variance trends (Fig. 5a, Supplementary Fig. 9 and Supplementary Note 7) of the real data very well. In order to compare BUSseq with existing methods, we compute the ARIs between the clustering of each method and the FACS labels. The resulting ARIs are 0.582 for BUSseq, 0.307 for LIGER, 0.575 for MNN, 0.518 for Scanorama, 0.197 for scVI, 0.266 for Seurat, and 0.348 for ZINB-WaVE (Supplementary Table 5 and Supplementary Note 8). BUSseq thus outperforms all of the other methods in being consistent with FACS labeling. BUSseq also has silhouette coefficients that are comparable to those of MNN, which are better than those of all the other methods (Fig. 4b and Supplementary Fig. 10a, b).

Specifically, BUSseq learns six cell types from the dataset. According to the FACS labels (Methods), Cluster 2, Cluster 5, and Cluster 6 correspond to CMP, MEP, and GMP, respectively (Figs. 4c, 6a–c). Cluster 1 is composed of long-term hematopoietic stem cells and multi-potent progenitors (MPP). These are cells from the early stage of differentiation. Cluster 4 consists of a mixture of MEP and CMP, while Cluster 3 is dominated by cells labeled as other. Comparison between the subpanel for BUSseq in Figs. 4c and 6b indicates that Cluster 4 are cells from an intermediate cell type between CMP and MEP. In particular, according to Fig. 6e, the marker genes *Apoe* and *Gata2* are highly expressed in Cluster 4 but not in CMP (Cluster 2) and MEP (Cluster 6), and the marker gene *Ctse* is expressed in

MEP (Cluster 6) but not in Cluster 4 and CMP (Cluster 2). Therefore, cells in Cluster 4 do form a unique group with distinct expression patterns. This intermediate cell stage between CMP and MEP is missed by all of the other methods considered. Moreover, we find that well known B-cell lineage genes[45], *Ebf1*, *Vpreb1*, *Vpreb3*, and *Igll1*, are highly expressed in Cluster 3, but not in the other clusters (Fig. 6c, e). To identify Cluster 3, which is dominated by cells labeled as other by Nestorowa et al.[43], we map the mean expression profile of each cluster learned by BUSseq to the Haemopedia RNA-seq dataset[46]. It turns out that Cluster 3 aligns well to common lymphoid progenitors (CLP) that give rise to T-lineage cells, B-lineage cells and natural killer cells (Fig. 6d). Therefore, Cluster 3 represents cells that differentiate from lymphoid-primed multipotent progenitors (LMPP)[44]. Once again, all the other methods fail to identify these cells as a separate group.

Thus, although BUSseq does not assume any temporal ordering between cell types, it is able to preserve the differentiation trajectories (Fig. 6a, b); although BUSseq assumes that each cell belongs to one cell type rather than conducts semisoft clustering[47], it is capable of capturing the subtle changes across cell types and within a cell type due to continuous processes such as development and differentiation (Supplementary Fig. 11 and Supplementary Note 10).

We further inspect the functions of the intrinsic genes that distinguish different cell types. BUSseq detects 1,419 intrinsic genes at the Bayesian false discovery rate (FDR) cutoff of 0.05 ("Methods"). The gene set enrichment analysis[48] shows that 51

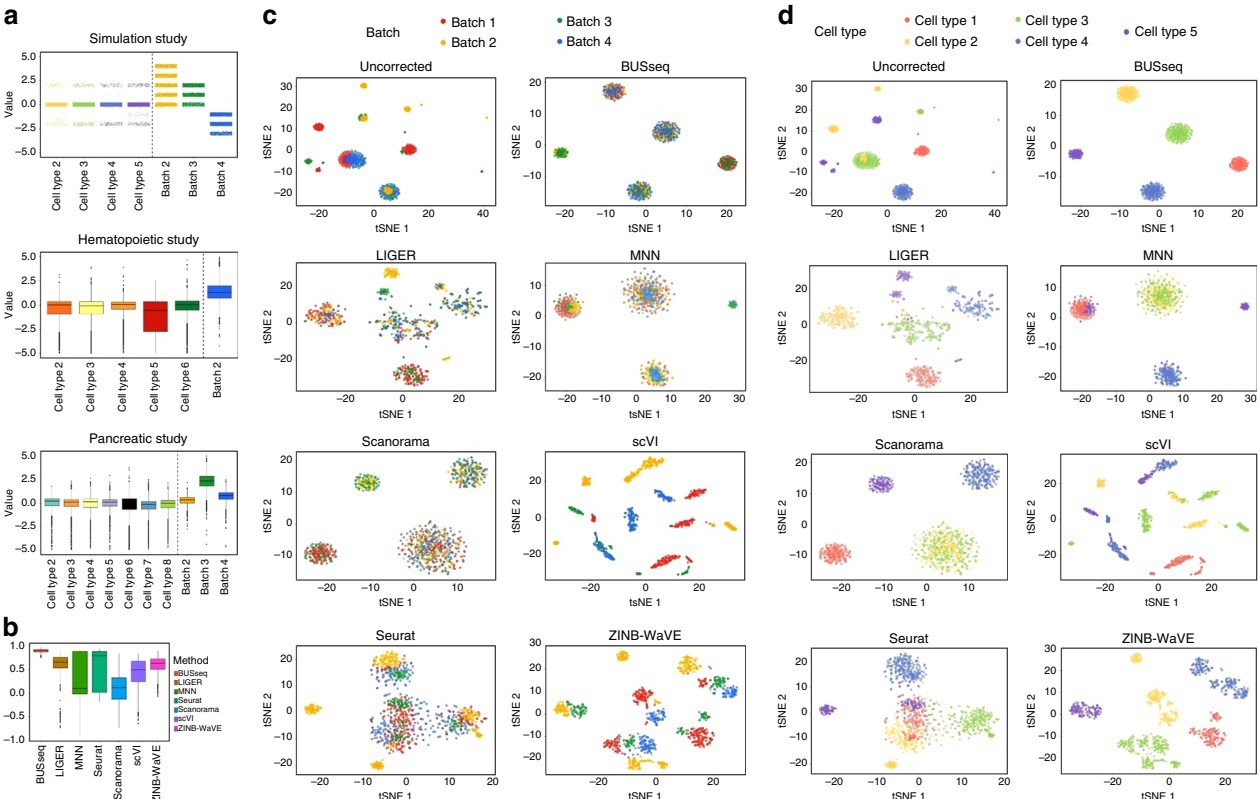

**Fig. 3 Comparison of batch effects correction methods in the simulation study. a** Comparison of the magnitude of cell type effects and batch effects in the simulation study and two real applications. The subpanel for the simulation study jitters around the assumed values for $\beta$ and $\nu$. The boxplots show the distributions of the estimated cell type effects $\widehat{\beta}$ and batch effects $\widehat{\nu}$ by BUSseq in the two real studies. The magnitude of the batch effects and cell type effects in the simulation study were chosen to mimic the real data scenarios. **b** The boxplots of silhouette coefficients for all compared methods. In these boxplots, the central line denotes the median; the upper and lower bounds represent the first and third quartiles; and the whiskers extend to a maximum of 1.5 times the interquartile (IQR) beyond the box, where the IQR is the difference between the third and the first quartiles. **c** T-distributed Stochastic Neighbor Embedding (t-SNE) plots colored by batch for each compared method. **d** t-SNE plots colored by true cell type labels for each compared method. BUSseq successfully corrects the batch effects and perfectly clusters cells into different cell types in the simulation study, whose batch effects and cell type effects are at the same scale as those of the two real datasets.

Kyoto Encyclopedia of Genes and Genomes (KEGG) pathways[49] are enriched among the intrinsic genes ($p$ values < 0.05) (Supplementary Note 11). The highest ranked pathway is the Hematopoietic Cell Lineage Pathway, which corresponds to the exact biological process studied in the two datasets. Among the remaining 50 pathways, 13 are related to the immune system, and another 9 are associated with cell growth and differentiation (Supplementary Table 6). Therefore, the pathway analysis demonstrates that BUSseq is able to capture the underlying true biological variability, even if the batch effects are severe, as shown in Figs. 3a and 4a.

**BUSseq outperforms existing method on pancreas data**. We further studied the four scRNA-seq datasets of human pancreas cells[50–52] analyzed in Haghverdi et al.[14]. These cells were isolated from deceased organ donors with and without type 2 diabetes. As each patient has at least two pancreas cell types—alpha cells and beta cells, the four datasets follow the chain-type design. We obtained 7095 cells after quality control (Methods) and treated each dataset as a batch following Haghverdi et al.[14].

BUSseq recapitulates the properties of real scRNA-seq data very well in terms of the zero rates (Table 1 and Supplementary Note 9) and the mean-variance trend (Fig. 5b and Supplementary Fig. 12). In particular, the posterior predictive check shows that BUSseq fits the zero rates much better than a model that ignores

dropout events, especially when scRNA-seq data are assayed by protocols that do not incorporate UMI counts, such as SMART-seq2.

We can compare the clustering results from each batch effects correction method with the cell-type labels provided by Segerstolpe et al.[52] and Lawlor et al.[51] (Fig. 7a, b and Supplementary Fig. 8c, d). The pancreas is highly heterogeneous and consists of two major categories of cells: islet cells and non-islet cells. Islet cells include alpha, beta, gamma, and delta cells, while non-islet cells include acinar and ductal cells. BUSseq identifies a total of eight cell types: five for islet cells, two for non-islet cells and one for the labeled other cells. Specifically, the five islet cell types identified by BUSseq correspond to three groups of alpha cells, a group of beta cells, and a group of delta and gamma cells. The two non-islet cell types identified by BUSseq correspond exactly to the acinar and ductal cells. Compared with all of the other methods, BUSseq gives the best separation between islet and non-islet cells, as well as the best segregation within islet cells. In particular, the median silhouette coefficient by BUSseq is higher than that of any other method (Fig. 7c and Supplementary Fig. 10c).

The ARIs of all methods are 0.608 for BUSseq, 0.542 for LIGER, 0.279 for MNN, 0.527 for Scanorama, 0.282 for scVI, 0.287 for Seurat, and 0.380 for ZINB-WaVE ("Methods" and Supplementary Table 5). Thus, BUSseq outperforms all of the other methods in being consistent with the cell-type labels

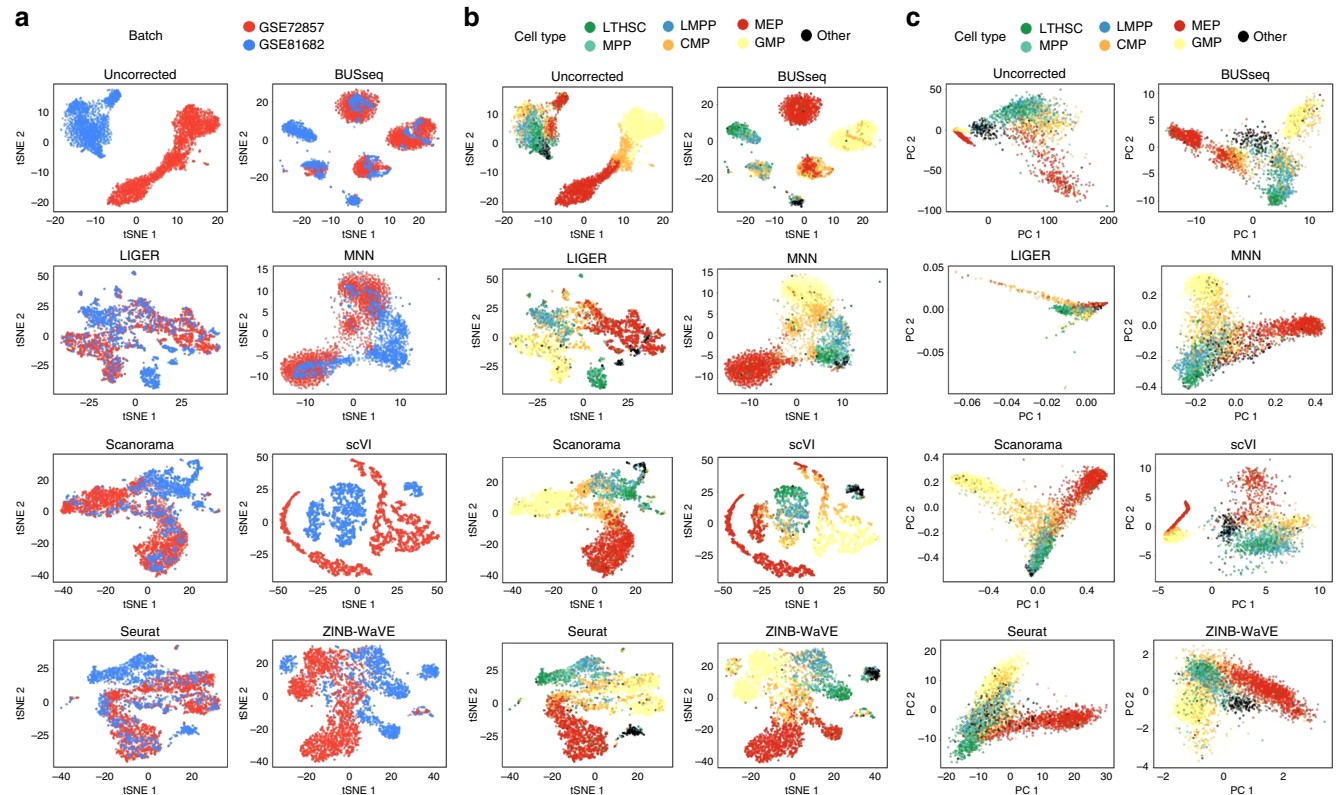

**Fig. 4 t-SNE and Principal Component Analysis (PCA) plots for the hematopoietic data. a** t-SNE plots colored by batch. **b** t-SNE plots colored by FACS cell type labels. **c** PCA plots colored by FACS cell type labels. BUSseq adjusts batch effects and correctly clusters cells according to their FACS cell type labels in the hematopoietic study.

**Table 1 Zero-count rates and dropout rates of the hematopoietic and pancreas studies.**

| Study | Protocol | UMI | $\rho_0$ | $\rho_d$ | $\widehat{\rho}_0^{BUSseq}$ | $\lvert\widehat{\rho}_0^{BUSseq} - \rho_0\rvert$ | $\widehat{\rho}_0^{BUSseq\text{-}nzf}$ | $\lvert\widehat{\rho}_0^{BUSseq\text{-}nzf} - \rho_0\rvert$ |
|---|---|---|---|---|---|---|---|---|
| Hematopoietic | MARS-seq | Yes | 0.892 | <0.001 | 0.887 | 0.005 | 0.874 | 0.018 |
| Hematopoietic | SMART-seq2 | No | 0.421 | <0.001 | 0.424 | 0.003 | 0.445 | 0.024 |
| Pancreas | CEL-seq2 | Yes | 0.689 | <0.001 | 0.625 | 0.064 | 0.682 | 0.007 |
| Pancreas | CEL-seq2 | Yes | 0.517 | 0.017 | 0.558 | 0.041 | 0.617 | 0.100 |
| Pancreas | SMART-seq2 | No | 0.609 | 0.167 | 0.531 | 0.078 | 0.430 | 0.179 |
| Pancreas | SMART-seq2 | No | 0.480 | 0.551 | 0.485 | 0.005 | 0.329 | 0.161 |

$\rho_0$ denotes the observed zero rate in each batch; $\rho_d$ represents the inferred dropout rate by BUSseq; $\widehat{\rho}_0^{BUSseq}$ denotes the posterior mean of zero rate inferred by BUSseq; and $\widehat{\rho}_0^{BUSseq\text{-}nzf}$ represents the posterior mean of zero rate inferred by a reduced model of BUSseq that ignores dropout events and hence uses negative binomial distribution without zero inflation, abbreviated as BUSseq-nzf. BUSseq detects the existence of dropout events automatically and performs better than BUSseq-nzf in terms of the posterior predictive check of zero rates.

according to marker genes. In Fig. 7d, the locally high expression levels of marker genes for each cell type show that BUSseq correctly clusters cells according to their biological cell types.

BUSseq identifies 426 intrinsic genes at the Bayesian FDR cutoff of 5% (Methods). We conducted the gene set enrichment analysis[48] with the KEGG pathways[49] (Supplementary Note 11). There are 14 enriched pathways (*p* values < 0.05). Among them, three are diabetes pathways; two are pancreatic and insulin secretion pathways; and another two pathways are related to metabolism (Supplementary Table 7). Recall that the four datasets assayed pancreas cells from type 2 diabetes and healthy individuals, therefore, the pathway analysis once again confirms that BUSseq provides biologically and clinically valid cell typing.

**BUSseq is applicable to droplet-based scRNA-seq data.** We further analyzed a dataset that contains samples assayed by

droplet-based scRNA-seq protocols. Comparing the performance of different methods on real scRNA-seq data is challenging due to the lack of true cell type labels in real application. Fortunately, Tian et al.[53] created scRNA-seq datasets with known cell type labels by profiling cells from cancer cell lines. In one experiment, they assayed three lung adenocarcinoma (LUAD) cell lines—HCC827, H1975, and H2228 on three platforms with CELseq2, 10x Chromium and Drop-seq protocols, respectively. As a result, 1401 cells were totally measured on three batches. Each batch consists of all of the three cell types, and data from different batches have different levels of sparsity. Consequently, this study satisfies the complete setting, which is a special case of both the reference-panel design and the chain-type design.

We selected the top 6000 highly variable genes (HVGs) within each batch and obtained 2267 common HVGs across three batches ("Methods"). The t-SNE and PCA plots of the raw count data show that significant batch effects occur across the three

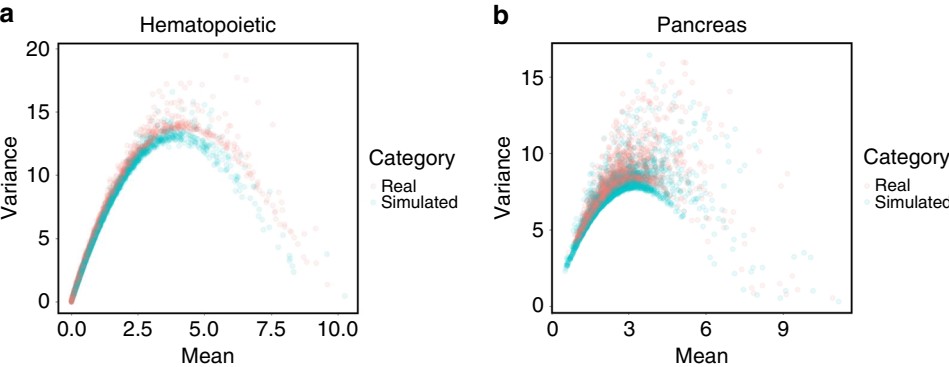

**Fig. 5 BUSseq recapitulates the mean-variance trends of the real data.** Scatter plot of the variance versus the mean of each gene for **a** the hematopoietic study and **b** the pancreas study. Red points are the observed values from real data; blue points correspond to the values of the data simulated according to the estimates of BUSseq for the real data.

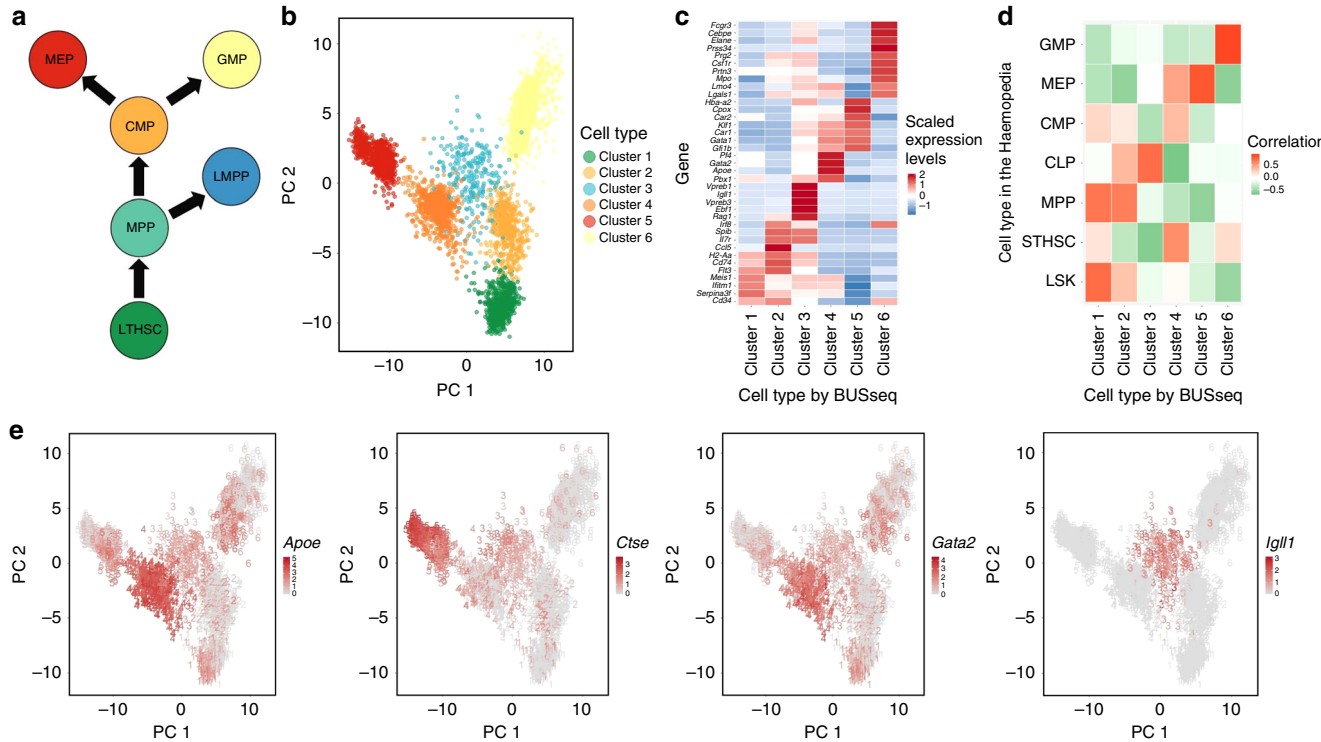

**Fig. 6 BUSseq preserves the hematopoietic stem and progenitor cells (HSPC) differentiation trajectories. a** The diagram of HSPC differentiation trajectories. Figure 6a is adapted from Fig. 3e of Haghverdi et al.[14] with the orientation and color coding changed. Reprinted by permission from Springer Nature, Nature Biotechnology, Batch effects in single-cell RNA-sequencing data are corrected by matching mutual nearest neighbors, Haghverdi et al. Copyright 2018. **b** The PCA plot of the corrected count matrix from BUSseq colored according to the estimated cell types by BUSseq. **c** The heatmap of scaled expression levels of key genes for HSPC. **d** The heatmap of correlation between gene expression profiles of each cell type inferred by BUSseq and those in the Haemopedia RNA-seq datasets. **e** The expression levels of four marker genes, *Apoe*, *Gata2*, *Ctse*, and *Igll1*, shown in the PCA plots of corrected count data by BUSseq, respectively. The digit labels denote the corresponding clusters identified by BUSseq.

protocols (Fig. 8a, b and Supplementary Fig. 13a, b). We applied BUSseq and varied the number of cell type $K$ from 2 to 6. Although the BIC selects four cell types instead of three cell lines (Supplementary Fig. 14), two of the four identified clusters correspond to two subpopulations of the H1975 cell lines (Supplementary Table 8). We further visualized the log-scale mean expression levels of intrinsic genes of the four learned cell types (Fig. 8e). The first two cell types have similar expression patterns, but some differentially expressed genes are observed between them. Moreover, the t-SNE (Fig. 8c, d) and PCA

(Supplementary Fig. 13c, d) plots demonstrate the high level of similarity of the first two estimated cell types and confirm that the corrected count data $\tilde{x}_{big}$ obtained by BUSseq cluster cells by cell type instead of by batch (Fig. 8f).

We also applied the benchmarked methods to compare their clustering accuracy. The ARIs of all methods are 0.841 for BUSseq, 0.825 for LIGER, 0.650 for MNN, 0.637 for Scanorama, 0.429 for scVI, 0.324 for Seurat, and 0.398 for ZINB-WaVE. Thus, BUSseq outperforms all of the other methods in clustering accuracy. We further compared BUSseq with a recently proposed

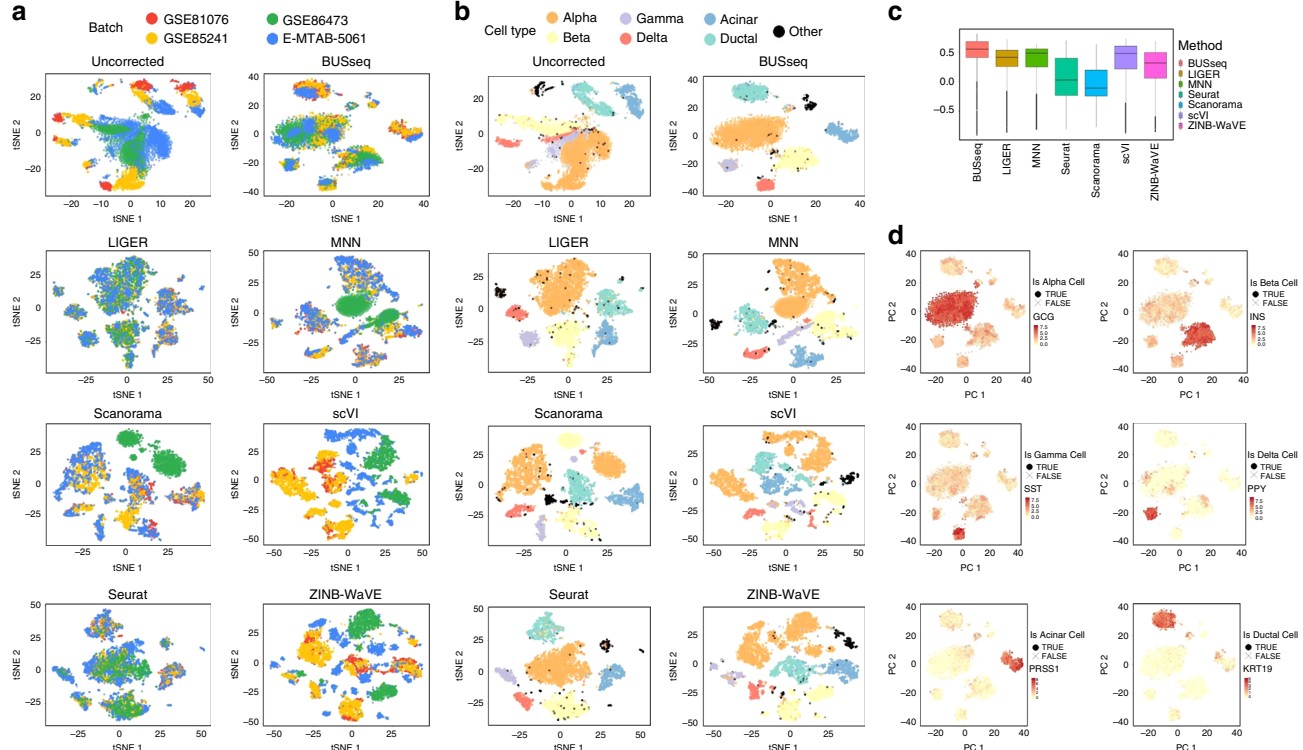

**Fig. 7 t-SNE plots for the pancreas data. a** t-SNE plots colored by batch. **b** t-SNE plot colored by FACS cell type labels. **c** The boxplot of silhouette coefficients for all of the compared methods. In the boxplot, the central line denotes the median; the upper and lower bounds represent the first and third quartiles; and the whiskers extend to a maximum of 1.5 times the interquartile (IQR) beyond the box, where the IQR is the difference between the third and the first quartiles. **d** The expression levels of six marker genes, *GCG* for alpha cells, *INS* for beta cells, *SST* for gamma cells, *PPY* for delta cells, *PRSS1* for acinar cells, and *KRT19* for ductal cells, shown in the t-SNE plot of the corrected count data of BUSseq, respectively.

semi-supervised batch-effect-correction methods, CellAssign. CellAssign requires the number of cell types and the input of a set of marker genes for each cell type. It then annotates scRNA-seq into predefined or de novo cell types[54]. To allow a fair comparison, we also set the number of cell types as the priori known three for BUSseq, and the resulting ARI for BUSseq becomes 0.993. Even though CellAssign is semi-supervised whereas BUSseq is unsupervised, BUSseq outperforms CellAssign in the LUAD dataset as well (ARI for CellAssign is 0.972, Supplementary Table 9). Thus, BUSseq also works very well for scRNA-seq data with high levels of sparsity, such as those generated by droplet-based protocols.

## Discussion

For the completely randomized experimental design, it seems that everyone is talking but no one is listening. Due to time and budget constraints, it is always difficult to implement a completely randomized design in practice. Consequently, researchers often pretend to be blind to the issue when carrying out their scRNA-seq experiments. In this paper, we mathematically prove and empirically show that under the more realistic reference panel and chain-type designs, batch effects can also be adjusted for scRNA-seq experiments. We hope that our results will alarm researchers of confounded experimental designs and encourage them to implement valid designs for scRNA-seq experiments in real applications.

BUSseq provides one-stop services. In contrast, most existing methods are multi-stage approaches—clustering can only be performed after the batch effects have been corrected and the differential expressed genes can only be called after the cells have been clustered. The major issue with multi-stage methods is that

uncertainties in the previous stages are often ignored. For instance, when cells have been first clustered into different cell types and then differential gene expression identification is conducted, the clustering results are taken as if they were the underlying truth. As the clustering results may be prone to errors in practice, this can lead to false positives and false negatives. In contrast, BUSseq simultaneously corrects batch effects, clusters cell types, imputes missing data, and identifies intrinsic genes that differentiate cell types. BUSseq thus accounts for all uncertainties and fully exploits the information embedded in the data. As a result, BUSseq is able to capture subtler changes between cell types, such as the cluster corresponding to LMPP lineage that is missed by all of the state-of-the-art methods.

BUSseq employs MCMC algorithm for statistical inference. Although MCMC algorithms are well-known for heavy computation load, fortunately, the computational complexity of BUSseq is $O(\sum_{b=1}^{B} n_b GK)$, which is both linear in the number of genes $G$ and in the total number of cells $N = \sum_{b=1}^{B} n_b$. Moreover, most steps of the MCMC algorithm for BUSseq are parallelizable (Supplementary Note 12). We implement a parallel multi-core-CPU version and a parallel GPU version of the algorithm, respectively. Running the GPU version of the algorithm with a single core of an Intel Xeon Gold 6132 Processor and one NVIDIA Tesla P100 GPU took 0.35, 1.15, 1.5 h for the simulation, the hematopoietic, and the human pancreas data, respectively (Supplementary Table 10). Experiments show that the running time and random-access memory (RAM) usage are indeed linear in the number of genes $G$ and the number of cells $N$ for both the CPU and the GPU parallel version of BUSseq (Fig. 9 and Supplementary Note 13). Moreover, by writing the posterior samples to the hard disk every a few iterations, we can further reduce the

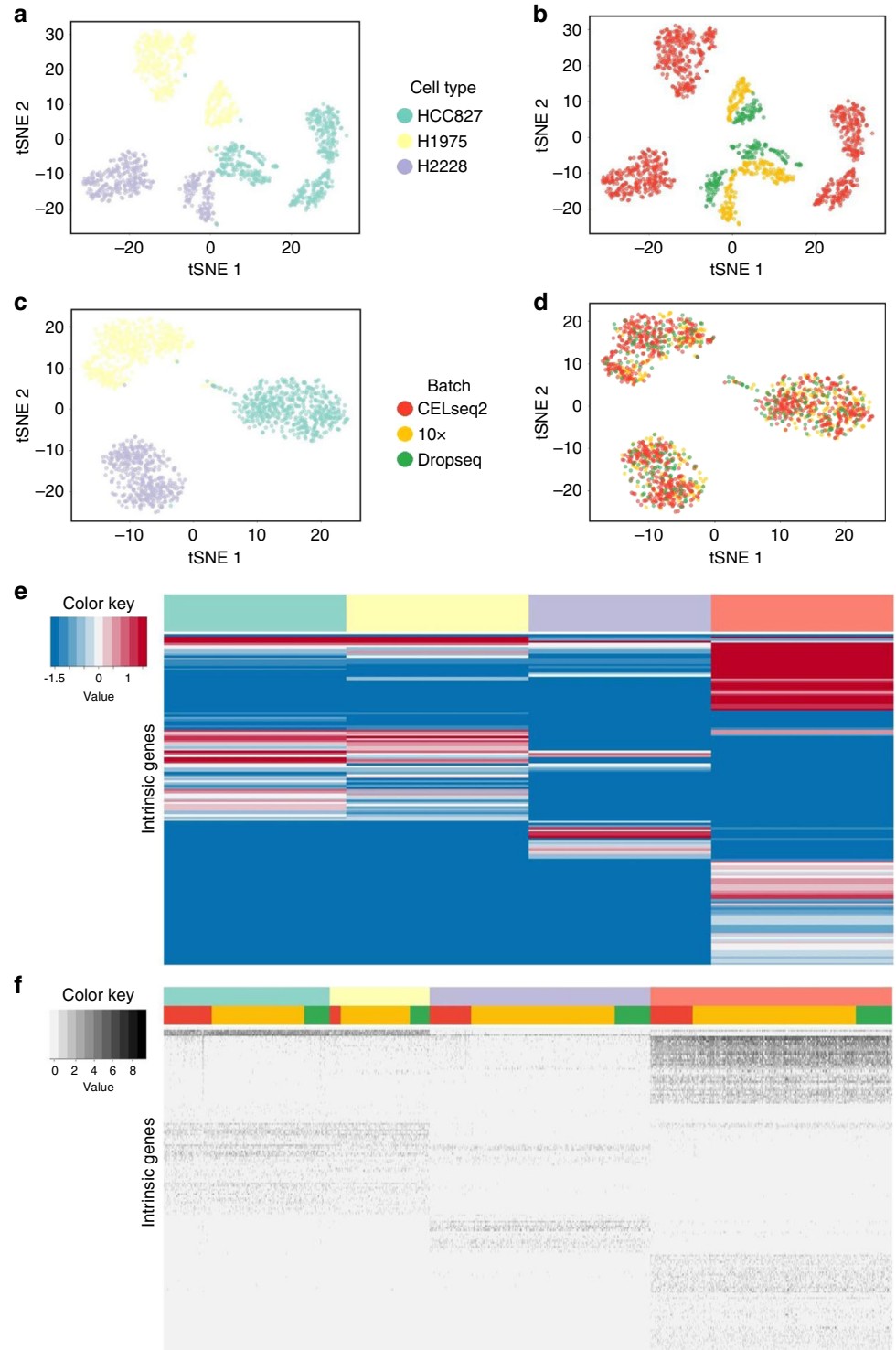

**Fig. 8 Patterns of the LUAD study. a** t-SNE plot of the raw count data colored by cell line (H1975, H2228 and HCC827). **b** t-SNE plot of the raw count data colored by protocol (CELseq2, 10x Chromium, and Dropseq). **c** t-SNE plot of the corrected count data by BUSseq colored by cell line. **d** t-SNE plot of the corrected count data by BUSseq colored by protocol. **e** Heatmap of log-scale mean expression levels $\boldsymbol{\alpha} + \boldsymbol{\beta}_k$ of cell types $k = 1, 2, 3, 4$. Each row represents an intrinsic gene. The colored bar indicates the cell type of each column. **f** Heatmap of the log-scale corrected count data by BUSseq. Each row again represents an intrinsic gene, but each column corresponds to a cell. The upper colored bar indicates the estimated cell type of each cell, and the lower colored bar denotes the batch to which the corresponding cell belongs.

RAM usage so that BUSseq is affordable by a commonly available cluster node rather than a high-end one (Supplementary Table 11 and Supplementary Fig. 15). Compared with the time for preparing samples and conducting the scRNA-seq experiments, the

computation time of BUSseq is affordable and worthwhile for the accuracy.

Practical and valid experimental designs are urgently required for scRNA-seq experiments. We envision that the flexible

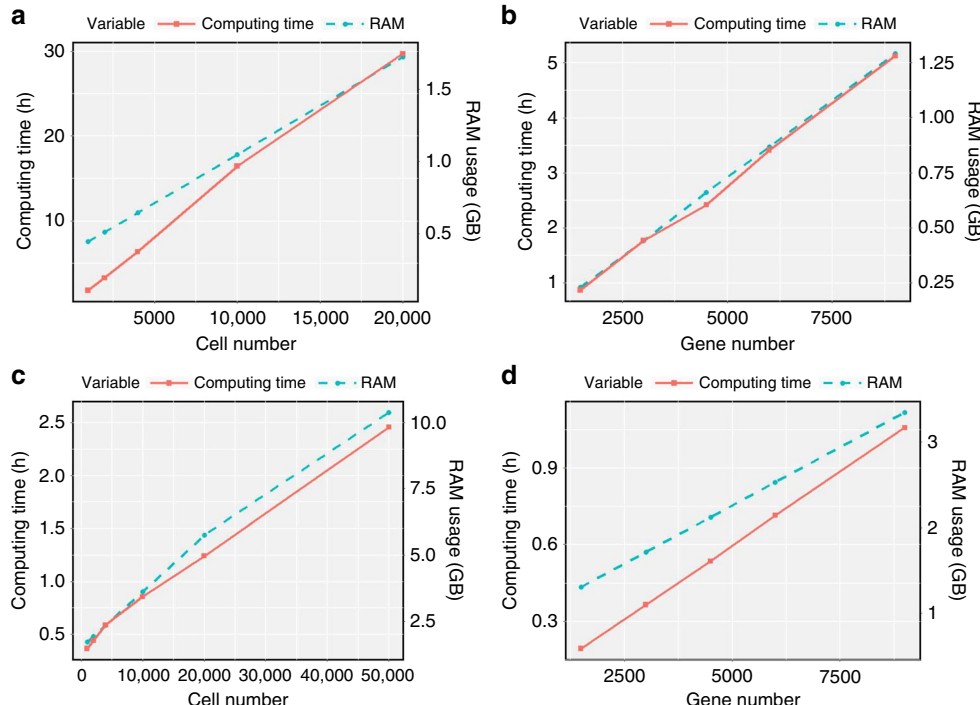

**Fig. 9 The trend of running time and RAM usage with respect to gene and cell numbers.** Running time and RAM usage for **a,b** the central processing unit (CPU) parallel version of BUSseq on eight cores of Dual Intel Xeon E5-2650 v2 2.60 GHz processors and **c,d** the graphics processing unit (GPU) parallel version of BUSseq on a single core of an Intel Xeon Gold 6132 processor and one NVIDIA Tesla P100 GPU. **a** When $G = 3000$ and $n_s = 1000$, $N$ varies from 1000 to 20,000. **b** When $N = 1000$ and $n_s = 1000$, $G$ varies from 1500 to 9000. **c** When $G = 3000$, $N$ varies from 1000 to 50,000. **d** When $N = 1000$, $G$ varies from 1500 to 9000.

reference panel and the chain-type designs will be widely adopted in scRNA-seq experiments and BUSseq will greatly facilitate the analysis of scRNA-seq data.

## Methods

**BUSseq model**. The hierarchical model of BUSseq can be summarized as:

$$\Pr(W_{bi} = k) = \pi_{bk}, \sum_{k=1}^{K} \pi_{bk} = 1;$$

$$X_{big}|W_{bi} = k \sim \mathrm{NB}(\mu_{big}, \phi_{bg}), \log(\mu_{big}) = \alpha_g + \beta_{gk} + \nu_{bg} + \delta_{bi};$$

$$Z_{big}|X_{big} = x_{big} \sim \mathrm{Bernoulli}(p_{big}), \log\left(\frac{p_{big}}{1-p_{big}}\right) = \gamma_{b0} + \gamma_{b1}x_{big};$$

$$Y_{big} = X_{big}|Z_{big} = 0, Y_{big} = 0|Z_{big} = 1.$$

Collectively, $\mathbf{Y} = \{Y_{big}\}_{b=1,\cdots,B;i=1,\cdots,n_b}^{g=1,\cdots,G}$ are the observed data; the underlying expression levels $\mathbf{X} = \{X_{big}\}_{b=1,\cdots,B;i=1,\cdots,n_b}^{g=1,\cdots,G}$, the dropout indicators $\mathbf{Z} = \{Z_{big}\}_{b=1,\cdots,B;i=1,\cdots,n_b}^{g=1,\cdots,G}$ and the cell type indicators $\mathbf{W} = \{W_{bi}\}_{b=1,\cdots,B;i=1,\cdots,n_b}$ are all missing data; the log-scale baseline gene expression levels $\boldsymbol{\alpha} = \{\alpha_g\}_{g=1,\cdots,G}$, the cell type effects $\boldsymbol{\beta} = \{\beta_{gk}\}_{k=2,\cdots,K}^{g=1,\cdots,G}$, the location batch effects $\boldsymbol{\nu} = \{\nu_{bg}\}_{b=2,\cdots,B}^{g=1,\cdots,G}$, the overdispersion parameters $\boldsymbol{\phi} = \{\phi_{bg}\}_{b=1,\cdots,B}^{g=1,\cdots,G}$, the cell-specific size factors $\boldsymbol{\Delta} = \{\delta_{bi}\}_{b=1,\cdots,B}^{i=2,\cdots,n_b}$, the dropout parameters $\boldsymbol{\Gamma} = \{\gamma_{b0}, \gamma_{b1}\}_{b=1,\cdots,B}$ and the cell compositions $\boldsymbol{\pi} = \{\pi_{bk}\}_{b=1,\cdots,B}^{k=1,\cdots,K}$ are the parameters. Without loss of generality, for model identifiability, we assume that the first batch is the reference batch measured without batch effects with $\nu_{1g} = 0$ for every gene and the first cell type is the baseline cell type with $\beta_{g1} = 0$ for every gene. Similarly, we take the cell-specific size factor $\delta_{b1} = 0$ for the first cell of each batch. We gather all the parameters as $\boldsymbol{\Theta} = \{\boldsymbol{\alpha}, \boldsymbol{\beta}, \boldsymbol{\nu}, \boldsymbol{\phi}, \boldsymbol{\Delta}, \boldsymbol{\Gamma}, \boldsymbol{\pi}\}$.

Consequently, the observed data likelihood function becomes

$$L_o(\boldsymbol{\Theta}|\mathbf{y}) = \prod_{b=1}^{B} \prod_{i=1}^{n_b} \left[\sum_{k=1}^{K} \pi_{bk} \prod_{g=1}^{G} \Pr(Y_{big} = y_{big}|\boldsymbol{\Theta})\right], \quad (1)$$

where

$$\Pr(Y_{big} = y_{big}|\boldsymbol{\Theta}) = \begin{cases} \sum_{x=1}^{\infty} \frac{\exp(\gamma_{b0}+\gamma_{b1}x)}{1+\exp(\gamma_{b0}+\gamma_{b1}x)} f_{\mathrm{NB}}(x; \exp(\alpha_g + \beta_{gk} + \nu_{bg} + \delta_{bi}), \phi_{bg}) \\ \quad + f_{\mathrm{NB}}(0; \exp(\alpha_g + \beta_{gk} + \nu_{bg} + \delta_{bi}), \phi_{bg}) & y_{big} = 0, \\ \frac{1}{1+\exp(\gamma_{b0}+\gamma_{b1}y_{big})} f_{\mathrm{NB}}(y_{big}; \exp(\alpha_g + \beta_{gk} + \nu_{bg} + \delta_{bi}), \phi_{bg}) & y_{big} > 0. \end{cases}$$

and $f_{\mathrm{NB}}(x; \mu, \phi) = C_x^{\phi+x-1} \left(\frac{\mu}{\mu+\phi}\right)^x \left(\frac{\phi}{\mu+\phi}\right)^\phi$ denotes the probability mass function of the negative binomial distribution $\mathrm{NB}(\mu, \phi)$. For $y_{big} = 0$, $f_{\mathrm{NB}}(0; \exp(\alpha_g + \beta_{gk} + \nu_{bg} + \delta_{bi}), \phi_{bg})$ corresponds to a biological zero, whereas $\sum_{x=1}^{\infty} \frac{\exp(\gamma_{b0}+\gamma_{b1}x)}{1+\exp(\gamma_{b0}+\gamma_{b1}x)} f_{\mathrm{NB}}(x; \exp(\alpha_g + \beta_{gk} + \nu_{bg} + \delta_{bi}), \phi_{bg})$ corresponds to a dropout event.

**Experimental designs**. By creating a set of functions similar to the probability generating function, we prove that BUSseq is identifiable, in other words, if two sets of parameters are different, then their probability distribution functions for the observed data are different, for not only the complete setting but also the reference panel and the chain-type designs (see the proofs in Supplementary Note 3).

**Theorem 1** (The Complete Setting) *If $\pi_{bk} > 0$ for every batch $b$ and cell type $k$, given that (I) $\gamma_{b1} < 0$ for every $b$, (II) for any two cell types $k_1$ and $k_2$, there exist at least two differentially expressed genes $g_1$ and $g_2$—$\beta_{g_1 k_1} \neq \beta_{g_1 k_2}$ and $\beta_{g_2 k_1} \neq \beta_{g_2 k_2}$, and (III) for any two distinct cell-type pairs $(k_1, k_2) \neq (k_3, k_4)$, their differences in cell-type effects are not the same $\beta_{k_1} - \beta_{k_2} \neq \beta_{k_3} - \beta_{k_4}$, then BUSseq is identifiable (up to label switching) in the sense that $L_o(\boldsymbol{\Theta}|\mathbf{y}) = L_o(\boldsymbol{\Theta}^*|\mathbf{y})$ for any $\mathbf{y}$ implies that $\pi_{bk} = \pi_{b\rho(k)}^*, (\gamma_{b0}, \gamma_{b1}) = (\gamma_{b0}^*, \gamma_{b1}^*), \alpha_g + \beta_{gk} = \alpha_g^* + \beta_{g\rho(k)}^*, \nu_{gb} = \nu_{gb}^*, \delta_{bi} = \delta_{bi}^*$ and $\phi_{bg} = \phi_{bg}^*$ for every gene $g$ and batch $b$, where $\rho$ is a permutation of $\{1, 2, \cdots, K\}$.*

In the following, we denote the cell types that are present in batch $b$ as $C_b$ and count the number of cell types existing in batch $b$ as $K_b = |C_b|$.

**Theorem 2** (The Reference Panel Design) *If there are a total of $K$ cell types $\cup_{b=1}^{B} C_b = \{1, 2, \cdots, K\}$, $K_b \geq 2$ for every batch $b$, and there exists a batch $\bar{b}$ such that it contains all of the cell types $C_{\bar{b}} = \{1, 2, \cdots, K\}$, then given that conditions (I)–(III) hold, BUSseq is identifiable (up to label switching).*

**Theorem 3** (The Chain-type Design) *If there are a total of $K$ cell types $\cup_{b=1}^{B} C_b = \{1, 2, \cdots, K\}$ and every two consecutive batches share at least two cell types $|C_b \cap C_{b-1}| \geq 2$ for all $b \geq 2$, then given that conditions (I)–(III) hold, BUSseq is identifiable (up to label switching).*

Therefore, even for the reference panel and chain-type designs that do not assay all cell types in each batch, batch effects can be removed; cell types can be clustered;

and missing data due to dropout events can be imputed. Both the reference panel design and the chain-type design belong to the more general connected design.

**Theorem 4** (*The Connected Design*) *We define a batch graph $G = (V, E)$. Each node $b \in V$ represents a batch. There is an edge $e \in E$ between two nodes $b_1$ and $b_2$ if and only if batches $b_1$ and $b_2$ share at least two cell types. If the batch graph is connected and conditions (I)–(III) hold, then BUSseq is identifiable (up to label switching).*

**Statistical inference.** We conduct the statistical inference under the Bayesian framework. We assign independent priors to each component of $\Theta$ as follows (Supplementary Table 3):

$$\pi_b = (\pi_{b1}, \cdots, \pi_{bK}) \sim \text{Dirichlet}(\xi, \cdots, \xi), 1 \leq b \leq B;$$
$$\gamma_{b0} \sim \text{N}(0, \sigma_{z0}^2), 1 \leq b \leq B;$$
$$-\gamma_{b1} \sim \text{Gamma}(a_\gamma, b_\gamma), 1 \leq b \leq B;$$
$$\alpha_g \sim \text{N}(m_a, \sigma_a^2), 1 \leq g \leq G;$$
$$\nu_{bg} \sim \text{N}(m_c, \sigma_c^2), 2 \leq b \leq B, g = 1, \cdots, G;$$
$$\delta_{bi} \sim \text{N}(m_d, \sigma_d^2), 1 \leq b \leq B, 2 \leq i \leq n_b;$$
$$\phi_{bg} \sim \text{Gamma}(\kappa, \tau), 1 \leq b \leq B, 1 \leq g \leq G.$$

We are interested in detecting genes that differentiate cell types. Therefore, we impose a spike-and-slab prior[55] using a normal mixture to the cell-type effect $\beta_{gk}$. The spike component concentrates on zero with a small variance $\tau_{\beta0}^2$, whereas the slab component tends to deviate from zero, thus having a larger variance $\tau_{\beta1}^2$. We introduce another latent variable $L_{gk}$ to indicate which component $\beta_{gk}$ comes from. $L_{gk} = 0$ if gene $g$ is not differentially expressed between cell type $k$ and cell type one, and $L_{gk} = 1$, otherwise. We further define $D_g = \sum_{k=2}^K L_{gk}$. If $D_g > 0$, then the expression level of gene $g$ does not stay the same across cell types. Following Huo et al.[13], we call such genes intrinsic genes, which differentiate cell types. To control for multiple hypothesis testing, we let $L_{gk} \sim \text{Bernoulli}(p)$ and assign a conjugate prior $\text{Beta}(a_p, b_p)$ to $p$. We set $\tau_{\beta1}$ to a large number and let $\tau_{\beta0}$ follow an inverse-gamma prior $\text{Inv—Gamma}(a_\tau, b_\tau)$ with a small prior mean.

We develop an MCMC algorithm to sample from the posterior distribution (Supplementary Note 2). After the burn-in period, we take the mean of the posterior samples to estimate $\gamma_b, \alpha_g, \beta_{gk}, \nu_{bg}, \delta_{bi}$, and $\phi_{bg}$ and use the mode of posterior samples of $W_{bi}$ to infer the cell type for each cell.

We have actually also implemented an Expectation-Maximization (EM) algorithm[56] for a simplified version of the BUSseq model. Unfortunately, consistent with the literature[57,58], we found that inference by the EM algorithm can be very sensitive to small disturbance of observed data and the initial values. Thus, we choose to use the MCMC algorithm for inference. The extra benefit of the MCMC algorithm is that it not only provides point estimates but also explores the entire posterior distributions and hence allow the users to quantify the uncertainty of estimates.

*Identification of intrinsic genes.* When inferring the differential expression indicator $L_{gk}$, we control the Bayesian FDR[37] defined as

$$\text{FDR}(\kappa) = \frac{\sum_{g=1}^G \sum_{k=2}^K \xi_{gk} I(\xi_{gk} \leq \kappa)}{\sum_{g=1}^G \sum_{k=2}^K I(\xi_{gk} \leq \kappa)},$$

where $\xi_{gk} = \text{Pr}(L_{gk} = 0|\mathbf{y})$ is the posterior marginal probability that gene $g$ is not differentially expressed between cell type $k$ and cell type one, which can be estimated by the $T$ posterior samples $L_{gk}^{(t)}$s collected after the burn-in period as $\frac{1}{T}\sum_{t=1}^T(1 - L_{gk}^{(t)})$. Given a control level $\alpha$ such as 0.1, we search for the largest $\kappa_0 \leq 0.5$ such that the estimated $\widehat{\text{FDR}}(\kappa)$ based on $\widehat{\xi}_{gk}$s is smaller than $\alpha$ and declare $\widehat{L}_{gk} = 1$ if $\widehat{\xi}_{gk} \leq \kappa_0$. The upper bound 0.5 for $\kappa_0$ prevents us from calling differentially expressed genes with small posterior probability $\text{Pr}(L_{gk} = 1|\mathbf{y})$. Consequently, we identify the genes with $\widehat{D}_g = \sum_{k=2}^K \widehat{L}_{gk} > 0$ as the intrinsic genes. We set $\alpha = 0.05$ in both the simulation study and the real applications. Here, we follow Huo et al.[13] to define intrinsic genes as genes that are differentially expressed between at least two cell types. In contrast, marker genes are genes that feature certain cell types according to the literature. For example, in the pancreas study, *GCG* gene is known to be highly expressed in alpha islet cells, so this gene often serves as a marker to label alpha islet cells[51].

*Convergence of the MCMC algorithm.* To rigorously assess the convergence of the Markov chain, we adopt the EPSR factors criterion[59] (Supplementary Note 14). We are interested in the log-scale baseline expression level $\{\alpha_g, g = 1, 2, \cdots, G\}$, the cell type effects $\{\beta_{gk}, g = 1, 2, \cdots, G, k = 2, 3, \cdots, K\}$, the location batch effects $\{\nu_{bg}, g = 1, 2, \cdots, G, b = 2, 3, \cdots, B\}$ and the overdispersion parameters $\{\phi_{bg}, g = 1, 2, \cdots, G, b = 1, 2, \cdots, B\}$. To avoid the impact of label switching of cell types (Supplementary Fig. 16 and Supplementary Note 14), we consider the log-scale cell-type-specific expression level $\theta_{gk} = \alpha_g + \beta_{gk}, g = 1, 2, \cdots, G, k = 1, 2, \cdots, K$ and

match the cell type indicators in different chains such that most cells in the different chains are assigned to the same cell types. If the EPSR factors of most parameters are close to one, we treat the posterior sampling as attaining stationary. Thus, we use the following rule to diagnose the convergence of the MCMC algorithm for BUSseq:

1. More than 80% of $\{\text{EPSR}(\theta_{gk})\}$ are <1.3;
2. More than 80% of $\{\text{EPSR}(\nu_{bg})\}$ are <1.3;
3. More than 80% of $\{\text{EPSR}(\phi_{bg})\}$ are <1.3.

*Implementation of the MCMC algorithm.* In the simulation study, we ran the MCMC algorithm for 4000 iterations and discarded the first 2000 iterations as burn-ins. In the three real data analysis, we ran BUSseq for 8000 iterations and discarded the first 4000 iterations as burin-ins. Both the estimated potential scale reduction (EPSR) factors (Supplementary Table 12) and the acceptance rates of the Metropolis steps of the MCMC algorithm (Supplementary Tables 13 and 14, and Supplementary Note 14) demonstrate that the Markov chain has converged with good mixing.

*Selection of cell type numbers.* BUSseq allows the user to input the total number of cell types $K$ according to prior knowledge. When $K$ is unknown, BUSseq selects the number of cell types $\widehat{K}$ such that it achieves the minimum BIC[31]. BIC adds a penalty term to the observed data log-likelihood $L_o(\widehat{\Theta}|\mathbf{y})$ as Eq. (1).

$$\text{BIC}(K) = -2\log(L_o(\widehat{\Theta}|\mathbf{y})) + [K(B + G) + 2B + (2B - 1)G + \sum_{b=1}^B(n_b - 1)] \cdot \log(\sum_{b=1}^B n_b G),$$

where $\widehat{\Theta} = (\widehat{\alpha}, \widehat{\beta}, \widehat{\gamma}, \widehat{\nu}, \widehat{\phi}, \widehat{\delta}, \widehat{\pi})$ denotes the posterior mean of parameters. As a result, the penalty in BIC helps the model selection to balance between goodness-of-fit and the model complexity (Supplementary Figs. 17–19, Supplementary Tables 15 and 16, and Supplementary Note 15).

*Inference of dropout events.* In the BUSseq model, a dropout event occurs for gene $g$ in cell $i$ of batch $b$ if the observed value $y_{big} = 0$ but the imputed count data $\widehat{x}_{big} > 0$. The identification allows us to calculate the frequency of dropout events in each batch. We calculate the zero rate of each batch as following:

$$\rho_0 = \frac{1}{G \cdot n_b} \sum_{g=1}^G \sum_{i=1}^{n_b} I(y_{big} = 0), \tag{2}$$

and compute the dropout rate as the proportion of dropout events among the observations with zero counts:

$$\rho_d = \frac{\sum_{g=1}^G \sum_{i=1}^{n_b} I(y_{big} = 0 \text{ and } \widehat{x}_{big} > 0)}{\sum_{g=1}^G \sum_{i=1}^{n_b} I(y_{big} = 0)}.$$

*Posterior predictive check.* We evaluate how well BUSseq fits the data via posterior predictive checks[60]. In particular, we focus on the zero rates. In the posterior predictive check, we take MCMC samples of all the parameters after the burn-in iterations to simulate replicated datasets $Y_j^{rep}, j = 1, 2, \cdots, J$ for $G$ genes and $N = \sum_{b=1}^B n_b$ cells, where $J$ denotes the total number of collected iterations after burn-ins. In our real data analyses, we ran 8000 iterations with the first 4000 iterations as burn-ins, so we generated $J = 8000 - 4000 = 4000$ replicated datasets for both the hematopoietic and Pancreas studies. For each generated replicate dataset, we calculated the zero rates of each batch according to Eq. (2). Finally, we average the zero rates over all the $J$ iterations to calculate the posterior mean $\widehat{\rho}_0$ of the zero rate of each batch and compare it with the corresponding observed zero rate. Moreover, we also compare BUSseq with a reduced model of BUSseq which ignores dropout events and hence uses negative binomial distribution without zero inflation, abbreviated as BUSseq-nzf (Supplementary Note 9), via the posterior predictive check (Supplementary Note 16).

**Batch-effects-corrected values.** To facilitate further downstream analysis, we also provide a version of count data $\widetilde{X} = \{\widetilde{X}_{big}\}_{b=1,\cdots,B;i=1,\cdots,n_b}^{g=1,\cdots,G}$ for which the batch effects are removed and the biological variability is retained similar to that of ComBat[7]. Ideally, if $x_{big}$ is the $\alpha$th percentile of $\text{NB}(\exp(\widehat{\alpha}_g + \widehat{\beta}_{g\widehat{w}_{bi}} + \widehat{\nu}_{bg} + \widehat{\delta}_{bi}), \widehat{\phi}_{bg})$, we aim to take the $\alpha$th percentile of $\text{NB}(\exp(\widehat{\alpha}_g + \widehat{\beta}_{g\widehat{w}_{bi}}), \widehat{\phi}_{bg})$ as the corrected value $\widetilde{x}_{big}$. However, the negative binomial distribution is a discrete distribution. As a result, several $\widetilde{x}$s can lie between the $\text{Pr}(x \leq x_{big} - 1)$-percentile and $\text{Pr}(x \leq x_{big})$-percentile of the distribution of $\widetilde{X}_{big}$. For example, if $X_{big} \sim \text{NB}(\exp(2), 3)$, $\widetilde{X}_{big} \sim \text{NB}(\exp(3), 5)$, and our observed value $x_{big} = 8$, then $\text{Pr}(x_{big} \leq 7)$ and $\text{Pr}(x_{big} \leq 8)$ correspond to the 58.67th and 65.76th percentiles of $\text{NB}(\exp(2), 3)$. However, three numbers—21, 22, and 23—lie between 58.67th and 65.76th percentile of $\text{NB}(\exp(3), 5)$. Thus, to avoid bias, we draw one number uniformly from 21, 22, and 23 rather than take the maximum or the minimum to calculate $\widetilde{x}_{big}$.

Thus, we develop a quantile matching approach based on inverse sampling. Specifically, given the fitted model and the inferred underlying expression level $\widehat{x}_{big}$, we first sample $u_{big}$ from $\mathrm{Unif}[F_{\mathrm{NB}}(\widehat{x}_{big}-1;\exp(\widehat{\alpha}_g+\widehat{\beta}_{g\widehat{w}_{bi}}+\widehat{\nu}_{bg}+\widehat{\delta}_{bi}),\widehat{\phi}_{bg}),$ $F_{\mathrm{NB}}(\widehat{x}_{big};\exp(\widehat{\alpha}_g+\widehat{\beta}_{g\widehat{w}_{bi}}+\widehat{\nu}_{bg}+\widehat{\delta}_{bi}),\widehat{\phi}_{bg})]$ where $\mathrm{Unif}[a,b]$ denotes the uniform distribution on the interval $[a,b]$ and $F_{\mathrm{NB}}(\,\cdot\,;\mu,r)$ denotes the cumulative distribution function of a negative binomial distribution with mean $\mu$ and overdispersion parameter $r$. Next, we calculate the $u_{big}^{th}$ quantile of $\mathrm{NB}(\exp(\widehat{\alpha}_g+\widehat{\beta}_{g\widehat{w}_{bi}}),\widehat{\phi}_{1g})$ as the corrected value $\widetilde{x}_{big}$.

The corrected data $\widetilde{\mathbf{X}}$ are not only protected from batch effects but also impute the missing data due to dropout events. Moreover, further cell-specific normalization is not needed. Meanwhile, the biological variability is retained thanks to the quantile transformation and sampling step. Therefore, we can directly perform downstream analysis on $\widetilde{\mathbf{X}}$.

### Preprocessing of the real datasets

*Gene filtering.* A common practice of scRNA-seq data analysis is to focus on the set of HVGs[14,17,18,24,25] or the genes with the high mean expression levels across cells[61]. Although BUSseq is robust to gene filtering strategies in real studies (Supplementary Tables 17 and 18), comparing the ARIs resulting from the two gene filtering strategies, we recommend filtering HVGs in preprocessing (Supplementary Note 17). An intrinsic gene that well distinguishes cell types may be highly expressed in one cell type but lowly expressed in other cell types. As a result, its mean expression level across all of the cells may be low, and hence such a gene will be missed by filtering according to mean expression levels. Thus, filtering genes according to mean expression levels is likely to select genes whose expression levels are high but remain the same across all of the cell types. Unfortunately, such genes can provide very limited information for differentiating cell types. We therefore filter out HVGs for the downstream analysis in real data analyses.

*Hematopoietic study.* For the two hematopoietic datasets, we downloaded the read count matrix of the 1920 cells profiled by Paul et al.[44] and the 2729 cells labeled as myeloid progenitor cells by Nestorowa et al.[43] from the NCBI Gene Expression Omnibus (GEO) with the accession numbers GSE72857 and GSE81682. Following Brennecke et al.[62], we first labeled cells using FACS labels and then performed the size factor normalization within each batch. Next, we filtered out the common HVGs identified by Nestorowa et al.[43] between two datasets. These HVGs were denoted by Ensembl ID. The genes in the GSE81682 dataset were named by Ensembl ID, but the genes in the GSE72857 dataset were named by Gene Symbol. The R package biomaRt was used to query the corresponding Gene Symbol by Ensembl ID. Finally, we obtained 3,470 common HVGs shared by the two datasets.

*Pancreas study.* Two of the pancreas datasets profiled by the CEL-seq2 platform were downloaded from GEO with accession number GSE80150[50] and GSE85241[63]. The two datasets assayed by the SMART-seq2 platform were obtained from GSE86473[51] and from ArrayExpress accession number E-MATB-5061[52]. Following Haghverdi et al.[14], we excluded cells with low library sizes (<100,000 reads), low numbers of expressed genes (>40% total counts from ribosomal RNA genes), or high ERCC content (>20% of total counts from spike-in transcripts) resulting in 7095 cells. We selected the 2480 HVGs shared by the four datasets according to Brennecke et al.[62] by sorting the ratio of variance and mean expression level after adjusting technical noise with the variances of spike-in transcripts. GSE86473 and EMATB-5061 have the cell type labels for all of the cells, but the cell type labels of GSE81076 and GSE85241 were inferred by the marker genes used in the original publications by Lawlor et al.[51] and Grün et al.[50].

To assign cell type labels for the GSE81076 and GSE85241 datasets, following Haghverdi et al.[14], we first extracted the normalized expression levels of the selected HVGs within each dataset. Next, we obtained the low dimensional embedding of HVGs by tSNE for visualization. At the same time, we applied robust k-means clustering to the normalized expression levels of the selected HVGs using the pam function in the R package cluster. The number of clusters was set as nine. Next, we drew t-SNE plots colored by the expression levels of the marker genes. It is known that *GCG* is highly expressed in alpha islet cells, *INS* in beta islet cells, *SST* in delta islet cells, *PPY* in gamma islet cells (pancreatic polypeptide cells), *PRSS1* in acinar cells, *KRT19* in ductal cells and *COL1A1* in mescenchymal cells[50,51], so we labeled each cluster by its corresponding highly expressed marker gene.

*LUAD cancer cell line study.* We downloaded the raw count data from the GitHub repository https://github.com/LuyiTian/sc_mixology with accession number GSE118767. We selected the top 6000 HVGs within each batch using the trendVar and decomposeVar functions in the R package scran[64] and obtained 2,267 common HVGs across three batches (Supplementary Note 15).

### Naming clusters learned by BUSseq according to FACS labels

In the two real data examples, we first identify the cell type of each individual cell according to FACS labeling. Then, for each cluster learned by BUSseq, we calculate the proportion of labeled cell types. If a cell type accounts for more than one-third of the cells in a given cluster, we assign this cell type to the cluster. Although a cluster may be assigned more than one cell type, most identified clusters by BUSseq are dominated by only one cell type. For example, in the hematopoietic study, BUSseq identifies 1165 cells for Cluster 5. According to FACS labels, 1127 of the 1165 cells are megakaryocyte-erythrocyte progenitors (MEP). Therefore, we name Cluster 5 as MEP.

### Mapping clusters to haemopedia

Haemopedia is a database of gene expression profiles from diverse types of hematopoietic cells[46]. It collected flow sorted cell populations from healthy mice.

To understand Cluster 3 learned by BUSseq for the hematopoietic data, which is dominated by cells classified as other according to the FACS labeling, we mapped the cluster means learned by BUSseq to the Haemopedia RNA-seq dataset.

We first applied TMM normalization[35] to all the samples in the Haemopedia RNA-seq dataset. Then, we extracted seven types of hematopoietic stem and progenitor cells from Haemopedia, including $\mathrm{Lin^-Sca\text{-}1^+c\text{-}Kit^+}$ cells, short-term hematopoietic stem cells, MPP, CLP, CMP, MEP, and GMP. Each selected cell type had two RNA-seq samples in Haemopedia, so we averaged over the two replicates for each cell type. Further, we added one to the normalized expression levels as a pseudo read count to handle genes with zero read count and log-transformed the data. Finally, we scaled the data across the seven cell types for each gene. To be comparable, we transformed the cluster mean learned by BUSseq as $m_{gk} = \log(1+\exp(\alpha_g+\beta_{gk}))$ for gene $g$ in the cluster $k$ and scaled $m_{gk}$ across all cell types as well. Finally, we calculated the correlation between the cluster means inferred by BUSseq and the reference expression profiles in Haemopedia for 37 marker genes. The 37 marker genes were retrieved from Paul et al.[44] (31 maker genes for HSPC) and Herman et al.[45] (6 maker genes for LMPP).

## Software availability

The C++ source code of the parallel multi-core-CPU version of BUSseq is available on GitHub https://github.com/songfd2018/BUSseq-1.0, and the CUDA C source code of the GPU version of BUSseq is available on GitHub https://github.com/Anguscgm/BUSseq_gpu. All code for producing results and figures in this manuscript are also available on GitHub (https://github.com/songfd2018/BUSseq-1.1_implementation). Furthermore, we wrap C++ source code as an R package, BUSseq (https://github.com/songfd2018/BUSseq-Rpackage).

**Reporting summary.** Further information on research design is available in the Nature Research Reporting Summary linked to this article.

## Data availability

The published datasets used in this manuscript are available through the following accession numbers: SMART-seq2 platform hematopoietic data with GEO GSE81682 by Nestorowa et al.[43]; MARS-seq platform hematopoietic data with GEO GSE72857 by Paul et al.[44]; CEL-seq platform pancreas data with GEO GSE81076 by Grün et al.[50]; CEL-seq2 platform pancreas data with GEO GSE85241 by Muraro et al.[63]; SMART-seq2 platform pancreas data with GEO GSE86473 by Lawlor et al.[51]; and SMART-seq2 platform pancreas data with ArrayExpress E-MTAB-5061 by Segerstolpe et al.[52]; human lung adenocarcinoma cell line data with GEO GSE118767 by Tian et al.[65]. The parameter settings for the simulation study and the simulated data are available on GitHub (https://github.com/songfd2018/BUSseq-1.1_implementation).

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

## Acknowledgements

This work was supported by the Hong Kong Ph.D. Fellowship PF15-17417 and the General Research Funds 14306417 and 14305319 from the Hong Kong Research Grants Council of the Hong Kong Special Administrative Region of the People's Republic of China and Direct Grants from the Research Committee of the Chinese University of Hong Kong. We acknowledge Dr. Xiangyu Luo for helpful comments on an early version of our paper.

## Author contributions

F.S. developed the method and the proof, implemented the algorithm, prepared the software package, analyzed the data, and wrote the paper. G.M.A.C. implemented the algorithm and analyzed the data. Y.W. conceived and supervised the study, developed the method and the proof, and wrote the paper.

## Competing interests

The authors declare no competing interests.
