## [Peer Review File · Nature Communications]

Reviewers' comments:

Reviewer #1 (Remarks to the Author):

Song et al introduces BUSseq, an extension to BUS which is adapted to capture key features of scRNAseq datasets. BUSseq is a Bayesian hierarchical model that propagates statistical uncertainty across several steps of typical scRNAseq analysis pipelines: normalisation, batch effect correction, clustering, differential expression and imputation. Inference is implemented using MCMC, using parallel computing to enable scalability.

Batch effects are a big issue in scRNAseq datasets, particularly when dealing with patient data for which randomized experimental designs are typically not possible. As such, I believe that the method could be of great interest. It is also great to see that the authors have posted their code online (including the code that was used as part of the data analyses). However, there are some aspects of the performance comparison and the methodology that need to be revised before publication. Specific comments are provided below.

Major comments.

1. Page 4. The model assumes that a gene- and batch-specific over-dispersion parameter are shared across all clusters of cells. However, it is well known that highly expressed genes tend to have lower over-dispersion with a population of cells. Therefore, if a gene is differentially expressed between two cell types, one would expect different over-dispersion values for each cell type. To address this, I suggest the authors to: (i) assess how robust is BUSseq when their assumption does not hold and (ii) explore how well does the model recapitulate the properties of real scRNAseq data (such as the mean/over-dispersion trend, see also [1]), within each cluster and when combining all clusters.

2. Page 8. The performance of BUSseq, including the ability to identify the correct value for K (via BIC), is assessed using synthetic data. It seems that the data was generated using the BUSseq model. If that is the case, the comparison with other methods is biased. I suggest the authors to use instead datasets generated via alternative models (such as [2]) and the control datasets introduced by [2], assessing the performance of BUSseq across different levels of sparsity (e.g. 10X vs smart-seq2 data) and sample complexity (how distinct are the clusters). These comparisons will enable users to better understand the strengths and limitations of BUSseq (e.g. choice of K can be more challenging for less distinct clusters).

3. Page 11, analysis of hematopoietic dataset. MNN (and others) do not perform clustering. How were the cluster labels assigned to calculate the ARI with respect to the true labels? If different clustering methods are used across the methods, this could affect the comparison. Instead, the authors might consider applying the same clustering method using the "corrected" data returned by each method. Same comment applies to the pancreatic dataset.

4. Figure 4. tSNE and PCA plots for the uncorrected data do not match those reported by Haghverdi et al (Figure 3). Whilst this might be expected for tSNE (which results depend on a random seed), this is not the case for PCA. Please explain why do these differences arise. Similar comment for the pancreatic data.

5. MCMC implementation. (i) how were the prior hyper-parameters chosen and how does this affect posterior inference?, (ii) please provide diagnostic criteria to assess the convergence of the MCMC for all data analyses, (iii) how does the method scale in terms of the number of genes or cells? (iv) please describe what steps are parallelised in the GPU implementation.

6. Supplementary methods, MCMC: (i) for all MH steps, the authors should report what is the acceptance rate? The latter enables the reader to assess how well is the MCMC mixing [3] (ii) in the updates for x , the authors indicate that variance of the proposal varies across iterations. If this is the case, the proposal is not symmetric and there is a missing term in the acceptance probability. Moreover, please do include the derivation of the acceptance probability (some of the terms from the equation above appear to be missing).

Minor comments.

1. Page 2, first paragraph “an excessive number of zeros that result from dropout events”. While scRNAseq experiments indeed exhibit a high frequency of zero counts, not all zeros are missing data or “dropouts” (a gene’s expression is not captured by the assay). Instead, some of these can correspond to “biological” zeros, where the gene is not expressed in a cell. The latter is particularly relevant when mix populations (with transcriptionally distinct cell types) are processed. I believe it is important to discuss this distinction and how the different types of zeros would be treated by BUSseq.
2. Page 2, second paragraph “selecting control genes is often difficult”. Recently, [4] identified control genes that are reproducible and conserved across species (human and mouse). Such genes could be used as a reference in RUVseq or related methods. I suggest the authors to include the reference to [4], discussing that there has been some progress in the area.
3. Page 4, last paragraph. The authors indicate that MCMC was used to perform inference. However, the reference provided is generic (a book which introduces several MCMC methods). Please state the specific MCMC method here.
5. Page 23. As BUSseq is also introduced as an imputation and normalisation method, it would be good to see (for a single batch) how does the corrected data compare to what is generated by existing normalisation (e.g. [5]) and imputation tools (such as those discussed in [6]).
6. Supplementary material “Comparison measures” section. The authors specify that silhouette metrics were calculated based on tSNE coordinates. However, tSNE is known to not preserve the global structure of the data (such as the separation between clusters). As such, tSNE coordinates do not provide an appropriate metric for clustering performance.
7. Implementation. In order to facilitate adoption, the authors might consider to implement their method as an R/Python library. In particular, submission to Bioconductor would support usage across different OS (Github repository states that current implementation requires Linux OS).

References

- [1] doi:10.1186/s13059-017-1305-0
- [2] doi: 10.1038/s41592-019-0425-8
- [3] <https://doi.org/10.1198/jcgs.2009.06134>
- [4] <https://doi.org/10.1093/gigascience/giz106>
- [5] doi:10.1186/s13059-016-0947-7
- [6] <https://doi.org/10.12688/f1000research.16613.2>

Reviewer #2 (Remarks to the Author):

Summary

This paper provides two significant contributions to the field of single-cell gene expression analysis (scRNA-seq). First, the authors provide rigorous conditions and assumptions that must be met for cell types to be identifiable in the presence of batch effects. Secondly, they provide an integrated hierarchical Bayesian model for batch effect correction and clustering. The authors do a good job of using examples to clarify new concepts as they are presented. Both the theoretical results and the method are well motivated, convincing, and I believe they would be interesting and useful to other researchers. I am especially grateful to the authors for making all of their code publicly available, and for including a detailed algorithm in the supplementary materials that will facilitate reproducibility. Despite the overall quality of the paper, I have several concerns especially in how the authors benchmarked against other methods and in how applicable their method will be to large-scale datasets that are becoming prevalent in the field.

Major concerns

* The assumption of zero inflation or dropout on top of the negative binomial sampling is controversial. Recent studies have claimed that there is no zero inflation in droplet scRNA-seq with UMIs, for example: <https://www.biorxiv.org/content/10.1101/582064v1.abstract> . The authors do not distinguish between UMI counts and read counts even though these two have dramatically different distributional properties, including but not limited to differences in zero inflation. If this paper were restricting its scope to read count data only (eg SMART-seq2) I would not raise this concern, but they do include some MARS-seq and CEL-seq2 datasets which to my understanding do have UMIs. Therefore, I would like to see a more careful discussion of the pros and cons of the zero inflation component with UMI vs non-UMI datasets. Specifically, I would like to see them re-fit their model on the UMI datasets without the zero inflation component and assess which model performed better. It would certainly increase the number of users of their method if they allow the user to decide which samples/ batches should have zero inflation (eg read count data) and which should not (eg UMI count data) and it would likely be faster to run since there would be fewer parameters. By providing both options and demonstrating which is effective on different datasets, they would bypass the controversy over whether they should or should not use zero inflation. Also, whenever a dataset is first described, they should say whether it has UMIs or not.

* The choice of filtering for informative genes plays a crucial role in downstream analysis. The authors should compare how sensitive their clustering algorithm is to alternative gene filtering methods besides the Brennecke one they mention. At a minimum, they should try just filtering on something like "highly expressed" genes as described in this systematic comparison: <https://f1000research.com/articles/7-1141/v2> . It would also be nice to see how sensitive BUSseq accuracy is to different choices of the number of genes included in the filtering.

* I appreciate that the authors mention the computational limitations of MCMC in the discussion/conclusion section. However, this is not sufficient. While it is true that 1 hour of computation time is small compared to wet lab experiments, this is on a fairly small dataset by current standards and was done on a high-performance computing system with a GPU. If the authors want their method to become widely used, I would encourage them to offer an approximate version of BUSseq that runs much faster. For example, if they discard the zero inflation component and perform maximization of the penalized likelihood using expectation-maximization, or variational inference they may get a result that is "good enough" in a fraction of the time. Even if it is not feasible to implement such an alternative inference scheme in the current paper, they should at least offer suggestions for how it could be done in the discussion, so that others can build on their ideas. Furthermore, they should address memory consumption of the parallelized and non-parallelized implementation. What is the maximum number of cells/genes that BUSseq can handle on an ordinary desktop computer?

* In comparing against other methods, the simulation is not very convincing, because it is drawn from the BUSseq generative model. They should also try a simulation where the assumptions of BUSseq are at least slightly violated. Also, the numbers of zeros in the simulation are unrealistically low. They should have at least one batch (preferably most of the batches) where the number of zeros is >80%.

* They should simulate a much larger number of genes (say 20,000 instead of 3,000) with many (say 17,000) having low information/ noise and batch effects only, then apply the same gene filtering approach used in their real data analysis prior to running the methods on the filtered dataset.

* The real data comparisons do not use any droplet technologies. If this is intentional, the authors should explicitly say their method is intended for plate-type scRNA-seq and doesn't apply to droplet scRNA-seq such as 10x, drop-seq, indrop etc. Otherwise, they need to include a droplet dataset in their comparisons.

* In comparing against methods such as ZINB-WAVE, they need to be more explicit about what the number of latent factors was set to and what clustering algorithm was applied to the factors. It would be easy to make a dimension reduction method look bad by setting the number of PCs too low then applying k-means clustering. For a realistic comparison they should vary the number of

PCs and use a Louvain or Leiden-type algorithm downstream of all methods that don't produce clusters automatically. They need to generally do a better job of explaining how the competing methods produce clusters in the introduction. It's OK if this is put in the supplement but currently it's nowhere in the manuscript.

* Many scRNA-seq experiments seek to uncover rare cell types. The authors should artificially reduce the number of cells in one or more of the cell types from a real data comparison and see whether/how the performance of BUSseq changes as the number of cells in each type becomes unevenly distributed. The theoretical results suggest that each pair of batches has to contain at least two overlapping cell types, but what happens if one or both of these cell types are only represented by say 3 cells in each batch? This could be related to the experimental design concepts by showing how "power" to detect cell types changes as a function of "sample size".

* Methods "Assignment of FACS cell type labels to learned clusters"- the authors need to clarify whether they did this label assignment before or after computing adjusted Rand index (ARI) or other measures of cluster accuracy. ARI does not require the labels of clusters to match between the two partitions being compared. The assignment of FACS labels is appropriate only for interpretation of the clusters, not for assessment of accuracy.

Minor Concerns

* This paper seems to be solving two separate problems that are not very intrinsically linked. I wonder if the theoretical discussion about chain-type and reference panel designs, along with the detailed mathematical proofs, could be better suited to a statistical journal. I admit I do not have a deep expertise in these types of proofs so I'm not able to do a thorough job verifying them. Intuitively, I think the theoretical arguments make sense, I just don't see how they are really necessary to present the methods advance that is the most useful part for applications. My concern is that readers will think this paper is about experimental design when it is really a new method for simultaneous clustering and batch effects correction. My suggestion is the authors consider which of the two parts is the main focus of the paper and either split off the other part into a separate publication or reduce its prominence in the title/abstract so readers will not be confused.

* They should explain what are the experimental designs for the simulated and real datasets in the paper- chain-type, reference panel, etc.

* The figures have way too many sub-panels making it very difficult to understand the main message. Please simplify and move the non-essential parts to the supplemental figures. Perhaps only show a comparison to one or two competing methods in the main figure instead of all methods.

* The γ_{b1} parameter is the log-odds ratio not the odds ratio.

* Fig 1a orange and gray background makes the math hard to read.

* p. 6, second-to-last paragraph: the sentence "the latter further relaxes the assumption implied by the former", I have no idea what "the latter" and "the former" is referring to. Please make this more clear. Also, it seems like the authors are using the terms "identical" and "completely randomized" interchangeably here, whereas these terms do not mean the same thing.

* p. 7, last paragraph: "dropout rate depends on the underlying expression levels" needs a citation. Also, the sentence containing "creating a set of functions similar to the probability generating function" is confusing and probably unnecessary. Just say something like "we prove identifiability of BUSseq in theorems XXX in the supplement".

* p. 8 simulation- how many genes were differentially expressed? were any genes just set to be pure noise across cell types with only batch effects? How many cells per true cluster?

* What is the distinction between "intrinsic genes" and "marker genes"?

* How did the authors compute a BIC score from MCMC samples? Did they simply use a single draw as θ ? Or pick a draw that had a high log-likelihood?

* They should clarify whether the MCMC has to be run separately for each value of "k" used to compute the BIC for model selection. Since this is not a reversible jump or Bayesian nonparametric method I assume yes. Also, along those lines, why not use something like marginal likelihood instead of BIC?

- * They need to explain how they chose the perplexity parameter for tSNE, and whether PCA was run prior to tSNE as an initialization (it almost always is since tSNE cannot handle more than about 100 dimensions).
- * Are all the results from the simulation study from a single simulated dataset? Or did they draw multiple replicates and average the results across them?
- * Fig 3- what is the take-home message here? Please include the main conclusion in the figure legend.
- * Optionally, the authors could validate their model fit on the real data analyses by using posterior predictive checks. This would be interesting especially in comparing the zero inflated vs non-zero inflated versions of BUSseq, but would not be useful in comparing against non-Bayesian methods.
- * p. 11- "t-SNE plots confirm that BUSseq performs the best". This is really not convincing. tSNE plots are exploratory not confirmatory, and are known to easily produce artificial clusters. The ARI and silhouette scores are sufficient to make this claim.
- * Fig 4- what is the take-home message? Please include in the figure legend the main conclusion. What normalization was used prior to PCA?
- * Fig 5e- please use a different color scale, the white-red scale makes it hard to see the individual points. I would suggest blue-red or gray-red. Also what are the units of "expression" here?
- * p. 13- arguing that BUSseq preserves the differentiation trajectory is not very convincing given that it is essentially a clustering method. A better approach would be to use BUSseq clusters as input to a method like slingshot and compare that way. I would actually recommend just removing the whole discussion of differentiation trajectories as it seems tangential to the main arguments.
- * I would like to see them compare against the cellassign method (<https://www.nature.com/articles/s41592-019-0529-1>), which is semi-supervised and as such would be expected to outperform BUSseq since it can use more information (in the form of marker gene reference panels). Even though it's not a fair comparison, it would be very useful for the community to see the pros and cons of conducting clustering de novo (as in BUSseq) versus using reference panels to map cells to known types.
- * Methods p. 23- "use the mode of posterior samples of $W_{\{bi\}}$ to infer the cell type". Is this affected by label switching? What if there are two possible cluster assignments that have equal probability mass (ie more than one mode)? Did this ever happen in their experiments?
- * Methods "batch effects corrected values"- I'm confused why the authors want to draw random samples to produce a "batch corrected" dataset. This will still have zeros and other features that make it hard to analyze with something like PCA (eg we are going to have to log transform and/or try to do variance stabilization). Why not just use $(\alpha + \beta)$ as the imputed values because these are real-valued and likely have a nicer distribution than the negative binomial?

Referee Information

F. William Townes
 Department of Computer Science
 Princeton University

We thank the two reviewers and the editor for their thorough review, insightful comments and constructive suggestions. We appreciate their positive comments, such as “*capture key features*,” “*be of great interest*,” “*rigorous conditions and assumptions*,” and “*well motivated, convincing*.” We have taken the reviewers’ comments very seriously, added extensive simulation studies and comparisons, prepared a user-friendly software package, and revised the manuscript and Supplementary Materials accordingly.

Below, we present point-by-point responses to the reviewers’ comments. The comments made by the two reviewers and the associate editor are displayed in *blue italics*. In the main manuscript, we have highlighted our modifications in **red**.

Reviewer 1

Song et al introduces BUSseq, an extension to BUS which is adapted to capture key features of scRNAseq datasets. BUSseq is a Bayesian hierarchical model that propagates statistical uncertainty across several steps of typical scRNAseq analysis pipelines: normalisation, batch effect correction, clustering, differential expression and imputation. Inference is implemented using MCMC, using parallel computing to enable scalability.

Batch effects are a big issue in scRNAseq datasets, particularly when dealing with patient data for which randomized experimental designs are typically not possible. As such, I believe that the method could be of great interest. It is also great to see that the authors have posted their code online (including the code that was used as part of the data analyses). However, there are some aspects of the performance comparison and the methodology that need to be revised before publication. Specific comments are provided below.

We really appreciate the positive comments from the reviewer.

Major comments.

1. Page 4. The model assumes that a gene- and batch-specific over-dispersion parameter are common across all clusters of cells. However, it is well known that highly expressed genes tend to have lower over-dispersion with a population of cells. Therefore, if a gene is differentially expressed between two cell types, one would expect different over-dispersion values for each cell type. To address this, I suggest the authors to: (i) assess how robust is BUSseq when their assumption does not hold and (ii) explore how well does the model recapitulate the properties of real scRNAseq data (such as the mean/over-dispersion trend, see also [1]), within each cluster and when combining all clusters.

We thank the reviewer for the suggestions.

Following the reviewer’s advice, we first generated simulated data to evaluate the robustness of BUSseq when the overdispersion parameter is not only gene- and batch-specific but also cluster-specific. In our previous simulation setting, we assumed that each gene had the same overdispersion parameters ϕ_{bg} within each batch for all cell types.

In the new simulation setting, we let the overdispersion parameters vary across cell types. We regard a gene as highly expressed (HEG) in a given cell type if its expression level in that cell type is higher than its mean expression level across all cell types. In the new simulation dataset, the $K = 5$ cell types have 150, 100, 110, 100 and 155 HEGs, respectively. Under our parameterization of the negative binomial distribution, $f_{NB}(x; \mu, \phi) = C_x^{\phi+x-1} (\frac{\mu}{\mu+\phi})^x (\frac{\phi}{\mu+\phi})^\phi$, the variance is $\mu + \frac{\mu^2}{\phi}$. Thus, a gene with higher ϕ_{bg} has lower overdispersion. If a gene g is highly expressed in a given cell type, we set its overdispersion parameter ϕ_{bg}^{High} in that cell type to be five times its overdispersion parameter ϕ_{bg}^{Low} in cell types where gene g is not highly expressed. **Figure R1** shows that even though HEGs have much lower true cell type specific overdispersion parameters, BUSseq is still able to successfully recover the log-scale mean expression levels for each cell type $\alpha + \beta$, the location batch effects ν , the underlying expression levels before dropout events \mathbf{X} and the cell-specific size factors δ .

We present the results on pages 12-13 of the main manuscript and pages 57-58 of the Supplementary Material.

Figure R1: Patterns of the simulation dataset in the sensitivity analysis which allows cell-type-specific overdispersion parameters. **(a)** True log-scale mean expression levels $\alpha + \beta$ for each cell type. Each row represents a gene, and each column corresponds to a cell type. **(b)** True batch effects ν . Each row represents a gene, and each column corresponds to a batch. **(c)** True underlying expression levels \mathbf{X} . Each row represents a gene, and each column corresponds to a cell. The upper colored bar indicates the batch of each cell, and the lower colored bar represents the type of each cell. **(d)** The simulated observed data \mathbf{Y} . **(e)** The estimated log-scale mean expression levels $\hat{\alpha} + \hat{\beta}$ for each cell type. **(f)** Estimated batch effects $\hat{\nu}$. **(g)** Imputed expression levels $\hat{\mathbf{X}}$. **(h)** Corrected count data $\tilde{\mathbf{X}}$ grouped by batch. **(i)** Scatter plot of the estimated cell-specific size factor versus the true cell-specific size factor. Finally, we draw **(j)** the principal component analysis (PCA) plot for each cell colored by cell type, **(k)** the t-distributed stochastic neighbor embedding (t-SNE) plot colored by cell type and **(l)** the t-SNE plot colored by batch indicators based on the corrected count data shown in **(h)**. BUSseq precisely recovers the true parameter values and perfectly clusters cells by cell type even if the overdispersion parameter is not only gene- and batch-specific but also cell type specific, with HEGs having lower overdispersion.

We further investigated the mean-variance trend of the fitted model, following the reviewer's suggestions. Zappia et al. [47] proposed to use Splatter, an R Bioconductor package, to simulate a scRNA-seq dataset and compare its properties with those of real data.

As the first batch of each study is taken as the reference batch without batch effects, without loss of generality, here we show the mean-variance trends for the second batch $b = 2$ of both real datasets as illustrations. In the hematopoietic study, BUSseq identifies 6 cell types, with 234, 244, 216, 331, 257 and 638 cells, respectively. To generate the simulated count data for each cell type, we first calculated the estimated mean expression level $\hat{m}_{2ig} = \hat{\alpha}_g + \hat{\beta}_{g, \hat{w}_{2i}} + \hat{\nu}_{2g} + \hat{\delta}_{2i}$ for gene g in cell i . We then sampled the underlying expression level \hat{X}_{big} from a negative binomial distribution with mean \hat{m}_{2ig} and overdispersion $\hat{\phi}_{2g}$. Next, we generated the dropout indicators \hat{Z}_{2i} from a Bernoulli distribution with probability $\frac{\exp(\hat{\gamma}_{20} + \hat{\gamma}_{21} \hat{x}_{2ig})}{1 + \exp(\hat{\gamma}_{20} + \hat{\gamma}_{21} \hat{x}_{2ig})}$. If $\hat{Z}_{2g} = 1$, then $\hat{Y}_{2ig} = 0$; otherwise, $\hat{Y}_{2ig} = \hat{X}_{2ig}$. Following Zappia et al. [47], we plot the mean-variance trend on the log-scale. Therefore, we normalized the observed data Y_{2ig} and simulated data \hat{Y}_{2ig} by counts per million reads, added one to the normalized data, and then took the logarithm to obtain the log-scale normalized expression levels Y_{2ig}^* and \hat{Y}_{2ig}^* . Finally, we calculated the mean μ_{gk} and variance σ_{gk} of the log-scale normalized expression levels within the cell type k (**Figure R3**), as well as the overall mean μ_g and variance σ_g , combining all cell types (**Figure R2a**) as follows:

$$\begin{aligned}\mu_{gk} &= \frac{1}{m_k} \sum_{i=1}^{n_2} I(w_{2i} = k) y_{2ig}^*; \\ \sigma_{gk} &= \sqrt{\frac{1}{m_k - 1} \sum_{i=1}^{n_2} I(w_{2i} = k) (y_{2ig}^* - \mu_{gk})^2}; \\ \mu_g &= \frac{1}{n_2} \sum_{i=1}^{n_2} y_{2ig}^*; \\ \sigma_g &= \sqrt{\frac{1}{n_2 - 1} \sum_{i=1}^{n_2} \sum_{k=1}^K I(w_{2i} = k) (y_{2ig}^* - \mu_{gk})^2},\end{aligned}$$

where $m_k = \sum_{i=1}^{n_2} I(w_{2i} = k)$ represents the number of cells of cell type k , $n_2 = \sum_{k=1}^K m_k$ denotes the total number of cells in the second batch and y_{2ig}^* can be the log-scale normalized expression levels of observed data Y_{2ig}^* or simulated data \hat{Y}_{2ig}^* to calculate its corresponding mean and variance. **Figures R2b and R4** are drawn in the same manner.

Figures R2-R4 show that although the real data are slightly more overdispersed than the synthetic data, the mean-variance trends of the synthetic data generated according to the estimated values of BUSseq fit the mean-variance trends of the real data quite well.

We the results on pages 14 and 18 in the revised manuscript and pages 56-57 of the Supplementary Material.

Figure R2: Scatter plot of the variance versus the mean of each gene for (a) the hematopoietic study and (b) the pancreas study. Red points are the observed values from real data; blue points correspond to the values of the data simulated according to the estimates of BUSseq for the real data. BUSseq recapitulates the mean-variance trends of the real data. The same figure is shown as **Figure 5** of the main manuscript.

Figure R3: Scatter plot of the variance versus the mean of each gene within cluster 1-6 (a-f), respectively, for the hematopoietic study. Red points are the observed values from real data; blue points correspond to the values of the data simulated according to the estimates of BUSseq for the real data. The same figure is shown as **Supplementary Figure 9**.

Figure R4: Scatter plot of the variance versus the mean of each gene within cluster 1-8, respectively, for the pancreas study. Red points are the observed values from real data; blue points correspond to the values of the data simulated according to the estimates of BUSseq for the real data. The same figure is shown as **Supplementary Figure 12**.

2. Page 8. The performance of BUSseq, including the ability to identify the correct value for K (via BIC), is assessed using synthetic data. It seems that the data was generated using the BUSseq model. If that is the case, the comparison with other methods is biased. I suggest the authors to use instead datasets generated via alternative models (such as [2]) and the control datasets introduced by [2], assessing the performance of BUSseq across different levels of sparsity (e.g. 10X vs smart-seq2 data) and sample complexity (how distinct are the clusters). These comparisons will enable users to better understand the strengths and limitations of BUSseq (e.g. choice of K can be more challenging for less distinct clusters).

We thank the reviewer for the suggestion and for pointing us to the datasets in [41]. Tian et al. [41] generated two sets of benchmarking experiments.

Different levels of sparsity

The first set of experiments assayed three lung adenocarcinoma cell lines—HCC827, H1975 and H2228—on three platforms with CELseq2, 10x Chromium and Drop-seq protocols, respectively. As a result, each batch consists of three cell types, and data from different batches have different levels of sparsity. We downloaded the raw count data from the GitHub repository “https://github.com/LuyiTian/sc_mixology.” We selected the top 6,000 highly variable genes (HVGs) within each batch using the *trendVar* and *decomposeVar* functions in the R package *scran* [33] and obtained 2,267 common HVGs across three batches.

The t-SNE and PCA plots of the raw count data (**Figure R5**) show that significant batch effects occurred across the three protocols. Thus, we applied BUSseq and varied the number of cell type K from 2 to 6. The Bayesian information criterion (BIC) obtains its minimum value at $K = 4$ (**Figure R6**). Although the BIC selects four cell types instead of three cell lines, two of the four identified clusters actually correspond to two subpopulations of the H1975 cell lines (**Table R1**). We further visualize the log-scale mean expression levels of intrinsic genes of the four learned cell types (**Figure R7a**). The first two cell types have similar expression patterns, but some differentially expressed genes are observed between them. Moreover, the PCA (**Figure R8c**) and t-SNE (**Figure R8a**) plots demonstrate the high level of similarity of the first two estimated cell types. Meanwhile, the PCA (**Figure R8d**) and tSNE (**Figure R8b**) plots confirm that the corrected count data \tilde{x}_{big} obtained by BUSseq cluster cells by cell type rather than by batch (**Figure R7b**). We also applied the benchmarked methods to evaluate their clustering accuracy. Once again, BUSseq outperforms all of the other methods (see **Table R2**).

We have added a new section “BUSseq is applicable to droplet-based scRNA-seq protocols” on pages 19-21 of the main manuscript.

Figure R5: Patterns of raw count data of the LUAD study. **(a)** t-SNE plot colored by cell line (H1975, H2228 and HCC827). **(b)** t-SNE plot colored by protocol (CELseq2, 10x Chromium, and Dropseq). **(c)** PCA plot colored by cell line (H1975, H2228 and HCC827). **(d)** PCA plot colored by protocol (CELseq2, 10x Chromium, and Dropseq). The two t-SNE plots are shown as **Figure 8** of the main manuscript, and the two PCA plots are shown as **Supplementary Figure 13**.

Figure R6: BIC curve of the first LUAD study. The same figure is shown as **Supplementary Figure 14**.

	HCC827	H1975	H2228
1	1	317	2
2	1	191	0
3	424	0	0
4	0	3	462

Table R1: The contingency table between the estimated cell types $\hat{w}_{bi} \in \{1, 2, 3, 4\}$ and the known cell-line labels. The same table is shown as **Supplementary Table 10**.

Figure R7: Patterns of intrinsic genes of the LUAD study. **(a)** Heatmap of log-scale mean expression levels $\alpha + \beta_k$ of cell types $k = 1, 2, 3, 4$. Each row represents an intrinsic gene. The colored bar indicates the cell type of each column. **(b)** Heatmap of the log-scale corrected count data by BUSseq. Each row again represents an intrinsic gene, but each column corresponds to a cell. The upper colored bar indicates the estimated cell type of each cell, and the lower colored bar denotes the batch to which the corresponding cell belongs. These two heatmaps are shown as **Figure 8** of the main manuscript.

Figure R8: Patterns of corrected count data of BUSseq of the LUAD study. (a) t-SNE plot colored by cell line (H1975, H2228 and HCC827). (b) t-SNE plot colored by protocol (CELseq2, 10x Chromium, and Dropseq). (c) PCA plot colored by cell line (H1975, H2228 and HCC827). (d) PCA plot colored by protocol (CELseq2, 10x Chromium, and Dropseq). The two t-SNE plots are shown as **Figure 8** of the main manuscript, and the two PCA plots are shown as **Supplementary Figure 13**.

Methods	BUSseq	LIGER	MNN	Scanorama	scVI	Seurat	ZINB-WaVE
ARI	0.8406	0.8250	0.6497	0.6368	0.4294	0.3236	0.3983

Table R2: The ARI of benchmarked methods for the first set of LUAD datasets.

Different levels of sample complexity

In the second set of experiments, Tian et al. [41] assayed four batches of pseudo-cells. Specifically, in each batch, single cells from the three cell lines were sorted into 384-well plates with 9 cells per

well (a small proportion of wells had only 3 cells to mimic small cells) in different combinations. RNAs from all of the 9 cells in the same well were then pooled and sub-sampled such that an approximately single-cell quantity of RNA was extracted from each well. Tian et al. [41] used a triad (a, b, c) to denote a combination of a HCC827 cells, b H1975 cells and c H2228 cells for each well, and generated 35 groups (34 9-cell mixtures and 1 small (three-cell) mixture) of pseudo cells with distinct combinations. Pseudo cells are much more similar to each other than cells from the three pure cell lines, so we can assess the performance of BUSseq in the experiment with a complex cell population.

To assess the performance of selecting the number of cell types by BIC across different levels of sample complexity, we first applied BUSseq to three pure cell line mixtures, following [41]. Specifically, we collected cells in the three pure cell line mixtures—denoted by the triads $(9,0,0)$, $(0,9,0)$ and $(0,0,9)$ —in the four batches. Again, we downloaded the raw count data from the GitHub repository “https://github.com/LuyiTian/sc_mixology.” We selected the top 6,000 HVGs in each of the four batches using the R package *scraper* [33] and used the common HVGs across the four batches for the downstream analysis. The t-SNE and PCA plots in **Figure R9** show that batch effects occur but are smaller than those induced by different protocols in the first set of experiments. We applied BUSseq to the resulting dataset and varied the number of cell types K from 2 to 6. BUSseq successfully learned the number of cell types as $K = 3$ by BIC (**Figure R11b**) and perfectly clusters these cells such that the adjusted random index (ARI) is one, as in Case 1 in **Table R4**. After batch effects correction, cells from the same cell line but assayed in different batches are clustered together (**Figure R12a-d**).

We then incorporated two intermediate cell line mixtures $(4,0,5)$ and $(5,0,4)$ into the analysis in addition to the pure cell line mixtures $(9,0,0)$, $(0,9,0)$ and $(0,0,9)$, to mimic an intermediate “development state” between HCC827 cells and H2228 cells. We provided PCA and t-SNE plots of the raw count data in **Figure R10**. We applied BUSseq to this dataset with five cell line mixtures. **Table R3** shows the distribution of each cell line mixture involved.

Figure R9: PCA and t-SNE plots of the raw count data of three pure cell line ((9,0,0),(0,9,0) and (0,0,9)). Each node represents a cell colored by cell line mixture (a,c) and by batch (b,d). The same figure is shown as **Supplementary Figure 16**.

Figure R10: PCA and t-SNE plots of the raw count data of three pure cell lines ((9,0,0),(0,9,0) and (0,0,9)) and two more intermediate cell line mixtures ((5,0,4) and (4,0,5)). Each node represents a cell colored by cell line mixture (a,c) and by batch (b,d). The same figure is shown as **Supplementary Figure 17**.

Cell line mixture	Batch 1	Batch 2	Batch 3	Batch 4
(0,0,9)	9	10	10	10
(0,9,0)	19	19	19	19
(4,0,5)	3	2	4	4
(5,0,4)	4	3	4	4
(9,0,0)	18	18	18	18

Table R3: The distribution of each cell line mixture in the four batches. The same table is shown in **Supplementary Table 12**

Figure R11: BIC curves of the second LUAD studies. **(a)** Case 1 with only three pure cell line mixtures. **(b)** Case 2 with five cell line mixtures. The same figure is shown as **Supplementary Figure 15**.

Figure R12: PCA and t-SNE plots of the corrected count data by BUSseq. **(a-d)** Three pure cell line mixtures. **(e-h)** Five cell line mixtures. Each node represents a cell colored by cell line mixture **(a,c,e,g)** and by batch **(b,d,f,h)**. The first four subfigures are added in **Supplementary Figure 16**, and the remaining four subfigures are added in **Supplementary Figure 17**.

However, BIC still selected the optional number of cell types as $K = 3$ for the dataset with five cell line mixtures. **Table R4** shows that the cells in the cell line mixture (5,0,4) or (4,0,5) are clustered into the two pure cell line mixtures (9,0,0) and (0,0,9). After batch effects correction, these intermediate cells are located between the two pure cell line clusters in **Figure R12e**, but they fail to generate a new cluster. The main reason for the failure of BIC to identify the existence of four or five clusters is the limited sample size—there are only 215 cells in total for the five cell line mixtures. To differentiate between more subtle cell type differences, statistical inference requires larger sample sizes. Nevertheless, **Table R4** and **Figure R12e** demonstrate that BUSseq is able to preserve the “development trajectory.”

We added discussion on page 22 of the main manuscript and present the details in Supplementary Notes.

Cell line mixture	Case 1			Case 2		
	1	2	3	1	2	3
(9,0,0)	0	0	72	0	0	72
(0,9,0)	76	0	0	76	0	0
(0,0,9)	0	39	0	0	39	0
(5,0,4)	-	-	-	0	11	4
(4,0,5)	-	-	-	0	13	0

Table R4: The contingency table between the estimated cell types $\hat{w}_{bi} \in \{1, 2, 3, 4\}$ and the known cell-line labels.

3. Page 11, analysis of hematopoietic dataset. MNN (and others) do not perform clustering. How were the cluster labels assigned to calculate the ARI with respect to the true labels? If different clustering methods are used across the methods, this could affect the comparison. Instead, the authors might consider applying the same clustering method using the “corrected” data returned by each method. Same comment applies to the pancreatic dataset.

We thank the reviewer for the comments.

Each benchmarked method has its own clustering approach, as outlined in its respective publication [16, 20, 30, 36, 40, 45]. In our previous analysis, we respected the original clustering algorithm and analysis protocol of each benchmarked method. In the following, we first briefly describe the clustering approach used by each method and summarize the approaches in **Table R5**.

Linked Inference of Genomic Experimental Relationships (LIGER) [45] first identifies shared and batch-specific factors through integrative non-negative matrix factorization (iNMF). Next, it constructs a k-nearest-neighbor graph separately for each batch using the factor loadings of each cell. LIGER further connects two cells i and j from different batches if cell i and cell j 's neighbors in their corresponding batches have similar cluster membership distributions, thus leading to a shared factor neighborhood (SFN) graph. Finally, LIGER performs Louvain community detection on the SFN graph to identify cell clusters with the *quantileAlignSNF* function in the R package *liger*.

The mutual nearest neighbors (MNN) method [16] constructs a shared nearest neighbor (SNN) graph of the corrected count data and applies the “Walktrap” algorithm to the SNN graph.

Scanorama [20] first applies k-means clustering for the corrected gene expression space to obtain 40 clusters and then assigns cell types to each cluster using knowledge of the previously provided cell-type labels. However, we do not use the cell-type label information in the clustering of other benchmarked methods, so we only assume that we know the true number of cell types K and performed the robust k-means clustering with the *pam* function in the R package *cluster*.

scVI [30] applies k-means clustering to their low-dimensional embedding of cells.

Seurat [40] first applies PCA to the dataset and then constructs an SNN graph using the first 30 principal components with the *FindNeighbors* function in the R package *Seurat*. Finally, Seurat determines the clusters by applying the *FindClusters* function in the R package *Seurat* to the SNN graph (see *Assignment of cell type labels for pancreatic islet cells* paragraph in the START★METHOD of [40]).

ZINB-WaVE [36] finds subpopulations of cells using Seurat’s clustering algorithm, following the vignettes of the R package *zinbwave* (<https://bioconductor.org/packages/release/bioc/vignettes/zinbwave/inst/doc/intro.html#the>

zinb-wave-model).

Method	Output	Clustering	ARI by their own strategies		ARI by k-means clustering	
			hematopoietic	Pancreas	hematopoietic	Pancreas
BUSseq	The logarithm of the corrected count data of the identified intrinsic genes	The cluster indicators of mixture model	0.5822	0.6080	0.4757	0.880
LIGER	The cell-specific factor loadings	SFN graph + Louvain community detection	0.3066	0.5421	0.3188	0.8331
MNN	The logarithm of the corrected count data	SNN graph + Walk-trap algorithm	0.5754	0.2793	0.4522	0.4220
Scanorama	The batch-corrected data	k-means	0.5184	0.5272	0.5184	0.5272
scVI	The low-dimensional embedding of cells	k-means	0.1969	0.2819	0.4444	0.5202
Seurat	The first 30 principal components of the corrected data	SNN graph + modularity optimization	0.2663	0.2868	0.3413	0.1828
ZINB-WaVE	The low-dimensional embedding of cells	SNN graph + modularity optimization	0.3484	0.3804	0.4892	0.6404

Table R5: The outputs and original clustering strategies of all of the benchmarked methods. We list the ARIs of each method with reference to its original clustering method and the unified k-means clustering, with the number of clusters set to 7 as provided by the FACS labels, respectively. The same table is shown as **Supplementary Table 7**

Nevertheless, we completely understand Reviewer 1’s point. Following Reviewer 1’s suggestion, we applied robust k-means clustering using the *pam* function in the R package *cluster* to the corrected count data; if not available, we used the low-dimensional embeddings offered by each method. The mouse hematopoietic study involves 7 cell types labeled by FACS, including long-term hematopoietic stem cells, multi-potent progenitors, common myeloid progenitors, megakaryocyte-erythrocyte progenitor, granulocyte-monocyte progenitors, lymphoid-primed multipotent progenitors and other cells. In the human pancreas study, there are 7 cell types in the FACS labeling, including alpha cells, beta cells, delta cells, gamma cells, acinar cells, ductal cells and other cells. Thus, we set the numbers of clusters to 7 for both real datasets when applying k-means clustering. The resulting ARIs are presented in **Table R5**. We can see that BUSseq still ranks the second for the hematopoietic dataset and the first for the pancreas dataset.

We have explained the details in Supplementary Notes.

4. Figure 4. tSNE and PCA plots for the uncorrected data do not match those reported by Haghverdi et al (Figure 3). Whilst this might be expected for tSNE (which results depend on a random seed), this is not the case for PCA. Please explain why do these differences arise. Similar comment for the pancreatic data.

We thank the reviewer for the comments.

Let us first explain the case for the hematopoietic study.

To begin with, when running the source code of Haematopoiesis section of the GitHub repository MMN2017, we generated t-SNE and PCA plots that were different from those in Haghverdi et al. [16].

There were two batches of gene expression count data, one generated on the SMART-seq2 platform by [34] with accession number GSE81682 and the other generated on the MARS-seq platform by [35] with accession number GSE72857. The authors first labeled the cells using FACS and then performed size factor normalization within each batch. Next, the authors filtered out the common HVGs identified by [34] between the two datasets. These HVGs were denoted by their Ensembl ID. The genes in the GSE81682 dataset were named by Ensembl ID, but the genes in the GSE72857 dataset were named by Gene Symbol. The authors used the R package *biomaRt* to query the corresponding Gene Symbol by Ensembl ID.

However, a gene may have several synonyms as its Gene Symbol, and the corresponding Gene Symbol of an Ensembl ID can change with the version of the R package *biomaRt*. For example, the gene with Ensembl ID “ENSMUSG00000043153” corresponds to the Gene Symbol *Crppa* for the current *biomaRt* version 2.38.0 in January 2020, but to the Gene Symbol *Ispd* for January 2019. Thus, the queried Gene Symbol names may fail to match the Gene Symbol names of the GSE72857 dataset. We finally obtained 3,457 HVGs in common between two datasets, instead of 3,937 genes as noted in their manuscript [16]. It should also be noted that when we conducted the preprocessing to prepare the results in our manuscript, the version of the *biomaRt* package was also different from the current version 2.38.0. Thus, there were 3,470 common HVGs between the two datasets in our manuscript. When we drew the t-SNE and PCA plots by running their source code (**Figures R14a and R14b**), the PCA plots from the original paper and the reproduction were different. In addition to the PCA plots, even though a seed was set to ensure the reproductivity of the t-SNE plots, we still failed to obtain the same t-SNE plots as well. This difference may result from the use of different sets of HVGs.

Moreover, to alleviate the impact of sequencing depths, Haghverdi et al. [16] used m_g^{MARS} and m_g^{SMART} to denote the mean expression levels of gene g across all cells in the MARS-seq and SMART-seq2 batches, respectively, and scaled the MARS-seq data to match the coverage of the SMART-seq2 batch using the median of the mean expression ratio $\rho = Median(\frac{m_g^{SMART}}{m_g^{MARS}})$

across all genes: $y_{ig}^{scaled} = \rho \cdot y_{ig}$, where y_{ig} denotes the normalized expression levels of gene g in the i th cell of the MARS-seq batch. Finally, Paul et al. [35] used the logarithm of the normalized expression levels plus one as the uncorrected data to draw the t-SNE and PCA plots (**Figure R13**). Moreover, in the PCA plot, they only included **the three common cell types between two batches**—common myeloid progenitors (CMP), megakaryocyte-erythrocyte progenitors (MEP) and granulocyte-monocyte progenitors (GMP).

In contrast, because different correction methods apply different normalization strategies and some methods (BUSseq and ZINB-WaVE) do not require prior normalization, we created PCA and t-SNE plots using the raw count data without normalization. We only selected the 3,470 common HVGs between two datasets identified by [34] to obtain the raw read count data. Thus, compared with **Figures R14a and R14b**, **Figures R15a and R15b** show that the two batches are separated from each other. Moreover, we provide the PCA plots (**Figure R15c and R15d**) in our manuscript **with all of the cell types** instead of the **only three shared cell types** (**Figure R15b**). We have added a discussion on page 35 of the main manuscript and detailed out the whole workflow in the Supplementary Notes.

In case of interest, in **Figure R16**, we draw the t-SNE and PCA plots of the uncorrected count data with 3,470 common HVGs after applying size factor normalization and gene coverage scaling between the two datasets.

Reprinted by permission from Springer Nature, *Nature Biotechnology*, 36, 421–427 (2018). Haghverdi, L., Lun, A., Morgan, M. et al. Batch effects in single-cell RNA-sequencing data are corrected by matching mutual nearest neighbors. <https://doi.org/10.1038/nbt.4091>

Figure R13: Original t-SNE and PCA plots for the hematopoietic study obtained from Figure 3 of the paper [16].

Figure R14: t-SNE (a) and PCA (b) plots of the uncorrected count data for the hematopoietic study by rerunning the codes on their GitHub repository.

Figure R15: t-SNE (a) and PCA (b) plots of the uncorrected count data for the hematopoietic study after the modified preprocessing for cells from CMP, MEP and GMP. (c) The PCA plot colored by FACS labeling for all of the cells, i.e. cells from all of the cell types. (d) The PCA plot of the uncorrected count data is shown in our manuscript with different colors.

Figure R16: t-SNE (a,d) and PCA (b,c,e,f) plots of the count data after normalization but without batch-effect-correction for the hematopoietic study. Panels (b,c) only contain the three shared cell types between the two datasets, similar to **Figure R13f**, whereas panels (e,f) include all cell types. Panels (a-c) use the MNN coloring pattern, whereas panels (d-f) use the coloring in our manuscript. Panels (d) and (f) are shown as **Supplementary Figure 8**.

We followed the Pancreas section of the GitHub repository MMN2017 to reproduce the t-SNE and PCA plots in their paper, but we had to make some changes to label the cell types, which are explained below.

The authors first removed poor-quality cells and genes and then normalized the raw count data using the deconvolution methods proposed by Lun et al. [32]. Next, the authors identified HVGs within each batch following [4]. Two batches (GSE86473 and EMATB-5061) had the cell-type labels for all of the cells, but the cell type labels of the other two datasets profiled by the CEL-seq2 platform (GSE81076 and GSE85241) were inferred by the marker genes used in the original publications by Lawlor et al. [27] and Grün et al. [15]. **Figure R17** shows the original tSNE and PCA plots.

To assign cell type labels for the GSE81076 and GSE85241 datasets, the authors first extracted

the normalized expression levels of the selected HVGs within each dataset, respectively. Next, the authors obtained the low dimensional embedding of HVGs by tSNE for visualization (**Figure R18**). At the same time, the authors applied robust k-means clustering to the normalized expression levels of the selected HVGs using the *pam* function in the R package *cluster*. The number of clusters was set as 9. Next, the authors drew t-SNE plots colored by the expression levels of the marker genes. *GCG* is highly expressed in alpha islet cells, *INS* in beta islet cells, *SST* in delta islet cells, *PPY* in gamma islet cells (pancreatic polypeptide (PP) cells), *PRSS1* in acinar cells, *KRT19* in ductal cells and *COL1A1* in mesenchymal cells [15, 27]. Note that the authors allocate mesenchymal cells to the “other” cell type category in the following analysis. Meanwhile, the shape of each point in the t-SNE plot was determined by its corresponding k-means cluster. For example, if the cells in cluster 7 strongly express the *INS* gene, they were assigned to the category of beta islet cells (**Figure R18**). However, we found that the code posted in the GitHub repository MMN2017 incorrectly assigned cells in cluster 6 to beta islet cells as **Figure R19a**. Therefore, we modified the codes to assign cell type labels as in **Figure R19b**.

We now demonstrate the correctness of our modification from three viewpoints, using violin plots of the expression levels of marker genes within each cell type, the ARI compared with the clustering inferred by the MNN correction method, and a comparison with the t-SNE plot shown in their paper:

1. The violin plots of the expression levels of the marker genes after cell type label assignment **Figures R20 and R21** demonstrate that the marker genes are highly expressed in the corresponding cell types, so the modified codes correctly assign cell type labels. Note that “PP” in their code (**Figure R19a and R19c**) and “Gamma” in our code (**Figure R19b and R19d**) both represent gamma islet cells.
2. In addition, the ARI between the incorrect cell type labeling and the cell type labels inferred by the MNN correction method is 0.2045, while the ARI between the modified labeling and the inferred cell type labels by MNN is 0.2793.
3. According to the t-SNE plot colored by cell type labels, alpha islet cells (blue points) are mixed with delta islet cells (orange points) (**Figure R22a**). After modification, there is an entire region of alpha islet cells (**Figure R22a**). This coincides with the original t-SNE plot in their paper [16] (**Figure R17**).

Therefore, we used the corrected cell-type labeling in our manuscript. We have added explanation to page 36 of the main manuscript and provide the whole workflow in the Supplementary Notes.

Similar to the preprocessing pipeline in the hematopoietic study, we drew PCA and tSNE plots of the raw count data without normalization. In **Figure R24**, we show t-SNE and PCA plots of the normalized uncorrected count data.

Figure R17: Original PCA and t-SNE plots for the pancreas study obtained from Figure 4 and Supplementary Figure 5 of the paper [16]. 26

Figure R18: t-SNE plot of the normalized expression levels of HVGs in the GSE81076 colored by *INS* levels and shaped by k-means clustering.

```

a 116 table(marker_uber$Kmediods, marker_uber$markClust)
117 marker_uber$CellType <- ""
118 marker_uber$CellType[marker_uber$Kmediods %in% c(7)] <- "Alpha"
119 marker_uber$CellType[marker_uber$Kmediods %in% c(7) & marker_uber$PPY >= 4] <- "PP"
120 marker_uber$CellType[marker_uber$Kmediods == 6] <- "Beta"
121 marker_uber$CellType[marker_uber$Kmediods == 8] <- "Delta"
122 marker_uber$CellType[marker_uber$Kmediods %in% c(1, 2)] <- "Acinar"
123 marker_uber$CellType[marker_uber$Kmediods %in% c(3, 4, 5)] <- "Ductal"
124 marker_uber$CellType[marker_uber$Kmediods %in% c(9) & marker_uber$COL1A1 >= 2] <- "Mesenchyme"
125 table(marker_uber$CellType, marker_uber$Kmediods)
...

b 688 marker_uber$CellType <- ""
689 marker_uber$CellType[marker_uber$Kmediods %in% c(3, 4, 5)] <- "Ductal"
690 marker_uber$CellType[marker_uber$Kmediods %in% c(8) | (marker_uber$Kmediods %in% c(5) & marker_uber$GCG >= 7)] <- "Alpha"
691 marker_uber$CellType[marker_uber$Kmediods %in% c(8) & marker_uber$PPY >= 5] <- "Gamma"
692 marker_uber$CellType[marker_uber$Kmediods %in% c(7) | (marker_uber$Kmediods %in% c(5) & marker_uber$INS >= 7)] <- "Beta"
693 marker_uber$CellType[marker_uber$Kmediods %in% c(5, 7) & marker_uber$SST >= 8] <- "Delta"
694 marker_uber$CellType[marker_uber$Kmediods %in% c(1, 2, 6)] <- "Acinar"
695 marker_uber$CellType[marker_uber$Kmediods %in% c(9) & marker_uber$COL1A1 >= 2] <- "Mesenchyme"
696 table(marker_uber$CellType, marker_uber$Kmediods)
...

c 241 table(marker_uber$Kmediods, marker_uber$markClust)
242 marker_uber$CellType <- ""
243 marker_uber$CellType[marker_uber$Kmediods %in% c(8)] <- "PP"
244 marker_uber$CellType[marker_uber$Kmediods %in% c(4)] <- "Beta"
245 marker_uber$CellType[marker_uber$Kmediods %in% c(1, 7)] <- "Alpha"
246 marker_uber$CellType[marker_uber$Kmediods %in% c(3)] <- "Delta"
247 marker_uber$CellType[marker_uber$Kmediods %in% c(5, 9)] <- "Acinar"
248 marker_uber$CellType[marker_uber$Kmediods %in% c(6) ] <- "Ductal"
249 marker_uber$CellType[marker_uber$Kmediods %in% c(2)] <- "Mesenchyme"
250 table(marker_uber$CellType)
...

d 882 table(marker_uber$Kmediods, marker_uber$markClust)
883 marker_uber$CellType <- ""
884 marker_uber$CellType[marker_uber$Kmediods %in% c(3)] <- "Gamma"
885 marker_uber$CellType[marker_uber$Kmediods %in% c(9) | (marker_uber$Kmediods %in% c(3) & marker_merge$INS >= 8) ] <- "Beta"
886 marker_uber$CellType[marker_uber$Kmediods %in% c(1, 4) | (marker_uber$Kmediods %in% c(3) & marker_merge$GCG >= 8)] <- "Alpha"
887 marker_uber$CellType[marker_uber$Kmediods %in% c(3, 4, 9) & marker_merge$SST >= 9] <- "Delta"
888 marker_uber$CellType[marker_uber$Kmediods %in% c(5, 8)] <- "Acinar"
889 marker_uber$CellType[marker_uber$Kmediods %in% c(6,7) ] <- "Ductal"
890 marker_uber$CellType[marker_uber$Kmediods %in% c(2)] <- "Mesenchyme"
...

```

Figure R19: Snapshots of the code to assign cell type labels. (a) MNN code for GSE81076. (b) Our modified code for GSE81076. (c) MNN code for GSE85241. (d) Our modified code for GSE85241.

Figure R20: Violin plots of marker-gene expression levels—GCG, INS, SST, PPY, PRSS1, KRT19 and COL1A1—in terms of the original labeling from the MNN GitHub (left panel) and our modified labeling (right panel) for the GSE81076 dataset.

Figure R21: Violin plots of expression levels of marker-genes—GCG, INS, SST, PPY, PRSS1, KRT19 and COL1A1—in terms of the original labeling from MNN Github (left panel) and our modified labeling (right panel) for the GSE85241 dataset.

Figure R22: PCA and t-SNE plots of the uncorrected count data colored by two kinds of cell type labeling for the pancreas study: the original labeling from MNN Github (left panel, **(a,c)**) and the modified labeling (right-hand panel, **(b,d)**).

Figure R23: t-SNE (a,b,d,e) and PCA (c,f) plots of the raw count data without normalization for the pancreas study. Panels (a-c) follow the coloring pattern in the MNN paper, whereas panels (d-f) follow that of our manuscript.

Figure R24: t-SNE (a,d) and PCA (b,c,e,f) plots of the count data after normalization but without batch-effect-correction for the pancreas study. Panels (a-c) use the MNN coloring pattern, while panels (d-f) use the corresponding coloring pattern in our manuscript. These two subfigures (d) and (f) are added in **Supplementary Figure 8**.

5. MCMC implementation. (i) how were the prior hyper-parameters chosen and how does this affect posterior inference?, (ii) please provide diagnostic criteria to assess the convergence of the MCMC for all data analyses, (iii) how does the method scale in terms of the number of genes or cells? (iv) please describe what steps are parallelised in the GPU implementation.

We thank the reviewer for these very helpful suggestions. In the following, we address the reviewer’s comments one by one.

(i) Following the reviewer’s suggestion, we have conducted extensive sensitivity analyses with respect to the hyper-parameters. For the analyses in our paper, we assign independent priors to all of the parameters with hyper-parameters specified as shown in **Table R6**.

Parameter	Definition	Prior distribution	Hyper-parameters
$\boldsymbol{\pi}_b$	The cell type proportions in batch b	$\boldsymbol{\pi}_b \sim \text{Dirichlet}(\xi, \dots, \xi)$	$\xi = 2$
γ_{b0}	The odds ratio for the dropout events in the batch b	$\gamma_{b0} \sim N(0, \sigma_{\gamma_0}^2)$	$\sigma_{\gamma_0}^2 = 3$
γ_{b1}	The slope of the logistic regression for the dropout events in the batch b	$-\gamma_{b1} \sim \text{Gamma}(a_\gamma, b_\gamma)$	$a_\gamma = 0.001$ and $b_\gamma = 0.01$
α_g	The log-scale baseline expression level of gene g	$\alpha_g \sim N(m_g^a, \sigma_a^2)$	m_g^a is the empirical estimate of α_g and $\sigma_a^2 = 5$
β_{gk}	The cell type effects of gene g in the cell type k	$\beta_{gk} L_{gk} \sim N(0, \tau_{\beta, L_{gk}}^2)$	$\tau_{\beta, 1}^2 = 50$
p	The proportion of appearing differentially expressed genes	$p \sim \text{Beta}(a_p, b_p)$	$a_p = 1$ and $b_p = 3$
$\tau_{\beta, 0}^2$	The variance of the slab prior for the cell type effects	$\tau_{\beta, 0}^2 \sim \text{Inv-Gamma}(a_\tau, b_\tau)$	$a_\tau = 2$ and $b_\tau = 0.01$
ν_{bg}	The location batch effects of gene g in the batch b	$\nu_{bg} \sim N(m_{bg}^c, \sigma_c^2)$	m_{bg}^c is the empirical estimate of ν_{bg} and $\sigma_c^2 = 5$
δ_{bi}	The size factor of the cell i in the batch b	$\delta_{bi} \sim N(m_{bi}^d, \sigma_d^2)$	m_{bi}^d is the empirical estimate of δ_{bi} and $\sigma_d^2 = 5$
ϕ_{bg}	The overdispersion parameter of gene g in the batch b	$\phi_{bg} \sim \text{Gamma}(a_\phi, b_\phi)$	$a_\phi = 1$ and $b_\phi = 0.1$

Table R6: The prior distribution for all parameters. The same table is shown as **Supplementary Table 5**.

Here, we detail out how we obtain the empirical estimates for m_g^a s, m_{bg}^c s and m_{bi}^d s. We first give a crude estimate of the cell size factor δ_{bi} by taking the logarithm of the ratio of the total read

counts of cell i over the total read counts of the first cell in batch b :

$$m_{bi}^d = \log\left(\frac{\sum_{g=1}^G y_{big}}{\sum_{g=1}^G y_{b1g}}\right). \quad (\text{R.1})$$

Then, we randomly assign labels $w_{bi}^{(0)}$ to all cells. Because we assume $\beta_{g1} = 0$ and $\nu_{1g} = 0$, the cells in the first batch and the first cell type have the mean expression level $\exp(\alpha_g + \delta_{1i})$, so we use m_{1i}^d to give a crude estimate for δ_{1i} . Thus, we estimate the log-scale baseline expression levels α_g as following:

$$m_g^a = \frac{\sum_{i=1}^{n_1} I(w_{1i}^{(0)} = 1) \log\left(1 + \frac{Y_{1ig}}{\exp(m_{1i}^d)}\right)}{\sum_{i=1}^{n_1} I(w_{1i}^{(0)} = 1) + 1}, \quad (\text{R.2})$$

where we add one to the denominator to ensure that the denominator is positive. Furthermore, we compare the expression levels of the cells assigned to the first cell type across different batches to estimate the batch effects.

$$m_{bg}^c = \frac{\sum_{i=1}^{n_b} I(w_{bi}^{(0)} = 1) \log\left(1 + \frac{Y_{big}}{\exp(m_{bi}^d)}\right)}{\sum_{i=1}^{n_b} I(w_{bi}^{(0)} = 1) + 1} - \frac{\sum_{i=1}^{n_1} I(w_{1i}^{(0)} = 1) \log\left(1 + \frac{Y_{1ig}}{\exp(m_{1i}^d)}\right)}{\sum_{i=1}^{n_1} I(w_{1i}^{(0)} = 1) + 1}. \quad (\text{R.3})$$

We performed extensive sensitivity analysis to investigate how hyper-parameters affect the posterior inference in the simulation study. For each hyperparameter or each pair of hyperparameters in **Table R6**, we varied its value(s) to four different levels while fixing the other hyperparameters. Next, we applied BUSseq to each setting and calculated the corresponding ARI to evaluate the clustering accuracy (**Table R7**). If the ARI is equal to one, BUSseq clusters all of the cells perfectly.

Although we varied the hyperparameters with a fold change at the scale of dozens or even as large as 100 (**Table R7**), all of the ARIs are equal to one after varying the hyperparameter values, except when $\sigma_c^2 = 1$, whose corresponding ARI is 0.9971. Therefore, BUSseq is robust to the specification of hyperparameters.

We present the results on page 12 of the main manuscript and pages 47 and 57 of Supplementary Notes.

$\boldsymbol{\pi}_b \sim \text{Dirichlet}(\xi, \xi, \dots, \xi)$						$p \sim \text{Beta}(a_p, b_p)$					
ξ	0.1	0.5	2	5	10	(a_p, b_p)	(0.1,0.3)	(10,30)	(1,3)	(3,1)	(30, 10)
ARI	1.000	1.000	1.000	1.000	1.000	ARI	1.000	1.000	1.000	1.000	1.000
$\gamma_{b0} \sim N(0, \sigma_{\gamma_0}^2)$						$\tau_{\beta,0}^2 \sim \text{Inv-Gamma}(a_\tau, b_\tau)$					
$\sigma_{\gamma_0}^2$	0.3	1.5	3	6	30	ξ	(3,0.02)	(5,0.04)	(2,0.01)	(2,0.1)	(2, 1)
ARI	1.000	1.000	1.000	1.000	1.000	ARI	1.000	1.000	1.000	1.000	1.000
$-\gamma_{b1} \sim \text{Gamma}(a_\gamma, b_\gamma)$						$\nu_{bg} \sim N(m_{bg}^c, \sigma_c^2)$					
(a_γ, b_γ)	(0.1,0.3)	(1,10)	(0.001,0.01)	(0.01,0.001)	(1,0.1)	σ_c^2	1	2	5	10	100
ARI	1.000	1.000	1.000	1.000	1.000	ARI	0.997	1.000	1.000	1.000	1.000
$\alpha_g \sim N(m_g^a, \sigma_a^2)$						$\delta_{bi} \sim N(m_{bi}^d, \sigma_d^2)$					
σ_a^2	1	2	5	10	100	σ_d^2	1	2	5	10	100
ARI	1.000	1.000	1.000	1.000	1.000	ARI	1.000	1.000	1.000	1.000	1.000
$\beta_{gk} L_{gk} \sim N(0, \tau_{\beta, L_{gk}}^2)$						$\phi_{bg} \sim \text{Gamma}(a_\phi, b_\phi)$					
$\tau_{\beta,1}^2$	5	20	50	100	150	(a_ϕ, b_ϕ)	(10,1)	(0.1,0.01)	(1,0.1)	(5,0.1)	(0.1,1)
ARI	1.000	1.000	1.000	1.000	1.000	ARI	1.000	1.000	1.000	1.000	1.000

Table R7: Setting of hyperparameter values and their corresponding ARIs as inferred by BUSseq. The third value for each hyperparameter is the setting we apply in our inference for simulation and two real applications. The same table is shown as **Supplementary Table 6**.

(ii) To rigorously assess the convergence of the Markov chain Monte Carlo (MCMC) for data analysis, we adopt the estimated potential scale reduction (EPSR) criterion [13]. The detailed procedure of using EPSR to determine whether the Markov chain has reached convergence Gelman et al. [13] is as follows.

We start $\frac{m}{2}$ chains (assign $\frac{m}{2}$) with different initial values and run each chain for $4n$ iterations, where m is even ($m \geq 4$) and n is an integer. For each chain, we keep only the second half of the iterations, ignoring the first half as burn-ins. This results in $\frac{m}{2}$ chains with $2n$ iterations each. Next, we further cut each chain into two chains of equal length, obtaining $m = \frac{m}{2} \times 2$ chains, with $n = \frac{2n}{2}$ iterations each. Let θ denote a parameter of interest; we then use $\theta_{ij}, i = 1, 2, \dots, n, j = 1, 2, \dots, m$ to indicate the i^{th} sample of θ collected in chain j . With these notations, we define

$$\bar{\theta}_{\cdot j} = \frac{1}{n} \sum_{i=1}^n \theta_{ij}, \quad \bar{\theta}_{\cdot\cdot} = \frac{1}{m} \sum_{i=1}^n \bar{\theta}_{\cdot j}, \quad s_j^2 = \frac{1}{n-1} \sum_{i=1}^n (\theta_{ij} - \bar{\theta}_{\cdot j})^2,$$

$$B = \frac{n}{m-1} \sum_{j=1}^m (\bar{\theta}_{\cdot j} - \bar{\theta}_{\cdot\cdot})^2, \quad W = \frac{1}{m} \sum_{j=1}^m s_j^2.$$

B describes the between-chain variance, and W measures the within-chain variances. When multiple chains become stationary and mix well with each other, B should be very close to W . Subsequently, the EPSR factor for assessing the convergence of θ , which is defined as $\hat{R} = \sqrt{\frac{n-1}{n} + \frac{1}{n} \frac{B}{W}}$, should also be close to one.

We are interested in the log-scale baseline expression level $\{\alpha_g, g = 1, 2, \dots, G\}$, the cell type

effects $\{\beta_{gk}, g = 1, 2, \dots, G, k = 2, 3, \dots, K\}$, the location batch effects $\{\nu_{bg}, g = 1, 2, \dots, G, b = 2, 3, \dots, B\}$ and the overdispersion parameters $\{\phi_{bg}, g = 1, 2, \dots, G, b = 1, 2, \dots, B\}$. To avoid the impact of label switching of cell types, we consider the log-scale cell type specific expression level $\theta_{gk} = \alpha_g + \beta_{gk}, g = 1, 2, \dots, G, k = 1, 2, \dots, K$ and match the cell type indicators in different chains such that most cells in the different chains are assigned to the same cell types. If the EPSR factors of most parameters are close to one, we treat the posterior sampling as attaining stationary. Thus, we use the following rule to diagnose the convergence of the MCMC algorithm for BUSseq:

1. More than 80% of $\{EPSR(\theta_{gk})\}$ are less than 1.3
2. More than 80% of $\{EPSR(\nu_{bg})\}$ are less than 1.3
3. More than 80% of $\{EPSR(\phi_{bg})\}$ are less than 1.3

We calculated the EPSR factors in the simulation study and the two real applications. In the simulation, we initiated $\frac{m}{2} = 2$ chains. After running 4,000 iterations and taking the first 2,000 as burn-ins, 98.29% of $\{EPSR(\theta_{gk})\}$ are less than 1.3, 98.19% of $\{EPSR(\nu_{bg})\}$ are less than 1.3, and 99.09% of $\{EPSR(\phi_{bg})\}$ are less than 1.3. Therefore, the MCMC algorithm has converged after 2,000 burn-in iterations. In the mouse hematopoietic study, we also initiated two chains. After running 8,000 iterations and taking the first 4,000 as burn-ins, 89.09% of $\{EPSR(\theta_{gk})\}$ are less than 1.3, 88.73% of $\{EPSR(\nu_{bg})\}$ are less than 1.3, and 97.94% of $\{EPSR(\phi_{bg})\}$ are less than 1.3. In the pancreas study, we also initiated two chains. After running 8,000 iterations and taking the first 4,000 as burn-ins, 84.08% of $\{EPSR(\theta_{gk})\}$ are less than 1.3, 86.03% of $\{EPSR(\nu_{bg})\}$ are less than 1.3, and 97.96% of $\{EPSR(\phi_{bg})\}$ are less than 1.3.

We now discuss EPSR to pages 8, 14, 18, 33 of the main manuscript and explain the details in Supplementary Notes.

(iii) We ran BUSseq on simulation datasets with different numbers of genes or cells to evaluate the scalability of BUSseq. Theoretically, the time complexity of the MCMC algorithm is $O(NGK)$ in terms of the number of genes G and the total number of cells $N = \sum_{b=1}^B n_b$. We first fixed the number of genes $G = 3000$ and varied the number of cells N . **Figure R25a** and **Figure R25c** show that the running time is linear to the number of cells N for both the CPU parallel version and the GPU parallel version of BUSseq. Next, we fixed the number of cells $N = 1000$ and varied the number of genes G . **Figure R25b** and **Figure R25d** demonstrate that the running time is also linear to the number of genes G for both parallel versions of BUSseq. Thus, the simulation results are consistent with theory, regardless of whether the CPU or the GPU parallel version is used.

We have added discussions on page 24 of the main manuscript and explain the details in Supplementary Notes.

Figure R25: Running time and RAM usage for **(a,b)** the CPU parallel version of BUSseq on 8 cores of Dual Intel Xeon E5-2650 v2 2.60GHz processors and **(c,d)** the GPU parallel version of BUSseq on a single core of an Intel Xeon Gold 6132 processor and one NVIDIA Tesla P100 GPU. **(a)** When $G = 3,000$ and $n_s = 1,000$, N varies from 1,000 to 20,000. **(b)** When $N = 1,000$ and $n_s = 1,000$, G varies from 1,500 to 9,000. **(c)** When $G = 3,000$, N varies from 1,000 to 50,000. **(d)** When $N = 1,000$, G varies from 1,500 to 9,000. The same figure is shown in **Figure 9** of the main manuscript.

(iv) In the GPU version, for every iteration, parallelization is applied when:

- Proposing new values for a set of parameters or latent variables that are conditionally independent.
- Filling an array which carries the elements in the sum part of the log likelihoods/ acceptance rates.
- Summation of the individual elements in the calculation for the log likelihoods/ acceptance rates. Here, we adopt the reduction technique for GPU computing. Specifically, suppose that we want to sum over 1024 elements, in the first cycle, every two elements of the 1024 elements are summed to a total of 512 values in parallel, then the 512 numbers are further added up and reduced to 256 values in parallel in the second cycle and so on so forth. Finally, after 10 cycles, we obtain the summation over all the 1024 elements.

- Determine whether the newly proposed values are accepted and update them accordingly.
- For clearer illustration, I will explain the workflow for every parameter or latent variable involved in our algorithm.

For α_g :

- Firstly, an array with the length of G (no. of genes) is updated in parallel to store G α_g^* s, the proposed α_g s based on $\alpha_g^{[t-1]}$ s from the last iteration.
- Then, for $g = 1, \dots, G$,
 - An array of the length $\sum_{b=1}^B n_b = N$ (no. of samples) is filled in parallel with the N elements of the summation part in the log acceptance rate. In α_g , the log acceptance rate is

$$\sum_{b=1}^B \sum_{i=1}^{n_b} \left\{ (\alpha_g^* - \alpha_g^{[t-1]}) x_{big}^{[t]} + (\phi_{bg}^{[t-1]} + x_{big}^{[t]}) \log \left[\frac{\phi_{bg}^{[t-1]} + \exp(\alpha_g^{[t-1]} + \beta_{gw_{bi}^{[t-1]}} + \nu_{bg}^{[t-1]} + \delta_{bi}^{[t-1]})}{\phi_{bg}^{[t-1]} + \exp(\alpha_g^* + \beta_{gw_{bi}^{[t-1]}} + \nu_{bg}^{[t-1]} + \delta_{bi}^{[t-1]})} \right] \right\} - \frac{(\alpha_g^*)^2 - (\alpha_g^{[t-1]})^2}{2\sigma_a^2},$$

and the sum part is

$$\sum_{b=1}^B \sum_{i=1}^{n_b} \left\{ (\alpha_g^* - \alpha_g^{[t-1]}) x_{big}^{[t]} + (\phi_{bg}^{[t-1]} + x_{big}^{[t]}) \log \left[\frac{\phi_{bg}^{[t-1]} + \exp(\alpha_g^{[t-1]} + \beta_{gw_{bi}^{[t-1]}} + \nu_{bg}^{[t-1]} + \delta_{bi}^{[t-1]})}{\phi_{bg}^{[t-1]} + \exp(\alpha_g^* + \beta_{gw_{bi}^{[t-1]}} + \nu_{bg}^{[t-1]} + \delta_{bi}^{[t-1]})} \right] \right\}.$$

- The N elements are summed with a parallelized reduction to become a single value.
- The G sums are then, in parallel, added with the constant term $-\frac{(\alpha_g^*)^2 - (\alpha_g^{[t-1]})^2}{2\sigma_a^2}$ in the log acceptance rate, compared with G Uniform[0,1) random numbers to determine if the proposed α_g^* s are accepted, and the latest iteration of $\alpha_g^{[t]}$ s are then updated accordingly.

For Z_{ig} and X_{ig} :

- For $i = 1, \dots, N, g = 1, \dots, G$,
 - If Y_{ig} is not zero, we assume there is no dropout and the observed signal is the real signal.

- Otherwise, Z and X are updated simultaneously (i.e. in the same function) and in parallel across $N \times G$ based on $Z^{[t-1]}$ s and $X^{[t-1]}$ s from the last iteration.

For γ_{b0} and γ_{b1} :

- An array with the length of $2 \times B$ (γ_{b0} and γ_{b1} combined) is filled in parallel with γ_{b0}^* and γ_{b1}^* across B based on $\gamma_{b0}^{[t-1]}$ s and $\gamma_{b1}^{[t-1]}$ s from the last iteration.
- Then, for $b = 1, \dots, B$,
 - For γ_{b0} , an array of length $n_b \times G$ (n_b is the number of samples in batch b) is filled in parallel with the $n_b \times G$ elements of the summation part in the log acceptance rate of γ_{b0} , i.e.

$$\sum_{i=1}^{n_b} \sum_{g=1}^G \left\{ (\gamma_{b0}^* - \gamma_{b0}^{[t-1]}) z_{big}^{[t]} + \log \left[\frac{1 + \exp(\gamma_{b1}^{[t-1]} x_{big}^{[t]} + \gamma_{b0}^{[t-1]})}{1 + \exp(\gamma_{b1}^{[t-1]} x_{big}^{[t]} + \gamma_{b0}^*)} \right] \right\}.$$

- For γ_{b1} , an array of length $n_b \times G$ (n_b is the number of samples in batch b) is filled in parallel with the $n_b \times G$ elements of the summation part in the log acceptance rate of γ_{b1} , i.e.

$$\sum_{i=1}^{n_b} \sum_{g=1}^G \left\{ (\gamma_{b1}^* - \gamma_{b1}^{[t-1]}) x_{big}^{[t]} z_{big}^{[t]} + \log \left[\frac{1 + \exp(\gamma_{b1}^{[t-1]} x_{big}^{[t]} + \gamma_{b0}^{[t]})}{1 + \exp(\gamma_{b1}^* x_{big}^{[t]} + \gamma_{b0}^{[t]})} \right] \right\}.$$

- The two arrays are summed through parallelized reduction into 2 single values.
- The B sums of γ_{b0} are then added in parallel with the constant term $-\frac{(\gamma_{b0}^*)^2 - (\gamma_{b0}^{[t-1]})^2}{2\sigma_{z_0}^2}$ in the log acceptance rate, compared with B Uniform[0,1) random numbers to determine if the proposed γ_{b0}^* s are accepted, and the latest iteration of $\gamma_{b0}^{[t]}$ s are then updated accordingly.
- The B sums of γ_{b1} are then added in parallel with the constant term

$$a_\gamma \log\left(\frac{-\gamma_{b1}^*}{-\gamma_{b1}^{[t-1]}}\right) + b_\gamma (\gamma_{b1}^* - \gamma_{b1}^{[t-1]}) - \log \left[\frac{\Gamma(-10\gamma_{b1}^*)}{\Gamma(-10\gamma_{b1}^{[t-1]})} \right]$$

$$-10\gamma_{b1}^* \log(-\gamma_{b1}^{[t-1]}) + 10\gamma_{b1}^{[t-1]} \log(-\gamma_{b1}^*) - 10(\gamma_{b1}^* - \gamma_{b1}^{[t-1]}) (\log(10) + 1)$$

in the log acceptance rate, compared with B Uniform[0,1) random numbers to determine if the proposed γ_{b1}^* s are accepted, and the latest iteration of $\gamma_{b1}^{[t]}$ s are then updated accordingly.

For L_{gk} :

- For $g = 1, \dots, G, k = 2, \dots, K$, acceptance rates and random acceptances for every L_{gk} are computed simultaneously and in parallel.

For p and $(\tau_{\beta 0})^2$:

- For $g = 1, \dots, G, k = 2, \dots, K$,
 - The array of $L_{gk}^{[t]}$ is summed through parallelized reduction into one single value.
 - An array of length $G \times (K - 1)$ is filled in parallel with $1(L_{gk}^{[t]} = 0) \cdot (\beta_{gk}^{[t-1]})^2$, where 1 is the indicator function.
 - This array is then summed through parallelized reduction into one single value.
- A single Beta random variable, parameterized with terms consisting of the sum of $L_{gk}^{[t]}$, is generated as $p^{[t]}$ (p of the latest iteration).
- A single Inv-Gamma random variable, parameterized with terms consisting of the two sums above, is generated as $(\tau_{\beta 0}^{[t]})^2$ ($(\tau_{\beta 0})^2$ of the latest iteration).

For β_{gk} :

- Firstly, an array with the length of $G \times (K - 1)$ is updated in parallel to store $G \times (K - 1)$ β_{gk}^* s, the proposed β_{gk} s based on $\beta_{gk}^{[t-1]}$ s from the last iteration.
- For $g = 1, \dots, G, k = 2, \dots, K$,
 - An array of the length $\sum_{b=1}^B n_b = N$ (no. of samples) is filled in parallel with the N elements of the summation part in the log acceptance rate, i.e.

$$\sum_{b=1}^B \sum_{i=1}^{n_b} I(w_{bi}^{[t-1]} = k) \left\{ (\beta_{gk}^* - \beta_{gk}^{[t-1]}) x_{big}^{[t]} + (\phi_{bg}^{[t-1]} + x_{big}^{[t]}) \log \left(\frac{\phi_{bg}^{[t-1]} + \exp(\alpha_g^{[t]} + \beta_{gk}^{[t-1]} + \nu_{bg}^{[t-1]} + \delta_{bi}^{[t-1]})}{\phi_{bg}^{[t-1]} + \exp(\alpha_g^{[t]} + \beta_{gk}^* + \nu_{bg}^{[t-1]} + \delta_{bi}^{[t-1]})} \right) \right\}.$$

- The array is then summed by with a parallelized reduction to become a single value.
- The $G \times (K - 1)$ sums are then added in parallel with the constant term $-\frac{(\beta_{gk}^*)^2 - (\beta_{gk}^{[t-1]})^2}{2(\tau_{\beta L_{gk}^{[t]}})^2}$ in the log acceptance rate, compared with $G \times (K - 1)$ Uniform[0,1) random numbers to determine if the proposed β_{gk}^* s are accepted, and $\beta_{gk}^{[t]}$ (β_{gk} of the latest iteration) are then updated accordingly.

For ν_{bg} :

- Firstly, an array with the length of $G \times (B - 1)$ is updated in parallel to store $G \times (B - 1)$ ν_{bg}^* s, the proposed ν_{bg} s based on $\nu_{bg}^{[t-1]}$ s from the last iteration.
- For $b = 2, \dots, B, g = 1, \dots, G$,
 - An array with the length of n_b is filled in parallel with the n_b elements of the summation part in the log acceptance rate, i.e.

$$\sum_{i=1}^{n_b} \left\{ (\nu_{bg}^* - \nu_{bg}^{[t-1]}) x_{big}^{[t]} + (\phi_{bg}^{[t-1]} + x_{big}^{[t]}) \log \left[\frac{\phi_{bg}^{[t-1]} + \exp(\alpha_g^{[t]} + \beta_{gk}^{[t]} + \nu_{bg}^{[t-1]} + \delta_{bi}^{[t-1]})}{\phi_{bg}^{[t-1]} + \exp(\alpha_g^{[t]} + \beta_{gk}^{[t]} + \nu_{bg}^* + \delta_{bi}^{[t-1]})} \right] \right\}.$$
 - The array is then summed with a parallelized reduction to become a single value.
- The $(B - 1) \times G$ sums are then added in parallel with the constant term $-\frac{(\nu_{bg}^*)^2 - (\nu_{bg}^{[t-1]})^2}{2\sigma_c^2}$ in the log acceptance rate, compared with $(B - 1) \times G$ Uniform[0,1) random numbers to determine if the proposed ν_{bg}^* s are accepted, and $\nu_{bg}^{[t]}$ (ν_{bg} of the latest iteration) are then updated accordingly.

For δ_{bi} :

- Firstly, an array with the length of N is updated in parallel to store $\sum_{b=1}^B n_b = N$ δ_{bi}^* s, the proposed δ_{bi} s based on $\delta_{bi}^{[t-1]}$ s from the last iteration.
- For $b = 1, \dots, B, i = 2, \dots, n_b$,
 - An array with the length of G is filled in parallel with the G elements of the summation part in the log acceptance rate, i.e.

$$\sum_{g=1}^G \left\{ (\delta_{bi}^* - \delta_{bi}^{[t-1]}) x_{big}^{[t]} + (\phi_{bg}^{[t-1]} + x_{big}^{[t]}) \log \left[\frac{\phi_{bg}^{[t-1]} + \exp(\alpha_g^{[t]} + \beta_{gk}^{[t]} + \nu_{bg}^{[t]} + \delta_{bi}^{[t-1]})}{\phi_{bg}^{[t-1]} + \exp(\alpha_g^{[t]} + \beta_{gk}^{[t]} + \nu_{bg}^{[t]} + \delta_{bi}^*)} \right] \right\}.$$
 - The array is then summed with a parallelized reduction to become a single value.
- The N sums are then added in parallel with the constant term $-\frac{(\delta_{bi}^*)^2 - (\delta_{bi}^{[t-1]})^2}{2\sigma_d^2}$ in the log acceptance rate, compared with G Uniform[0,1) random numbers to determine if the proposed δ_{bi}^* s are accepted, and $\delta_{bi}^{[t]}$ (δ_{bi} of the latest iteration) are then updated accordingly.

For ϕ_{bg} :

- Firstly, an array with the length of $B \times G$ is updated in parallel to store $B \times G$ ϕ_{bg}^* s, the proposed ϕ_{bg} s based on $\phi_{bg}^{[t-1]}$ s from the last iteration.
- For $b = 1, \dots, B, g = 1, \dots, G$,

- Let $\eta_{big}^{[t]} = \exp(\alpha_g^{[t]} + \beta_{g w_{bi}^{[t-1]}}^{[t]} + \nu_{bg}^{[t]} + \delta_{bi}^{[t]})$ denotes the mean gene expression level for gene g in cell i of batch b .
- An array with the length of n_b is filled in parallel with the n_b elements of the summation part in the log acceptance rate, i.e.

$$\sum_{i=1}^{n_b} \left\{ \log \left[\frac{\Gamma(\phi_{bg}^* + x_{big}^{[t]})(\phi_{bg}^*)^{\phi_{bg}^*}}{\Gamma(\phi_{bg}^*)(\phi_{bg}^* + \eta_{big}^{[t]})^{\phi_{bg}^* + x_{big}^{[t]}}} \right] + \log \left[\frac{\Gamma(\phi_{bg}^{[t-1]})(\phi_{bg}^{[t-1]} + \eta_{big}^{[t]})^{\phi_{bg}^{[t-1]} + x_{big}^{[t]}}}{\Gamma(\phi_{bg}^{[t-1]} + x_{big}^{[t]})(\phi_{bg}^{[t-1]})^{\phi_{bg}^{[t-1]}}} \right] \right\},$$

- The array is then summed with a parallelized reduction to become a single value.
- The $B \times G$ sums are then added in parallel with the constant term $(\kappa - 1) \log(\frac{\phi_{bg}^*}{\phi_{bg}^{[t-1]}}) + (1 - \tau)(\phi_{bg}^* - \phi_{bg}^{[t-1]}) + \log \left[\frac{(\phi_{bg}^{[t-1]})^{\phi_{bg}^* - 1} \Gamma(\phi_{bg}^{[t-1]})}{(\phi_{bg}^*)^{\phi_{bg}^{[t-1]} - 1} \Gamma(\phi_{bg}^*)} \right]$ in the log acceptance rate, compared with $B \times G$ Uniform[0,1) random numbers to determine if the proposed ϕ_{bg}^* s are accepted, and $\phi_{bg}^{[t]}$ (ϕ_{bh} of the latest iteration) are then updated accordingly.

For w_{bi} :

- Firstly, an array with the length of N is updated in parallel to store $\sum_{b=1}^B n_b = N$ w_{bi}^* s, the proposed w_{bi} s based on $w_{bi}^{[t-1]}$ s from the last iteration.
- For $b = 1, \dots, B, i = 1, \dots, n_b$,
 - An array with the length of G is filled in parallel with the G elements of the summation part in the log acceptance rate, i.e.

$$\sum_{g=1}^G \left\{ (\beta_{gk^*}^{[t]} - \beta_{gk}^{[t]}) x_{big}^{[t]} + (x_{big}^{[t]} + \phi_{bg}^{[t]}) \log \left[\frac{\exp(\alpha_g^{[t]} + \beta_{gk}^{[t]} + \nu_{bg}^{[t]} + \delta_{bi}^{[t]}) + \phi_{bg}^{[t]}}{\exp(\alpha_g^{[t]} + \beta_{gk^*}^{[t]} + \nu_{bg}^{[t]} + \delta_{bi}^{[t]}) + \phi_{bg}^{[t]}} \right] \right\},$$

where $w_{bi}^* = k^*, w_{bi}^{[t-1]} = k$.

- The array is then summed with a parallelized reduction to become a single value.
- The N sums are then in parallel added with the constant term $\log(\frac{\pi_{bk^*}^{[t-1]}}{\pi_{bk}^{[t-1]}})$ in the log acceptance rate, compared with N Uniform[0,1) random numbers to determine if the proposed w_{bi}^* s are accepted, and $w_{bi}^{[t]}$ (w_{bi} of the latest iteration) are then updated accordingly.

For π_b :

- For $b = 1, \dots, B, k = 1, \dots, K$,
 - An array with the length of n_b is filled in parallel with the value of the function $1(w_{bi} = k)$, where 1 is the indicator function.

- The array is then summed with a parallelized reduction to become a single value.
- The $N \times K$ sums are then added in parallel with a constant ξ , the parameter of the prior distribution.
- For $b = 1, \dots, B$, $\boldsymbol{\pi}_b^{[t]}$, $\boldsymbol{\pi}_b$ of the latest iteration, is updated in parallel by sampling from the Dirichlet distribution with the sums above as parameters.

We have explained the details of the GPU parallelization in Supplementary Notes.

6. Supplementary methods, MCMC: (i) for all MH steps, the authors should report what is the acceptance rate? The latter enables the reader to assess how well is the MCMC mixing [3] (ii) in the updates for x , the authors indicate that variance of the proposal varies across iterations. If this is the case, the proposal is not symmetric and there is a missing term in the acceptance probability. Moreover, please do include the derivation of the acceptance probability (some of the terms from the equation above appear to be missing).

We thank the reviewer for the suggestions. We have calculated the acceptance rate for all of the Metropolis-Hastings (MH) steps and detailed the update for X .

First, we calculated the acceptance rates for the simulation, the hematopoietic study, and the pancreas study. The parameters using MH updates include $\gamma_{b0}, \gamma_{b1}, \alpha_g, \beta_{gk}, \nu_{bg}, \delta_{bi}$ and ϕ_{bg} . Gelman et al. [12] studied the most efficient symmetric jumping kernels for simulating a normal target distribution using the Metropolis algorithm. Theoretically, they found the optimal target acceptance rate is 0.44 when the target distribution is a one-dimensional normal distribution. The acceptance rates of most parameters of BUSseq except γ are close to 0.44 (**Table R8**). Although the acceptance rate of γ is low (**Table R8**), the comparison of the posterior means $\hat{\gamma}$ and their true values γ in **Table R9** shows that the inference of γ is precise. Moreover, the EPSR factors mentioned in the response to the reviewer’s last major point confirm the good mixing of cell type specific mean expression levels and batch effects.

We thank the reviewer for referring us to Roberts and Rosenthal [37]. Roberts and Rosenthal [37] mention that adaptive MCMC algorithms can have better mixing properties than comparable non-adaptive algorithms. However, because the proposed distribution of the adaptive MCMC requires calculation of empirical estimates of the covariance structure of the target distribution based on all previous runs, the algorithm can be slowed down. As our non-adaptive MCMC gives reasonably good mixing, we choose to stay with the non-adaptive algorithm.

We now discuss the acceptance rates on pages 8-9, 14 and 18 of the main manuscript and explain the details in Supplementary Notes.

Study	Parameter	Min	First quartile	Median	Mean	Third quartile	Max
Simulation	α	0.196	0.267	0.378	0.352	0.415	0.526
	β	0.209	0.258	0.274	0.290	0.286	0.850
	ν	0.293	0.486	0.625	0.596	0.702	0.850
	δ	0.116	0.172	0.198	0.208	0.233	0.338
	ϕ	0.132	0.352	0.499	0.529	0.740	0.873
	γ	0.010	0.012	0.025	0.026	0.038	0.044
Hematopoietic	α	0.073	0.356	0.471	0.488	0.619	0.951
	β	0.122	0.622	0.723	0.705	0.816	0.991
	ν	0.122	0.432	0.525	0.541	0.646	0.961
	δ	0.234	0.367	0.379	0.422	0.462	0.722
	ϕ	0.002	0.011	0.014	0.016	0.019	0.097
	γ	0.001	0.012	0.049	0.217	0.254	0.771
Pancreas	α	0.058	0.177	0.206	0.215	0.246	0.433
	β	0.116	0.408	0.476	0.490	0.565	0.951
	ν	0.088	0.279	0.366	0.415	0.553	0.798
	δ	0.135	0.283	0.304	0.340	0.441	0.515
	ϕ	0.006	0.015	0.032	0.152	0.173	0.857
	γ	0.004	0.014	0.020	0.038	0.027	0.179

Table R8: The acceptance rates of all BUSseq parameters updated by Metropolis steps in simulation and two real applications. The same table is shown as **Supplementary Table 3**.

Parameter	γ_{10}	γ_{11}	γ_{20}	γ_{21}	γ_{30}	γ_{31}	γ_{40}	γ_{41}
True values	-0.500	-0.200	-0.500	-0.200	-0.500	-0.200	-0.500	-0.200
Posterior mean	-0.475	-0.200	-0.461	-0.204	-0.461	-0.201	-0.443	-0.201

Table R9: Comparison of true values and posterior means for γ in the simulation study. γ can be precisely estimated by the MCMC algorithm, even if the acceptance rate is low. The same table is shown as **Supplementary Table 18**.

Here, we detail the derivation of the MH step for updating \mathbf{x} . In the MCMC algorithm, $z_{big}^{[t]}$ and $x_{big}^{[t]}$ are updated sequentially for (b, i, g) , if $y_{big} = 0$:

$$\begin{aligned}
z_{big}^{[t]} & \begin{cases} = 1 & , \text{ if } x_{big}^{[t-1]} > 0; \\ \sim \text{Bernoulli}\left(\frac{\exp(\gamma_{b0}^{[t-1]})}{1+\exp(\gamma_{b0}^{[t-1]})}\right) & , \text{ if } x_{big}^{[t-1]} = 0. \end{cases} \\
x_{big}^{[t]} & \begin{cases} = 0 & , \text{ if } z_{big}^{[t]} = 0; \\ \propto \frac{\exp(\gamma_{b0}^{[t-1]} + \gamma_{b1}^{[t-1]} x_{big}^{[t]})}{1+\exp(\gamma_{b0}^{[t-1]} + \gamma_{b1}^{[t-1]} x_{big}^{[t]})} \frac{\Gamma(\phi_{bg}^{[t-1]} + x_{big}^{[t]}) (\mu_{big}^{[t-1]})^{x_{big}^{[t]}}}{\Gamma(x_{big}^{[t]}) (\phi_{bg}^{[t-1]} + \mu_{big}^{[t-1]})^{\phi_{bg}^{[t-1]} + x_{big}^{[t]}}} & , \text{ if } z_{big}^{[t]} = 1. \end{cases}
\end{aligned}$$

where $\mu_{big}^{[t-1]} = \exp(\alpha_g^{[t-1]} + \beta_{gw_{bi}^{[t-1]}}^{[t-1]} + \nu_{bg}^{[t-1]} + \delta_{bi}^{[t-1]})$, and $\Gamma(\cdot)$ represents the Gamma function.

When $z_{big}^{[t]} = 1$, we find that

$$\begin{aligned}
f(x|\gamma_{b0}^{[t-1]}, \gamma_{b1}^{[t-1]}, \mu_{big}^{[t-1]}, \phi_{bg}^{[t-1]}) & \propto \frac{\exp(\gamma_{b0}^{[t-1]} + \gamma_{b1}^{[t-1]} x)}{1 + \exp(\gamma_{b0}^{[t-1]} + \gamma_{b1}^{[t-1]} x)} \cdot \frac{\Gamma(\phi_{bg}^{[t-1]} + x) (\mu_{big}^{[t-1]})^x}{\Gamma(x) (\phi_{bg}^{[t-1]} + \mu_{big}^{[t-1]})^{\phi_{bg}^{[t-1]} + x}} \\
& \propto \frac{\exp(\gamma_{b0}^{[t-1]} + \gamma_{b1}^{[t-1]} x)}{1 + \exp(\gamma_{b0}^{[t-1]} + \gamma_{b1}^{[t-1]} x)} \cdot p_{NB}(x|\mu_{big}^{[t-1]}, \phi_{bg}^{[t-1]}), \quad (\text{R.4})
\end{aligned}$$

where $p_{NB}(x|\mu, \phi)$ denotes the probability mass function (PMF) of the negative binomial distribution with mean μ and overdispersion ϕ . Therefore, we incorporate a Metropolis-Hasting (MH) step [18]. We sample x_{big}^* from the proposal distribution $NB(\mu_{big}^{[t-1]}, \phi_{bg}^{[t-1]})$ and accept the proposal with probability.

$$\begin{aligned}
\rho & = \min\left\{ \frac{f(x_{big}^*|\gamma_{b0}^{[t-1]}, \gamma_{b1}^{[t-1]}, \mu_{big}^{[t-1]}, \phi_{bg}^{[t-1]}) p_{NB}(x_{big}^{[t-1]}|\mu_{big}^{[t-1]}, \phi_{bg}^{[t-1]})}{f(x_{big}^{[t-1]}|\gamma_{b0}^{[t-1]}, \gamma_{b1}^{[t-1]}, \mu_{big}^{[t-1]}, \phi_{bg}^{[t-1]}) p_{NB}(x_{big}^*|\mu_{big}^{[t-1]}, \phi_{bg}^{[t-1]})}, 1 \right\} \\
& = \min\left\{ \frac{\frac{\exp(\gamma_{b0}^{[t-1]} + \gamma_{b1}^{[t-1]} x_{big}^*)}{1+\exp(\gamma_{b0}^{[t-1]} + \gamma_{b1}^{[t-1]} x_{big}^*)} p_{NB}(x_{big}^*|\mu_{big}^{[t-1]}, \phi_{bg}^{[t-1]}) p_{NB}(x_{big}^{[t-1]}|\mu_{big}^{[t-1]}, \phi_{bg}^{[t-1]})}{\frac{\exp(\gamma_{b0}^{[t-1]} + \gamma_{b1}^{[t-1]} x_{big}^{[t-1]})}{1+\exp(\gamma_{b0}^{[t-1]} + \gamma_{b1}^{[t-1]} x_{big}^{[t-1]})} p_{NB}(x_{big}^{[t-1]}|\mu_{big}^{[t-1]}, \phi_{bg}^{[t-1]}) p_{NB}(x_{big}^*|\mu_{big}^{[t-1]}, \phi_{bg}^{[t-1]})}, 1 \right\} \\
& = \min\left\{ \frac{\frac{\exp(\gamma_{b0}^{[t-1]} + \gamma_{b1}^{[t-1]} x_{big}^*)}{1+\exp(\gamma_{b0}^{[t-1]} + \gamma_{b1}^{[t-1]} x_{big}^*)}}{\frac{\exp(\gamma_{b0}^{[t-1]} + \gamma_{b1}^{[t-1]} x_{big}^{[t-1]})}{1+\exp(\gamma_{b0}^{[t-1]} + \gamma_{b1}^{[t-1]} x_{big}^{[t-1]})}}, 1 \right\} \\
& = \min\left\{ \frac{1 + \exp(-\gamma_{b0}^{[t-1]} - \gamma_{b1}^{[t-1]} x_{big}^{[t-1]})}{1 + \exp(-\gamma_{b0}^{[t-1]} - \gamma_{b1}^{[t-1]} x_{big}^*)}, 1 \right\}.
\end{aligned}$$

Although our proposal is not symmetric, its effect has been accounted for in the acceptance ratio. Because the conditional distribution of x_{big} in Equation (R.4) is the result of multiplying

the odds ratio and the PMF of the negative binomial distribution, the proposal distribution can be canceled out with the PMF part of the conditional distribution. As a result, the acceptance rate of updating x_{big} only involves the odds ratio terms.

If $y_{big} > 0$, then $z_{big}^{[t]} = 0$ and $x_{big}^{[t]} = y_{big}$.

We have updated the Supplementary Notes with the more detailed derivation of the MH step.

Minor comments.

1. Page 2, first paragraph "an excessive number of zeros that result from dropout events". While scRNAseq experiments indeed exhibit a high frequency of zero counts, not all zeros are missing data or "dropouts" (a gene's expression is not captured by the assay). Instead, some of these can correspond to "biological" zeros, where the gene is not expressed in a cell. The latter is particularly relevant when mix populations (with transcriptionally distinct cell types) are processed. I believe it is important to discuss this distinction and how the different types of zeros would be treated by BUSseq.

We thank the reviewer for this helpful comment. We have revised the sentence on page 2 of the main manuscript as follows:

"Moreover, compared to bulk RNA-seq data, scRNA-seq data can have an excessive number of zeros that result from either biological zeros—that is, a gene is not expressed in a given cell—or dropout events—that is, the expression of some genes are not detected even though they are actually expressed in the cell due to amplification failure prior to sequencing."

As pointed out by the reviewer, when we observe a zero read count $Y_{big} = 0$, there are two possibilities: a non-expressed gene—biological zeros—or a dropout event. When gene g is not expressed in cell i of batch b ($X_{big} = 0$), we always have $Y_{big} = 0$; when gene g is actually expressed in cell i of batch b ($X_{big} > 0$) but a dropout event occurs, we can only observe $Y_{big} = 0$, and hence $Z_{big} = 1$.

We have added discussions on pages 4 and 30 of the main manuscript.

2. Page 2, second paragraph "selecting control genes is often difficult". Recently, [4] identified control genes that are reproducible and conserved across species (human and mouse). Such genes could be used as a reference in RUVseq or related methods. I suggest the authors to include the reference to [4], discussing that there has been some progress in the area.

We thank the reviewer for the suggestion and the reference. We have revised the original sentence on page 2 of the main manuscript,

"However, selecting control genes is often difficult for scRNA-seq experiments."

as

"However, selecting control genes is still challenging for scRNA-seq experiments, and recently there has been active research on identifying stably expressed genes that are reproducible and conserved across species for single cells [29]."

3. Page 4, last paragraph. The authors indicate that MCMC was used to perform inference. However, the reference provided is generic (a book which introduces several MCMC methods). Please state the specific MCMC method here.

We thank the reviewer for the suggestion. We actually used the Metropolis-within-Gibbs algorithm proposed by Tierney [42]. As the Gibbs sampler fails to update a series of parameters in our proposed model, including cell type effects β and batch effects ν , we incorporate extra Metropolis-Hastings steps. We have revised the relevant sentence on page 5 of the main manuscript.

5. Page 23. As BUSseq is also introduced as an imputation and normalisation method, it would be good to see (for a single batch) how does the corrected data compare to what is generated by existing normalisation (e.g. [5]) and imputation tools (such as those discussed in [6]).

We thank the reviewer for the suggestion.

Lun et al. [32] proposed a deconvolution normalization method and compared its performance with that of five existing normalization methods, including DESeq normalization [31], trimmed mean of M-values (TMM) normalization [38] and library size normalization. The authors drew a scatter plot of the estimated values by each normalization method against the true cell size factors and evaluated how close these scattered points were to the diagonal.

We compared the performance of the above methods with the size factor estimated by BUSseq on the simulation dataset, as the “golden truth” of parameters and latent variables is only available in the simulation study. Moreover, to avoid the impact of batch effects, we focused on the first batch with $b = 1$. Following Lun et al. [32], we applied the *estimateSizeFactorsForMatrix* function of the R package *DESeq2* with the option *geoMeans = gm* for DESeq normalization, the *calcNormFactors* function of the R package *edgeR* for TMM normalization and directly calculated the total read count of each cell as the size factor for library size normalization. For the deconvolution method, we ran the *computeSumFactors* function with and without prior clustering, respectively, using the R package *scan*. In BUSseq, the estimate of the size factor is the exponential of the posterior mean $\exp(\hat{\delta}_{1i})$. **Figure R26** shows that BUSseq correctly estimates the cell size factors and outperforms all existing methods.

We now present the results on page 9 of the main manuscript and explain the details in Supplementary Notes.

Figure R26: Performance of benchmarked normalization methods with the simulation dataset. The size factor estimates for all cells are plotted against the true values for **(a)** BUSseq, **(b)** DESeq, **(c)** TMM, **(d)** library size normalization, **(e)** deconvolution and **(f)** deconvolution with clustering. The axes are in the log-scale. For comparison, the size factor estimates by each method are scaled such that the grand mean across cells was the same as that for the true values. The diagonal line means that the scaled estimate is equal to the true factors. The four cell types are colored by blue, orange, green and pink, respectively. The same figure is shown as **Supplementary Figure 4**.

Andrews and Hemberg [1] compared six different imputation methods—SAVER [22], DrImpute [14], scImpute [28], dca [10], MAGIC [43] and knn-smooth [44]. Only the first three imputation methods use models to distinguish true biological zeros from zero induced by the dropout events. Therefore, we compared the first three methods with BUSseq in terms of the Euclidean distance between the imputed values and true values for all of the observed zero values and the accuracy in recovering the rates of biological zeros. We continue to use the first batch of our simulation dataset so that the true biological zero rate can be calculated for the underlying read count data x_{big} .

Following Andrews and Hemberg [1], we first conducted library-size normalization and then applied SAVER to the normalized data. We applied scImpute to the raw observed count data and set the dropout threshold option as 0.5 and the number of clusters as the true value four.

We took the natural logarithm of the normalized data plus one and applied DrImpute to the log-scale data with the number of clusters set to four.

For comparison, we first calculated the Euclidean distance between the imputed values and the underlying true read counts x_{big} s for all of the observed zeros. Notably, however, only scImpute and BUSseq worked on the raw count data. Therefore, to be fair, we standardized the imputed values of SAVER by multiplying the size factor of each cell back to obtain $\hat{x}_{1ig}^{imputed}$. Similarly, we exponentiated the imputed values output by DrImputed, subtracted one, and finally multiplied the resulting values by the size factor of each cell for standardization to get $\hat{x}_{1ig}^{imputed}$. Finally, we calculated the Euclidean distance d between x_{1ig} and $\hat{x}_{1ig}^{imputed}$ for all of the observed zeros:

$$d = \sum_{i=1}^{n_1} \sum_{g=1}^G I(y_{1ig} = 0) \sqrt{[\log(1 + x_{1ig}) - \log(1 + \hat{x}_{1ig}^{imputed})]^2},$$

where n_1 denotes the number of cells in the first batch, x_{1ig} is the underlying true read count of gene g for cell i in the first batch and $\hat{x}_{1ig}^{imputed}$ represents the (standardized) imputed values. We measured distance on the log-scale to eliminate the impact of outliers.

We further compared the zero rates of the imputed values. As the three compared methods generated continuous values, we rounded the (standardized) imputed values to imputed read counts and then calculated the zero rates.

According to **Table R10**, the read counts imputed by BUSseq are closest to the true values in terms of both the Euclidean distance and the rates of biological zeros.

We now present the results on page 9 of the main manuscript and explain the details in Supplementary Notes.

Count Data	True	Observed	BUSseq	SAVER	DrImpute	scImpute
d	0	169525.4	4115.656	68894.57	44003.8	37254.39
Zero rates	27.92%	44.13%	27.58%	38.12%	22.27%	24.39%

Table R10: Comparison of Euclidean distance d and zero rates in the observed count data \mathbf{Y} (Observed), the underlying count data \mathbf{X} (True), the data imputed by BUSseq, the data imputed by SAVER, the data imputed by DrImpute and the data imputed by scImpute for the first batch of the simulation dataset. The same table is shown as **Supplementary Figure 4**.

6. Supplementary material "Comparison measures" section. The authors specify that silhouette metrics were calculated based on tSNE coordinates. However, tSNE is known to not preserve the global structure of the data (such as the separation between clusters). As such, tSNE coordinates

do not provide an appropriate metric for clustering performance.

We thank the reviewer for the comment.

We acknowledge the concern raised by the reviewer, as t-SNE performs nonlinear dimension reduction. Nevertheless, as t-SNE has been widely applied in analyses of scRNA-seq data, we feel that the calculation of silhouette coefficients based on t-SNE coordinates can also provide a quantification of clustering performance that is consistent with researchers' impressions of t-SNE plots [16]. Furthermore, we also computed the silhouette coefficients according to the first 10 PCs for all of the methods. We set the number of PCs as 10, as the lowest dimension of the corrected data output by all of the methods is 10, consistent with ZINB-WaVE. According to the heatmap of silhouette coefficients (**Figure R27**), under the new metric, BUSseq also has outstanding clustering performance in both the hematopoietics study and the pancreas study.

We have added a discussion in Supplementary Notes.

Figure R27: Silhouette coefficients of all cells in the (a) hematopoietics study and (b) pancreas study based on the first 10 PCs of the corrected data by each method. We add these two boxplots to **Supplementary Figure 10**.

7. Implementation. In order to facilitate adoption, the authors might consider to implement their method as an R/Python library. In particular, submission to Bioconductor would support usage across different OS (Github repository states that current implementation requires Linux OS).

Thanks a lot for this helpful suggestion. We have wrapped C++ source code into an R package *BUSseq* on the GitHub (<https://github.com/songfd2018/BUSseq-Rpackage>). This package can be applied on both Linux and Windows operating system. We will submit this package to Bioconductor soon.

The GitHub link is also added on page 37 of the main manuscript.

References

- [1] [doi:10.1186/s13059-017-1305-0](https://doi.org/10.1186/s13059-017-1305-0)
- [2] [doi: 10.1038/s41592-019-0425-8](https://doi.org/10.1038/s41592-019-0425-8)
- [3] <https://doi.org/10.1198/jcgs.2009.06134>
- [4] <https://doi.org/10.1093/gigascience/giz106>
- [5] [doi:10.1186/s13059-016-0947-7](https://doi.org/10.1186/s13059-016-0947-7)
- [6] <https://doi.org/10.12688/f1000research.16613.2>

Reviewer #2:

This paper provides two significant contributions to the field of single-cell gene expression analysis (scRNA-seq). First, the authors provide rigorous conditions and assumptions that must be met for cell types to be identifiable in the presence of batch effects. Secondly, they provide an integrated hierarchical Bayesian model for batch effect correction and clustering. The authors do a good job of using examples to clarify new concepts as they are presented. Both the theoretical results and the method are well motivated, convincing, and I believe they would be interesting and useful to other researchers. I am especially grateful to the authors for making all of their code publicly available, and for including a detailed algorithm in the supplementary materials that will facilitate reproducibility. Despite the overall quality of the paper, I have several concerns especially in how the authors benchmarked against other methods and in how applicable their method will be to large-scale datasets that are becoming prevalent in the field.

We truly appreciate the comments from Dr. Townes.

Major concerns

** The assumption of zero inflation or dropout on top of the negative binomial sampling is controversial. Recent studies have claimed that there is no zero inflation in droplet scRNA-seq with UMIs, for example: <https://www.biorxiv.org/content/10.1101/582064v1.abstract>. The authors do not distinguish between UMI counts and read counts even though these two have dramatically different distributional properties, including but not limited to differences in zero inflation. If this paper were restricting its scope to read count data only (eg SMART-seq2) I would not raise this concern, but they do include some MARS-seq and CEL-seq2 datasets which to my understanding do have UMIs. Therefore, I would like to see a more careful discussion of the pros and cons of the zero inflation component with UMI vs non-UMI datasets. Specifically, I would like to see them re-fit their model on the UMI datasets without the zero inflation component and assess which model performed better. It would certainly increase the number of users of their method if they allow the user to decide which samples/ batches should have zero inflation (eg read count data) and which should not (eg UMI count data) and it would likely be faster to run since there would be fewer parameters. By providing both options and demonstrating which is effective on different datasets, they would bypass the controversy over whether they should or should not use*

zero inflation. Also, whenever a dataset is first described, they should say whether it has UMIs or not.

We thank the reviewer for this invaluable point and all of the suggestions.

The two hematopoietic datasets were profiled by MARS-seq and SMART-seq2, respectively. Two of the pancreas datasets were profiled by CEL-seq2, and the other two were profiled by SMART-seq2. As the reviewer mentioned, CEL-seq2 and MARS-seq incorporate UMI in the cDNA generation, whereas SMART-seq2 does not ([49]). We have added **Table R11** to indicate whether a dataset has UMIs or not.

In our BUSseq model, a dropout event occurs for gene g in cell i of batch b if the observed value $y_{big} = 0$ but the imputed count data $\hat{x}_{big} > 0$. This identification allows us to calculate the frequency of dropout events in each batch. We calculate the zero rate of each batch as follows:

$$\rho_0 = \frac{1}{G \cdot n_b} \sum_{g=1}^G \sum_{i=1}^{n_b} I(y_{big} = 0), \quad (\text{R.5})$$

and compute the dropout rate as the proportion of dropout events among the observations with zero counts:

$$\rho_d = \frac{\sum_{g=1}^G \sum_{i=1}^{n_b} I(y_{big} = 0 \text{ and } \hat{x}_{big} > 0)}{\sum_{g=1}^G \sum_{i=1}^{n_b} I(y_{big} = 0)}.$$

Study	Protocol	UMI	ρ_0	ρ_d	$\hat{\rho}_0^{BUSseq}$	$ \hat{\rho}_0^{BUSseq} - \rho_0 $	$\hat{\rho}_0^{nzf}$	$ \hat{\rho}_0^{nzf} - \rho_0 $
Hematopoietic	MARS-seq	Yes	0.892	< 0.001	0.887	0.005	0.874	0.018
	SMART-seq2	No	0.421	< 0.001	0.424	0.003	0.445	0.024
Pancreas	CEL-seq2	Yes	0.689	< 0.001	0.625	0.064	0.682	0.007
	CEL-seq2	Yes	0.517	0.017	0.558	0.041	0.617	0.100
	SMART-seq2	No	0.609	0.167	0.531	0.078	0.430	0.179
	SMART-seq2	No	0.480	0.551	0.485	0.005	0.329	0.161

Table R11: Zero-count rates and dropout rates of the hematopoietic and pancreas studies. ρ_0 denotes the observed zero rate in each batch; ρ_d represents the inferred dropout rate by BUSseq; $\hat{\rho}_0^{BUSseq}$ denotes the posterior mean of zero rate inferred by BUSseq; and $\hat{\rho}_0^{nzf}$ represents the posterior mean of zero rate inferred by a reduced model of BUSseq that ignores dropout events and hence uses negative binomial distribution without zero inflation, abbreviated as “BUSseq-nzf.” BUSseq detects the existence of dropout events automatically and performs better than BUSseq-nzf in terms of the posterior predictive check of zero rates. The same table is shown as **Table 1** of the main manuscript.

Consequently, we can see from **Table R11** that the analyses by BUSseq confirm that SMART-seq2 does have many more dropout events than the other protocols with the UMI barcodes,

as expected, although the difference is marginal for the hematopoietic study. As the second dataset in the pancreas study adopted the CEL-seq2 protocol and still has 1.7% dropout events, we feel that it may be safer to apply the full BUSseq model to the data and let data speak for themselves whether there is a need to model dropout events rather than conclude that the incorporation of UMI guarantees the elimination of dropout events.

We have added discussions on pages 14, 18 and 34 of the main manuscript.

Nevertheless, we feel that the reviewer’s suggestion of providing the users’ the option to choose whether they want to model zero inflation is a fantastic idea. Therefore, we first refitted the model without zero inflation for each dataset. The proposed model then becomes a negative binomial mixture model, abbreviated as “nzf” in **Table R11**, with the following structure:

$$Pr(W_{bi} = k) = \pi_{bk}, \sum_{k=1}^K \pi_{bk} = 1;$$

$$Y_{big}|W_{bi} = k \sim NB(\mu_{big}, \phi_{bg}), \log(\mu_{big}) = \alpha_g + \beta_{gk} + \nu_{bg} + \delta_{bi}.$$

We compared BUSseq and BUSseq-nzf in terms of both model fitting, using a posterior predictive check [11] for zero rates, and clustering accuracy with ARIs. In the posterior predictive check, we take MCMC samples of all of the parameters after the burn-in iterations to simulate replicated datasets $Y_j^{rep}, j = 1, 2, \dots, J$ for G genes and $N = \sum_{b=1}^B n_b$ cells, where J denotes the total number of collected iterations after burn-ins. In our real data analyses, we ran 8,000 iterations with the first 4,000 iterations as burn-ins, so we generated $J = 8,000 - 4,000 = 4,000$ replicated datasets for both the hematopoietic and pancreas studies. For each generated replicate dataset, we calculated the zero rates of each batch according to Equation (R.5). Finally, we averaged the zero rates over all J iterations to calculate the posterior mean $\hat{\rho}_0$ of the zero rate of each batch and compared it with the corresponding observed zero rate. As a result, comparing the the seventh and ninth columns of **Table R11**, we can see that BUSseq fits the zero rate much better than BUSseq-nzf.

For ARIs between the estimated cell type labels \hat{w}_{bi} and FACS labels, the value drops from 0.582 by BUSseq to 0.564 by BUSseq-nzf and 0.608 by BUSseq to 0.437 by BUSseq-nzf. As the modeling of dropout events does not reduce the clustering accuracy, and improves the model fit dramatically, our recommendation is to use the full version of BUSseq.

We have added discussions on pages 14, 18 and 34 of the main manuscript and detail out the “BUSseq-nzf” model in Supplementary Notes.

Nevertheless, we completely agree with the reviewer that it is better to leave the option of modeling dropout events to the discretion of the user. We have updated both our parallel CPU and our GPU code for BUSseq to allow the user to decide which batches should have zero

inflation and which should not.

** The choice of filtering for informative genes plays a crucial role in downstream analysis. The authors should compare how sensitive their clustering algorithm is to alternative gene filtering methods besides the Brennecke one they mention. At a minimum, they should try just filtering on something like "highly expressed" genes as described in this systematic comparison: <https://f1000research.com/articles/7-1141/v2>. It would also be nice to see how sensitive BUSseq accuracy is to different choices of the number of genes included in the filtering.*

We thank the reviewer for the suggestion.

In addition to filtering the highly variable genes, we also followed Duò et al. [8] in selecting the genes with the highest mean expression levels across all cells. In preprocessing, Duò et al. [8] first excluded the low quality genes and cells. Next, the normalization factors for the count values were calculated by the deconvolution method of the R package *scater*. For gene filtering, the authors retained genes with the top 10% highest average of log-scale expression values across all cells.

In the hematopoietic study, there are two batches of gene expression count data, one generated on the SMART-seq2 platform by Nestorowa et al. [34] with accession number GSE81682 and the other generated on the MARS-seq platform by Paul et al. [35] with accession number GSE72857. After removing low-quality genes, GSE81682 includes 46,175 genes, and GSE72857 contains 27,297 genes. Notably, the genes in the GSE81682 dataset were named by Ensembl ID, but the genes in the GSE72857 dataset were named by Gene Symbol. The R package *biomaRt* was applied to query the corresponding Gene Symbol by Ensembl ID. However, if we had retained only genes with the top 10% highest average of log-scale expression values, only 74 common genes would have been filtered out by both datasets. Instead, we retained genes with the top 50% highest average of log-scale expression values such that 2,843 common genes were retained from the two datasets for downstream analysis. We denote these genes as top ranked genes in terms of mean expression levels across cell types (TEG).

For comparison, in our original analysis, following Haghverdi et al. [16], we had a dataset with 3,470 highly variable genes (HVG). We compared the results of applying BUSseq to the new dataset with the 2,843 TEGs to that of our original analysis (**Table R12**). We can see that the diagonal elements dominate each row. The ARI between the cell labeling of TEGs and FACS labeling is 0.482, whereas the ARI between the cell labeling of HVGs and FACS labeling is 0.582.

We also compared the intrinsic genes identified by BUSseq when applied to the two datasets with different gene filtering criteria. The 3,470 selected HVGs and the 2,843 selected TEGs share only 475 genes. Of these 475 genes, 194 genes are identified as intrinsic genes by the dataset with HVGs, and 193 of these 194 genes were also called intrinsic genes by the dataset

with TEGs (the dataset with TEGs called a total of 174 intrinsic genes for the 475 shared genes). Thus, the selection of intrinsic genes is robust to gene filtering methods.

TEG \ HVG	Cluster1 (LTHSC & MPP)	Cluster2 (CMP)	Cluster3 (CLP)	Cluster4 (CMP & MEP)	Cluster5 (GMP)	Cluster6 (MEP)
Cluster1 (LTHSC & MPP)	556	80	6	6	1	0
Cluster2 (CMP)	66	171	52	80	60	1
Cluster3 (CLP)	2	0	164	0	0	0
Cluster4 (CMP & MEP)	1	5	5	573	0	79
Cluster5 (GMP)	14	315	24	66	1216	0
Cluster6 (MEP)	0	0	2	19	0	1085

Table R12: Contingency table between the cell type labeling learned by BUSseq when filtering HVGs (columns) and filtering genes with high mean expression levels (rows) in the hematopoietic study. The same table is shown as **Supplementary Table 15**.

In the pancreas study, four batches of gene expression count data were assayed. GSE81076 was profiled using the CEL-seq protocol, and GSE85241 was profiled using the CEL-seq1 protocol. Two remaining batches (GSE86473 and EMATB-5061) were profiled using the SMART-seq2 protocol. After removing low-quality cells and genes, we normalized the raw count data through the deconvolution methods via R package *scater* [32]. Four batches were obtained, containing 7,676, 10,312, 11,383 and 12,486 genes, respectively. If we had retained only genes with the 10% highest mean expression levels, no common gene would have been shared by the four batches. Instead, we also selected the genes with the 50% highest mean expression levels in terms of log-normalized values. As a result, 408 common TEGs were retained.

Similar to the pancreas study, we compared the cell labeling results of two gene filtering methods (**Table R13**). Once again, the diagonal elements dominate each row. The ARI between the cell labeling of TEGs and FACS labeling is 0.437, whereas the ARI between the cell labeling of HVGs and FACS labeling is 0.608.

In the pancreas study, no gene is shared between the 2,480 HVGs and the 408 TEGs, BUSseq still offers consistent cell clustering in this extreme case. Therefore, BUSseq is robust to the gene filtering strategy.

TEG \ HVG	Cluster1 (Alpha)	Cluster2 (Alpha)	Cluster3 (Alpha)	Cluster4 (Beta)	Cluster5 (Gamma & Delta)	Cluster6 (Acinar)	Cluster7 (Ductal)	Cluster8 (Other)
Cluster1 (Alpha)	1452	143	7	0	73	0	0	0
Cluster2 (Alpha)	194	678	106	447	139	1	1	0
Cluster3 (Beta)	5	9	59	6	8	7	7	20
Cluster4 (Beta)	2	2	6	785	192	0	0	0
Cluster5 (Gamma & Delta)	22	285	4	14	559	1	0	0
Cluster6 (Acinar)	0	0	3	1	1	625	9	0
Cluster7 (Ductal)	0	0	8	0	0	66	916	6
Cluster8 (Other)	0	0	0	0	0	1	2	223

Table R13: Contingency table between the cell type labeling learned by BUSseq when filtering HVGs (columns) and filtering genes with high mean expression levels (rows) in the pancreas study. The same table is shown as **Supplementary Table 16**.

According to the ARI calculations for the two studies, we recommend filtering HVGs during preprocessing. An intrinsic gene that well distinguishes cell types may be highly expressed in one cell type but lowly expressed in other cell types. However, such a gene will be missed by filtering according to TEGs, as its mean expression level across all of the cells is low. The filtering of TEGs is likely to select genes whose expression levels are high but remain the same across all of the cell types, and such genes can provide very limited information for differentiating cell types.

A comparison of the two gene filtering strategies has been added to *Discussion* of the main manuscript on pages 24-25 and the details are explained in Supplementary Notes.

** I appreciate that the authors mention the computational limitations of MCMC in the discussion/conclusion section. However, this is not sufficient. While it is true that 1 hour of computation time is small compared to wet lab experiments, this is on a fairly small dataset by current standards and was done on a high-performance computing system with a GPU. If the authors want their method to become widely used, I would encourage them to offer an approximate version of BUSseq that runs much faster. For example, if they discard the zero inflation component and perform maximization of the penalized likelihood using expectation-maximization, or variational inference they may get a result that is "good enough" in a fraction of the time. Even if it is not feasible to implement such an alternative inference scheme in the current paper, they should at least offer suggestions for how it could be done in the discussion, so that others can build on their ideas. Furthermore, they should address memory consumption of the parallelized and non-parallelized implementation. What is the maximum number of cells/genes that BUSseq can handle on an ordinary desktop computer?*

We thank the reviewer for the suggestions.

We have derived and implemented the expectation-maximization (EM) algorithm for a simplified BUSseq model, as suggested by the reviewer. In the simplified version, we do not consider the dropout events or cell-specific size factor δ_{bi} . The reason is that the derivative with respect to the cell type effects β_{gk} in the M-step of the EM algorithm is related to the cell-specific size

factor of all of the cells, the batch effects of all of the batches, and the overdispersion parameters of all of the genes. As a result, we have to update all of these parameters with the dimension $G * (K + 2B - 1) + \sum_{b=1}^B n_b$ simultaneously by the Newton-Raphson method, and the dimension is too high to calculate the inverse of the observed information matrix. Instead, if we estimate the size factors according to some normalization methods and remove them from the model, then we can update the cell type effects β_{gk} , batch effects ν_{bg} and overdispersion parameters ϕ_{bg} gene by gene. The BUSseq model then reduces to:

$$\begin{aligned} Pr(W_{bi} = k) &= \pi_{bk}, \sum_{k=1}^K \pi_{bk} = 1; \\ Y_{big}|W_{bi} = k &\sim NB(\mu_{big}, \phi_{bg}), \log(\mu_{big}) = b_{gk} + \nu_{bg}, \end{aligned} \quad (\text{R.6})$$

where b_{gk} denotes the log-scale mean expression levels of gene g and cell type k in the first batch, which is equivalent to the summation of the log-scale baseline expression levels α_g and the cell type effects β_{gk} in the BUSseq model, that is, $b_{gk} = \alpha_g + \beta_{gk}$.

Furthermore, in the M step, we need to update the estimates for the parameters of the negative binomial distribution. [26] suggested that it is better to reparameterize the overdispersion parameter ϕ_{bg} as its reciprocal $\xi_{bg} = \frac{1}{\phi_{bg}}$. Then, we have the complete-data log-likelihood as

$$\begin{aligned} l(\mathbf{Y}, \mathbf{Z}|\boldsymbol{\pi}, \boldsymbol{\alpha}, \mathbf{b}, \boldsymbol{\xi}) &= \sum_{b=1}^B \sum_{i=1}^{n_b} \sum_{k=1}^K I(w_{bi} = k) [\log(\pi_{bk}) + \sum_{g=1}^G \log(\Gamma(y_{big} + 1)) + \log(\Gamma(\phi_{bg})) \\ &\quad - \log(\Gamma(y_{big} + \frac{1}{\xi_{bg}}))] + y_{big} \log(\mu_{big}) - \frac{1}{\xi_{bg}} \log(\xi_{bg}) - (y_{big} + \frac{1}{\xi_{bg}}) \log(\mu_{big} + \frac{1}{\xi_{bg}}) \end{aligned}$$

In the E step, let $\Theta^{(t)} = (\boldsymbol{\pi}^{(t)}, \mathbf{b}^{(t)}, \boldsymbol{\nu}^{(t)}, \boldsymbol{\xi}^{(t)})$ denote all of the parameters at the t^{th} iteration.

$$E(I(w_{bi} = k)|\Theta^{(t)}) = Pr(w_{bi} = k|\Theta^{(t)}) \propto \pi_{bk}^{(t)} \prod_{g=1}^G \frac{\exp(y_{big}(b_{gk}^{(t)} + \nu_{bg}^{(t)}))}{(\exp(b_{gk}^{(t)} + \nu_{bg}^{(t)}) + \frac{1}{\xi_{bg}^{(t)}})^{y_{big} + \frac{1}{\xi_{bg}^{(t)}}}}.$$

Therefore,

$$p_{bik}^{(t)} \triangleq Pr(w_{bi} = k|\Theta^{(t)}) = \frac{\pi_{bk}^{(t)} \prod_{g=1}^G \frac{\exp(y_{big}(b_{gk}^{(t)} + \nu_{bg}^{(t)}))}{(\exp(b_{gk}^{(t)} + \nu_{bg}^{(t)}) + \frac{1}{\xi_{bg}^{(t)}})^{y_{big} + \frac{1}{\xi_{bg}^{(t)}}}}}{\sum_{l=1}^K \pi_{bl}^{(t)} \prod_{g=1}^G \frac{\exp(y_{big}(b_{gl}^{(t)} + \nu_{bg}^{(t)}))}{(\exp(b_{gl}^{(t)} + \nu_{bg}^{(t)}) + \frac{1}{\xi_{bg}^{(t)}})^{y_{big} + \frac{1}{\xi_{bg}^{(t)}}}}}.$$

Finally, we obtain the function $Q(\Theta|\Theta^{(t)})$:

$$\begin{aligned} Q(\Theta|\Theta^{(t)}) &\triangleq E(l(\mathbf{Y}, \mathbf{Z}|\boldsymbol{\pi}, \boldsymbol{\alpha}, \mathbf{b}, \boldsymbol{\xi})|\Theta^{(t)}) \\ &= \sum_{b=1}^B \sum_{i=1}^{n_b} \sum_{k=1}^K p_{bik}^{(t)} [\log(\pi_{bk}) + \sum_{g=1}^G (\log(\Gamma(\phi_{bg})) - \log(\Gamma(y_{big} + \frac{1}{\xi_{bg}}))) \\ &\quad + y_{big}(b_{gk} + \nu_{bg}) - \frac{1}{\xi_{bg}} \log(\xi_{bg}) - (y_{big} + \frac{1}{\xi_{bg}}) \log(\exp(b_{gk} + \nu_{bg}) + \frac{1}{\xi_{bg}})] + C, \end{aligned}$$

where C is a constant irrelevant to the parameter Θ .

In the M step, we first take the derivative of the function $Q(\Theta|\Theta^{(t)})$ with respect to π_{bk} , $b = 1, 2, \dots, B, k = 1, 2, \dots, K - 1$ to obtain

$$\frac{\partial Q(\Theta|\Theta^{(t)})}{\partial \pi_{bk}} = \sum_{i=1}^{n_b} \frac{p_{bik}^{(t)}}{\pi_{bk}} - \sum_{i=1}^{n_b} \frac{p_{biK}^{(t)}}{1 - \sum_{l=1}^{K-1} \pi_{bkl}} = 0.$$

Therefore,

$$\pi_{bk} = \frac{\sum_{i=1}^{n_b} p_{bik}^{(t)}}{\sum_{l=1}^K \sum_{i=1}^{n_b} p_{bil}^{(t)}} = \frac{1}{n_b} \sum_{i=1}^{n_b} p_{bik}^{(t)}.$$

To update $\mathbf{b}_g, \boldsymbol{\nu}_g^{(t)}$ and $\boldsymbol{\xi}_g$, we define $\eta_g = (b_{g1}, b_{g2}, \dots, b_{gk}, \nu_{2g}, \nu_{3g}, \dots, \nu_{bg}, \xi_{1g}, \xi_{2g}, \dots, \xi_{Bg})$. Note that $\nu_{1g} = 0$, for all g . For a one-way negative binomial model,

$$y_{ij} \sim NB(\mu_i, \frac{1}{\xi_i}), i = 1, 2, \dots, n, j = 1, 2, \dots, m. \quad (\text{R.7})$$

It is easy to directly obtain the MLE $\hat{\mu}_i = \frac{1}{m} \sum_{j=1}^m y_{ij}$, and the MLE of ξ_i is given by solving the first derivative $\frac{\partial l(\boldsymbol{\mu}, \boldsymbol{\xi})}{\partial \xi_i} = 0$ at $\mu_i = \hat{\mu}_i$. Here, we actually consider a two-way negative binomial model. Thus, all parameters η_g of a given gene g are not separable in the first and second derivatives of the function $Q(\Theta|\Theta^{(t)})$, so we apply the Newton-Rasphon method to find the MLE of ξ_g . We set the initial value as $\eta_g^{(0)} = (\boldsymbol{\beta}_g^{(t)}, \boldsymbol{\nu}_g^{(t)}, \boldsymbol{\xi}_g^{(t)})$ and update $\eta_g^{(s+1)} = \eta_g^{(s)} - (\frac{\partial^2 Q(\Theta|\Theta^{(t)})}{\partial \eta_g \partial \eta_g^T})^{-1} \frac{\partial Q(\Theta|\Theta^{(t)})}{\partial \eta_g}$ until the difference $\|\eta_g^{(s+1)} - \eta_g^{(s)}\|_\infty$ is smaller than the pre-specified tolerance. Finally, $\mathbf{b}_g^{(t+1)}, \boldsymbol{\nu}_g^{(t+1)}$ and $\boldsymbol{\xi}_g^{(t+1)}$ are updated as the corresponding values of $\eta_g^{(s+1)}$.

Next, for gene $g, g = 1, 2, \dots, G$, we take the first derivative of the function $Q(\Theta|\Theta^{(t)})$ with respect to η_g .

$$\begin{aligned}
\frac{\partial Q(\Theta|\Theta^{(t)})}{\partial b_{gk}} &= \sum_{b=1}^B \sum_{i=1}^{n_b} p_{bik}^{(t)} \left[y_{big} - \left(y_{big} + \frac{1}{\xi_{bg}} \right) \frac{\exp(b_{gk} + \nu_{bg})}{\exp(b_{gk} + \nu_{bg}) + \frac{1}{\xi_{bg}}} \right] \\
&= \sum_{b=1}^B \sum_{i=1}^{n_b} p_{bik}^{(t)} \frac{1}{1 + \xi_{bg} \exp(b_{gk} + \nu_{bg})} [y_{big} - \exp(b_{gk} + \nu_{bg})] \\
&= \sum_{b=1}^B \sum_{i=1}^{n_b} p_{bik}^{(t)} \rho_{bgk} [y_{big} - \exp(b_{gk} + \nu_{bg})]; \\
\frac{\partial Q(\Theta|\Theta^{(t)})}{\partial \nu_{bg}} &= \sum_{b=1}^B \sum_{i=1}^{n_b} p_{bik}^{(t)} \left[y_{big} - \left(y_{big} + \frac{1}{\xi_{bg}} \right) \frac{\exp(b_{gk} + \nu_{bg})}{\exp(b_{gk} + \nu_{bg}) + \frac{1}{\xi_{bg}}} \right] \\
&= \sum_{i=1}^{n_b} \sum_{k=1}^K p_{bik}^{(t)} \rho_{bgk} [y_{big} - \exp(b_{gk} + \nu_{bg})]; \\
\frac{\partial Q(\Theta|\Theta^{(t)})}{\partial \xi_{bg}} &= \sum_{i=1}^{n_b} \sum_{k=1}^K p_{bik}^{(t)} \left[\frac{1}{\xi_{bg}^2} \Psi\left(\frac{1}{\xi_{bg}}\right) - \frac{1}{\xi_{bg}^2} \Psi\left(\frac{1}{\xi_{bg}} + y_{big}\right) + \frac{1}{\xi_{bg}^2} \log(\xi_{bg}) - \frac{1}{\xi_{bg}^2} \right. \\
&\quad \left. + \frac{1}{\xi_{bg}^2} \log\left(\exp(b_{gk} + \nu_{bg}) + \frac{1}{\xi_{bg}}\right) + \left(y_{big} + \frac{1}{\xi_{bg}} \right) \frac{\frac{1}{\xi_{bg}^2}}{\exp(b_{gk} + \nu_{bg}) + \frac{1}{\xi_{bg}}} \right] \\
&= \sum_{i=1}^{n_b} \sum_{k=1}^K p_{bik}^{(t)} \frac{1}{\xi_{bg}^2} \left[\Psi\left(\frac{1}{\xi_{bg}}\right) - \Psi\left(\frac{1}{\xi_{bg}} + y_{big}\right) - \log(\rho_{bgk}) + \rho_{bgk} \xi_{bg} (y_{big} - \exp(b_{gk} + \nu_{bg})) \right],
\end{aligned}$$

where $\rho_{bgk} = \frac{1}{1 + \xi_{bg} \exp(b_{gk} + \nu_{bg})}$ together with

$$\begin{aligned}
\frac{\partial \rho_{bgk}}{\partial b_{gk}} &= \frac{\partial \rho_{bgk}}{\partial \nu_{bg}} = -\frac{\xi_{bg} \exp(b_{gk} + \nu_{bg})}{(1 + \xi_{bg} \exp(b_{gk} + \nu_{bg}))^2} = -\rho_{bgk} (1 - \rho_{bgk}), \\
\frac{\partial \rho_{bgk}}{\partial \xi_{bg}} &= -\frac{\exp(b_{gk} + \nu_{bg})}{(1 + \xi_{bg} \exp(b_{gk} + \nu_{bg}))^2} = -\frac{1}{\xi_{bg}} \rho_{bgk} (1 - \rho_{bgk}),
\end{aligned}$$

and $\Psi(x) = (\log(\Gamma(x)))' = \frac{\Gamma'(x)}{\Gamma(x)}$.

Furthermore, we have the second derivative $\frac{\partial^2 Q(\Theta|\Theta^{(t)})}{\partial \eta_g \partial \eta_g^T}$ by

$$\begin{aligned}
\frac{\partial^2 Q(\Theta|\Theta^{(t)})}{\partial b_{gk} \partial b_{gl}} &= \begin{cases} -\sum_{b=1}^B \sum_{i=1}^{n_b} p_{bik}^{(t)} \rho_{bgk} [y_{big}(1 - \rho_{bgk}) - \exp(b_{gk} + \nu_{bg}) \rho_{bgk}] & l = k \\ 0 & l \neq k \end{cases} \\
\frac{\partial^2 Q(\Theta|\Theta^{(t)})}{\partial \nu_{bg} \partial \nu_{cg}} &= \begin{cases} -\sum_{i=1}^{n_b} \sum_{k=1}^K p_{bik}^{(t)} \rho_{bgk} [y_{big}(1 - \rho_{bgk}) - \exp(\beta_{gk} + \nu_{bg}) \rho_{bgk}] & c = b \\ 0 & c \neq b \end{cases} \\
\frac{\partial^2 Q(\Theta|\Theta^{(t)})}{\partial \xi_{bg} \partial \xi_{cg}} &= \begin{cases} \sum_{i=1}^{n_b} \sum_{k=1}^K \frac{p_{bik}^{(t)}}{\xi_{bg}^4} [2\xi_{bg}(\Psi(\frac{1}{\xi_{bg}} + y_{big}) - \Psi(\frac{1}{\xi_{bg}})) + (\Psi'(\frac{1}{\xi_{bg}} + y_{big}) - \Psi'(\frac{1}{\xi_{bg}})) \\ + \xi_{bg}(2\log(\rho_{bgk}) + (1 - \rho_{bgk})) - \xi_{bg}^2 \rho_{bgk}(2 - \rho_{bgk})(y_{big} - \exp(b_{gk} + \nu_{bg}))] & c = b \\ 0 & c \neq b \end{cases} \\
\frac{\partial^2 Q(\Theta|\Theta^{(t)})}{\partial b_{gk} \partial \nu_{bg}} &= -\sum_{i=1}^{n_b} p_{bik}^{(t)} \rho_{bgk} [y_{big}(1 - \rho_{bgk}) - \exp(b_{gk} + \nu_{bg}) \rho_{bgk}] \\
\frac{\partial^2 Q(\Theta|\Theta^{(t)})}{\partial b_{gk} \partial \xi_{bg}} &= -\sum_{i=1}^{n_b} \frac{p_{bik}^{(t)}}{\xi_{bg}} \rho_{bgk} (1 - \rho_{bgk}) [y_{big} - \exp(b_{gk} + \nu_{bg})] \\
\frac{\partial^2 Q(\Theta|\Theta^{(t)})}{\partial \nu_{bg} \partial \xi_{cg}} &= \begin{cases} -\sum_{i=1}^{n_b} \sum_{k=1}^K \frac{p_{bik}^{(t)}}{\xi_{bg}} \rho_{bgk} (1 - \rho_{bgk}) [y_{big} - \exp(b_{gk} + \nu_{bg})] & c = b \\ 0 & c \neq b \end{cases}
\end{aligned}$$

We first tested the above EM algorithm with a toy simulation dataset, which is generated according to the simplified model in Equation (R.6). The simulation dataset is comprised of three batches, each with 100 cells from three cell types. Unfortunately, the EM algorithm is very sensitive to the initial values of the parameters. The batch effects ν_{bg} and ξ_{bg} may explode to infinity when the algorithm starts from improper initial values. Previous studies have also noted such phenomenon. Saha and Paul [39] mentioned that under a one-way negative binomial model as in Equation (R.7), the convergence of the estimating equation with respect to ξ_i relies on the initial values. Willson et al. [46] also discussed the difficulty of estimating ξ_i and claimed that “the MLE of ξ_i would be highly variable and affected by even small changes in sample values.”

In the MCMC algorithm, we randomly assign each cell to K cell types as the initial values of w_{bi} . Next, according to the random cell type assignment, we give crude estimates of the cell type effects and batch effects and take them as the initial values. However, this strategy will induce a set of improper initial values for the EM algorithm, perhaps because the function $Q(\Theta|\Theta^{(t)})$ is not convex with respect to η_g . Instead, if we apply k-means clustering to the first batch and estimate the cell type effects by the clustering results as the initial values, then in the toy example, the EM algorithm can correctly distinguish cell type effects from batch effects.

However, under the simulation setting in our manuscript, the initialization according to k-means clustering also fails. The reason is that the first batch does not include all of the cell types and k-means wrongly clusters cells from two different cell types together.

Beyond the MLE, Clark and Perry [5] discussed the method of moment (MM) and the maximum extended quasilielihood (EQL) estimates of ξ_i , and Saha and Paul [39] proposed a double extended quasilielihood (DEQL) estimation process. Nevertheless, “all the existing works pointed out that finding the MLE for overdispersion parameters is challenging, since the score function may have no root, one root or more than one root” [6]. These strategies become even harder to implement for a two-way negative model, as in our simplified version of BUSseq.

Consequently, the EM algorithm still suffers from some limitations, especially the sensitivity to initial values, even for one-way negative binomial models. Due to the high-dimensionality of scRNA-seq data, it is hard to achieve an efficient and robust deterministic algorithm at this stage for BUSseq. Therefore, we still opt to use MCMC for inference.

We have discussed the choice of EM algorithm and MCMC on pages 23-24 of the main manuscript.

We understand the reviewer’s concern regarding the speed of MCMC. We have witnessed the popularity of MCMC for phylogeny over the years, and therefore we envision that MCMC will become equally popular for the analysis of scRNA-seq data with advances in computational power and statistical algorithms. For the current scRNA-seq protocols, we have seen a trade off between the number of cells and the number of detected genes per cell. For currently available datasets, we feel that BUSseq works reasonably well in terms of speed given its accuracy, but we will definitely work hard to further increase its speed in our future work.

Now let’s analyze and record the memory consumption. In our algorithm, the dominating memory consumption is the storage of the posterior samples in the MCMC algorithm. Fortunately, given a machine with small random-access memory (RAM), we can write the posterior samples to the hard disk every n_s iterations and then load the recorded posterior samples one iteration a time when conducting posterior inference. Consequently, by storing the posterior samples from the RAM to the hard disk every a few iterations, we can control the memory consumption. For instance, we can run a total of 4,000 iterations and store the posterior samples to the hard disk after every 1,000 iterations. Theoretically, when n_s is large, the RAM consumption is proportional to n_s and to the total number of parameters $|\Theta|$. $|\Theta| = O(N + 2(B + K)G)$, where B denotes the number of batches and K represents the number of cell types. On the other hand, when n_s is small, the RAM consumption is dominated by the two count data matrices \mathbf{X} and \mathbf{Y} and the dropout indicators \mathbf{Z} , whose memory consumption is $O(N \cdot G)$. Therefore, the RAM consumption should be $O((N + 2(B + K)G)n_s + 3NG)$ when both K and B are far smaller than N or G , the typical case for scRNA-seq data.

To verify the theoretical analysis, we recorded how the RAM consumption and the running time varied with the number of genes G , the number of cells N and the number of iterations n_s per hard-disk writing for the CPU parallel version of the code. **Table R14** shows that the RAM consumption increases linearly with respect to the number of cells N , the number of genes

G and the number of iterations per storage n_s . Meanwhile, the running time is linear to the number of cells N and the number of genes G . Although we can expect that writing to the hard-disk can take longer than writing to RAM, it turned out that the running time is almost constant to the frequency of writing to hard disk when running BUSseq on a cluster node with two 8-core Dual Intel Xeon E5-2650 v2 2.60GHz processors. Therefore, in principle, as long as we write out the posterior samples to the hard disk frequently (e.g. every 10 iterations), we can run BUSseq on a huge scRNA-seq dataset with 100,000 cells and 3,000 genes on a desktop with 8G RAM. However, taking time consumption into consideration, we recommend running BUSseq on a cluster node instead.

We present the scalability of BUSseq on page 24 of the main manuscript and explain the details in Supplementary Notes.

$G = 3000$ and $n_s = 1000$					
N	1000	2000	4000	10000	20000
Running time (hour)	1.768	3.263	6.343	16.426	29.702
RAM peak usage (GB)	0.442	0.509	0.643	1.046	1.726
$N = 1000$ and $n_s = 1000$					
N	1500	3000	4500	6000	9000
Running time (hour)	0.868	1.768	2.416	3.411	5.123
RAM peak usage (GB)	0.229	0.442	0.661	0.867	1.291
$N = 1000$ and $G = 3000$					
n_s	100	200	500	1000	2000
Running time (hour)	1.738	1.726	1.733	1.768	1.783
RAM peak usage (GB)	0.100	0.138	0.252	0.442	0.799
n_s	2	5	10	20	50
Running time (hour)	1.711	1.679	1.718	1.778	1.752
RAM peak usage (GB)	0.062	0.062	0.065	0.069	0.081

Table R14: Running time and RAM usage for the CPU parallel version of BUSseq with respect to the number of cells, the number of genes and the number of iterations per storage, respectively. The CPU code were run on 8 cores of Dual Intel Xeon E5-2650 v2 2.60GHz Processors.

Figure R28: Running time and RAM usage for (a,b) the CPU parallel version of BUSseq on 8 cores of Dual Intel Xeon E5-2650 v2 2.60GHz processors and (c,d) the GPU parallel version of BUSseq on a single core of an Intel Xeon Gold 6132 processor and one NVIDIA Tesla P100 GPU. (a) When $G = 3,000$ and $n_s = 1,000$, N varies from 1,000 to 20,000. (b) When $N = 1,000$ and $n_s = 1,000$, G varies from 1,500 to 9,000. (c) When $G = 3,000$, N varies from 1,000 to 50,000. (d) When $N = 1,000$, G varies from 1,500 to 9,000. The same figure is shown as **Figure 9** of the main manuscript.

Figure R29: Running time and RAM usage for the CPU parallel version of BUSseq on 8 cores of Dual Intel Xeon E5-2650 v2 2.60GHz processors, when n_s varies from 2 to 2,000 with $N = 1,000$ and $G = 3,000$ fixed. This figure is plotted in log-scale. The same figure is shown as **Supplementary Figure 18**.

$G = 3000$						
N	1000	2000	4000	10000	20000	50000
Running time (hour)	0.365	0.440	0.588	0.856	1.240	2.455
RAM peak usage (GB)	1.712	1.925	2.347	3.624	5.745	10.372
$N = 1000$						
N	1500	3000	4500	6000	9000	
Running time (hour)	0.194	0.365	0.535	0.714	1.057	
RAM peak usage (GB)	1.304	1.712	2.122	2.532	3.347	

Table R15: Running time and RAM usage for the GPU parallel version of BUSseq with respect to the number of cells and the number of genes, respectively. The GPU code were run on a single core of an Intel Xeon Gold 6132 Processor and one NVIDIA Tesla P100 GPU.

** In comparing against other methods, the simulation is not very convincing, because it is drawn from the BUSseq generative model. They should also try a simulation where the assumptions of BUSseq are at least slightly violated. Also, the numbers of zeros in the simulation are unrealistically low. They should have at least one batch (preferably most of the batches) where the number of zeros is $> 80\%$.*

We thank the reviewer for the suggestion.

First, we have followed the Reviewer 1's suggestion and considered a scenario where the overdispersion parameters vary across cell types even within the same batch. More specifically, we regard a gene as a highly expressed gene (HEG) in a given cell type if its expression level

in the given cell type is higher than its mean expression level across all of the cell types. In the new simulation setting, we let HEGs have lower overdispersion in their highly expressed cell types. Specifically, under our parameterizations of the negative binomial distribution $f_{NB}(x; \mu, \phi) = C_x^{\phi+x-1} (\frac{\mu}{\mu+\phi})^x (\frac{\phi}{\mu+\phi})^\phi$, the variance is $\mu + \frac{\mu^2}{\phi}$. Thus, a gene with a higher value of ϕ_{bg} has lower overdispersion. If a gene is highly expressed in a given cell type, then we set its overdispersion parameter ϕ_{bg}^{High} in that cell type to be five times the overdispersion parameter ϕ_{bg}^{Low} in the other cell types. Under such a model-misspecified setting, BUSseq can still perfectly cluster cells with the ARI between the true cell type labels and the estimated cell type indicators equal to one. BUSseq can also successfully recover the log-scale mean expression levels for each cell type $\alpha + \beta$, the location batch effects ν , the underlying expression levels before dropout events \mathbf{X} and the cell-specific size factors δ (**Figure R1**).

Next, we constructed a simulation dataset with one batch having more than 80% zero counts. More specifically, the zero rates of the four batches are 53.55%, 61.13%, 60.95% and 80.99%, respectively. We ran BUSseq for 4,000 iterations and regarded the first 2,000 iterations as burn-ins. Even though the last batch has more than 80% zero counts, BUSseq correctly identifies the presence of five cell types among the cells and estimates all the parameters well (**Figure R30**).

We present the results on page 12 of the main manuscript and explain the details in Supplementary Notes.

Figure R30: BUSseq correctly recovers the parameters even if one batch has more than 80% zero counts. (a) True log-scale mean expression level for each cell type $\alpha + \beta$. Each row represents a gene, and each column corresponds to a cell type. (b) True batch effects. Each row represents a gene, and each column corresponds to a batch. (c) True underlying expression levels \mathbf{X} . Each row represents a gene, and each column corresponds to a cell. The upper colored bar indicates the batches, and the lower colored bar represents the cell type. (d) The simulated observed data \mathbf{Y} . (e) BIC plot for different numbers of cell types. (f) The estimated log-scale mean expression level for each cell type $\hat{\alpha} + \hat{\beta}$. (g) Estimated batch effects. (h) Imputed expression levels $\hat{\mathbf{X}}$. (i) Corrected count data $\hat{\mathbf{X}}$ grouped by batches. (j) Scatter plot of the estimated versus true cell-specific size factors. The same figure is shown as **Supplementary Figure 5**.

* They should simulate a much larger number of genes (say 20,000 instead of 3,000) with many (say 17,000) having low information/ noise and batch effects only, then apply the same gene filtering approach used in their real data analysis prior to running the methods on the filtered dataset.

We thank the reviewer for the suggestion.

Following the reviewer’s suggestion, we simulated a dataset with $G = 20,000$ genes and a total number of $N = 1,000$ cells of $K = 5$ cell types, measured by $B = 4$ batches. Among all 20,000 genes, there are 1,326 intrinsic genes, which are differentially expressed between at least two different cell types.

We then followed the same gene filtering step in the real data analyses to select the top 12,000 highly variable genes (HVG) within each batch using the *trendVar* and *decomposeVar* function in the R package *scrn* [33]. As a result, we obtained 3,461 common HVGs across four batches. Of the 1,326 assumed intrinsic genes, 1,223 are chosen as HVGs for the downstream analysis.

Next, we applied BUSseq to the dataset with HVGs and varied the number of cell types K from three to eight. In the MCMC algorithm, we ran 4,000 iterations and regarded the first 2,000 iterations as burn-ins. BUSseq correctly identifies the presence of five cell types, as the BIC attains the minimum value at $K = 5$ (**Figure R31e**). Meanwhile, 1,236 intrinsic genes are identified by BUSseq when we control the Bayesian false discovery rate at 5%. All 1,223 genes are detected, and the actual false discovery rate is $13/1,236 \times 100\% = 1.05\%$. Moreover, BUSseq can accurately estimate the log-scale mean expression levels $\alpha_g + \beta_{gk}$ of each cell type, the batch effects ν_{bg} , and the cell specific size factors δ_{bi} , simultaneously.

We present the results on page 13 of the main manuscript.

Figure R31: BUSseq correctly recovers the parameters when gene filtering is performed before applying BUSseq. (a) True log-scale mean expression level for each cell type $\alpha + \beta$. Each row represents a gene, and each column corresponds to a cell type. (b) True batch effects. Each row represents a gene, and each column corresponds to a batch. (c) True underlying expression levels \mathbf{X} . Each row represents a gene, and each column corresponds to a cell. The upper colored bar indicates the batches, and the lower colored bar represents the cell types. (d) The simulated observed data \mathbf{Y} . (e) The BIC plot under different numbers of cell types. (f) The estimated log-scale mean expression level for each cell type $\hat{\alpha} + \hat{\beta}$. (g) Imputed expression levels $\hat{\mathbf{X}}$. (h) Corrected count data $\hat{\mathbf{X}}$ grouped by batch. (i) Scatter plot of the estimated versus true cell-specific size factors. The same figure is shown as **Supplementary Figure 7**

* *The real data comparisons do not use any droplet technologies. If this is intentional, the authors should explicitly say their method is intended for plate-type scRNA-seq and doesn't apply to droplet scRNA-seq such as 10x, drop-seq, indrop etc. Otherwise, they need to include a droplet dataset in their comparisons.*

We thank the reviewer for the great comment. Reviewer 1 suggested running BUSseq on lung adenocarcinoma (LUAD) cell lines dataset [41] profiled by the CELseq2, 10x Chromium and Drop-seq protocols. Accordingly, we have added a third real data analysis example to pages 19-21 of the main manuscript.

Specifically, one experiment of the lung adenocarcinoma cell line dataset assayed three lung adenocarcinoma cell lines—HCC827, H1975 and H2228—on three platforms with CELseq2, 10x Chromium and Drop-seq protocols, respectively. As a result, each batch consists of three cell types, and data from different batches have different levels of sparsity. We downloaded the raw count data from the GitHub repository “https://github.com/LuyiTian/sc_mixology.” We selected the top 6,000 highly variable genes (HVGs) within each batch using the *trendVar* and *decomposeVar* functions of the R package *scrn* [33] and obtained 2,267 common HVGs across three batches.

The t-SNE and PCA plots of the raw count data (**Figure R32**) show that significant batch effects occurred across the three protocols. Thus, we applied BUSseq and varied the number of cell type K from 2 to 6. The Bayesian information criterion (BIC) obtains its minimum value at $K = 4$ (**Figure R33**). Although the BIC selects four cell types instead of three cell lines, two of the four identified clusters actually correspond to two subpopulations of the H1975 cell lines (**Table R16**). We further visualize the log-scale mean expression levels of intrinsic genes of the four learned cell types (**Figure R34a**). The first two cell types have similar expression patterns, but some differentially expressed genes are observed between them. Moreover, the PCA (**Figure R35c**) and t-SNE (**Figure R35a**) plots demonstrate the high level of similarity of the first two estimated cell types. Meanwhile, the PCA (**Figure R35d**) and tSNE (**Figure R35b**) plots confirm that the corrected count data \tilde{x}_{big} obtained by BUSseq cluster cells by cell type rather than by batch (**Figure R34b**).

We also applied the benchmarked methods to evaluate their clustering accuracy. Once again, BUSseq outperforms all of the other methods (see **Table R17**).

Figure R32: Patterns of raw count data of the LUAD study. (a) t-SNE plot colored by cell line (H1975, H2228 and HCC827). (b) t-SNE plot colored by protocol (CELseq2, 10x Chromium, and Dropseq). (c) PCA plot colored by cell line (H1975, H2228 and HCC827). (d) PCA plot colored by protocol (CELseq2, 10x Chromium, and Dropseq). The two t-SNE plots are shown as **Figure 8** of the main manuscript, and the two PCA plots are shown as **Supplementary Figure 13**.

Figure R33: BIC curve of the first LUAD study. The same figure is shown as **Supplementary Figure 14**

	HCC827	H1975	H2228
1	1	317	2
2	1	191	0
3	424	0	0
4	0	3	462

Table R16: The contingency table between the estimated cell types $\hat{w}_{bi} \in \{1, 2, 3, 4\}$ and the known cell-line labels. The same table is shown as **Supplementary Table 10**.

Figure R34: Patterns of intrinsic genes of the LUAD study. **(a)** Heatmap of log-scale mean expression levels $\alpha + \beta_k$ of cell types $k = 1, 2, 3, 4$. Each row represents an intrinsic gene. The colored bar indicates the cell type of each column. **(b)** Heatmap of the log-scale corrected count data by BUSseq. Each row again represents an intrinsic gene, but each column corresponds to a cell. The upper colored bar indicates the estimated cell type of each cell, and the lower colored bar denotes the batch to which the corresponding cell belongs. These two heatmaps are shown as **Figure 8** of the main manuscript.

Figure R35: Patterns of corrected count data of BUSseq of the LUAD study. **(a)** t-SNE plot colored by cell line (H1975, H2228 and HCC827). **(b)** t-SNE plot colored by protocol (CELseq2, 10x Chromium, and Dropseq). **(c)** PCA plot colored by cell line (H1975, H2228 and HCC827). **(d)** PCA plot colored by protocol (CELseq2, 10x Chromium, and Dropseq). The two t-SNE plots are shown as **Figure 8** of the main manuscript, and the two PCA plots are shown **Supplementary Figure 13**.

Methods	BUSseq	LIGER	MNN	Scanorama	scVI	Seurat	ZINB-WaVE
ARI	0.8406	0.8250	0.6497	0.3237	0.6368	0.4294	0.3983

Table R17: The ARI of benchmarked methods for the first set of LUAD datasets.

** In comparing against methods such as ZINB-WaVE, they need to be more explicit about what the number of latent factors was set to and what clustering algorithm was applied to the factors. It would be easy to make a dimension reduction method look bad by setting the number of PCs too low then applying k-means clustering. For a realistic comparison they should vary the*

number of PCs and use a Louvain or Leiden-type algorithm downstream of all methods that don't produce clusters automatically. They need to generally do a better job of explaining how the competing methods produce clusters in the introduction. It's OK if this is put in the supplement but currently it's nowhere in the manuscript.

We thank the reviewer for these suggestions.

As for the ZINB-WaVE method, we set the number of factors as 10 (see page 15 of Risso et al. [36]—*Clustering methods* paragraph in Methods). Specifically, the vignette of R package *zinbwave* suggests clustering cells using the *FindNeighbors* and *FindClusters* functions of the R package *Seurat* [40]. When the *FindClusters* function is run, the Louvain algorithm is the default modularity optimization method for clustering. In our analyses, we decide to follow the protocol of Risso et al. [36] faithfully.

We truly appreciate the reviewer's suggestion that we explain the competing methods in detail. We have summarized all of the clustering strategies of each benchmarked method in **Table R18** and elaborated the details in the Supplementary Notes Section *Benchmarked batch-effect-correction methods* (**Table R18**).

Method	Output	Clustering	ARI by their own strategies		ARI by k-means clustering	
			hematopoietic	Pancreas	hematopoietic	Pancreas
BUSseq	The logarithm of the corrected count data of the identified intrinsic genes	The cluster indicators of mixture model	0.5822	0.6080	0.4757	0.8797
LIGER	The cell-specific factor loadings	SFN graph + Louvain community detection	0.3066	0.5421	0.3188	0.8331
MNN	The logarithm of the corrected count data	SNN graph + Walktrap algorithm	0.5754	0.2793	0.4522	0.4220
Scanorama	The batch-corrected data	k-means	0.5184	0.5272	0.5184	0.5272
scVI	The low-dimensional embedding of cells	k-means	0.1969	0.2819	0.4444	0.5202
Seurat	The first 30 principal components of the corrected data	SNN graph + modularity optimization	0.2663	0.2868	0.3413	0.1828
ZINB-WaVE	The low-dimensional embedding of cells	SNN graph + modularity optimization	0.3484	0.3804	0.4892	0.6404

Table R18: The outputs and original clustering strategies of all of the benchmarked methods. We list the ARIs of each method with reference to its original clustering method and the unified k-means clustering, with the number of clusters set to 7 as provided by the FACS labels, respectively. The same table is shown as **Supplementary Table 7**.

* *Many scRNA-seq experiments seek to uncover rare cell types. The authors should artificially reduce the number of cells in one or more of the cell types from a real data comparison and see whether/how the performance of BUSseq changes as the number of cells in each type becomes*

unevenly distributed. The theoretical results suggest that each pair of batches has to contain at least two overlapping cell types, but what happens if one or both of these cell types are only represented by say 3 cells in each batch? This could be related to the experimental design concepts by showing how "power" to detect cell types changes as a function of "sample size".

We thank the reviewer for the suggestion.

We conducted subsampling experiments with the lung adenocarcinoma (LUAD) cell lines dataset [41] to mimic unevenly distributed rare cell types. Here, we consider the three pure cell lines that correspond to the HCC827, H1975 and H2228 cell lines assayed by the CELseq 2 protocol, respectively, in the LUAD dataset. **Table R19** shows the cell numbers of the three pure cell lines in the four batches. We reduced the number of HCC827 cells by random sampling and investigated whether we can still identify HCC827 cells when they become rare.

Cell line mixture	Batch 1	Batch 2	Batch 3	Batch 4
HCC827	18	18	18	18
H1975	19	19	19	19
H2228	9	10	10	10

Table R19: The distribution of each cell line mixture in the four batches.

Specifically, we subsampled m HCC827 cells from each batch, varying m among (10, 8, 6, 5, 4, 3, 2, 1). Note that $m = 18$ indicates that all HCC827 cells were involved. We then applied BUSseq to the resulting datasets with $K = 3$. For each dataset, we selected the top 6,000 HVGs within each of the four batches using the R package *scran* [33] and took the common HVGs across the four batches for the downstream analysis. We ran BUSseq for 8,000 iterations and discarded the first 4,000 iterations as burn-ins. Finally, we calculated the ARIs between the identified cell clusters and the true cell type labels. BUSseq perfectly clusters the rare cell type HCC827 unless there is only one HCC827 cell in each batch (**Table R20**).

We present the results on page 23 of the main manuscript and explain the details in Supplementary Notes.

The number of HCC827	$m = 18$	$m = 10$	$m = 8$	$m = 6$	$m = 5$	$m = 4$	$m = 3$	$m = 2$	$m = 1$
ARI	1.00	1.00	1.00	1.00	1.00	1.00	1.00	1.00	0.785

Table R20: The ARI between the known cell line types and the estimated cell type indicators by BUSseq under different downsampling levels of the HCC827 cells. The same table is shown as **Supplementary Table 13**.

** Methods "Assignment of FACS cell type labels to learned clusters"- the authors need to clarify whether they did this label assignment before or after computing adjusted Rand index (ARI)*

or other measures of cluster accuracy. ARI does not require the labels of clusters to match between the two partitions being compared. The assignment of FACS labels is appropriate only for interpretation of the clusters, not for assessment of accuracy.

We sincerely thank the reviewer for pointing out this potential confusion. We intended to explain how we gave each cell type identified by BUSseq a name for easier interpretation in the section “Assignment of FACS cell type labels to learned clusters” in Methods. However, the word “assignment” may be misleading as pointed out by the reviewer. This word is irrelevant to computing the ARI or other measures of clustering accuracy.

For example, in the hematopoietic study, BUSseq identifies 1,165 cells for Cluster 5. According to FACS labels, 1,127 of the 1,165 cells are megakaryocyte-erythrocyte progenitors (MEP). Therefore, we name Cluster 5 as MEP. We further verify the assignment of the cell type names according to the marker genes and the reference Haemopedia dataset (**Figure 5 (c) and (d)** in our manuscript).

When performing the comparisons, we calculated the ARI between the FACS cell type labels of all individual cells and the estimated cell type indicators \hat{w}_{bi} of these cells; the silhouette coefficients were calculated according to t-SNE coordinates of the output of each method with the cell types annotated by FACS labels. Thus, the calculations of ARI and silhouette coefficients are irrelevant to the names of the clusters identified by BUSseq as mentioned by the reviewer.

To clarify this point, we have renamed the section “Assignment of FACS cell type labels” as “Naming clusters learned by BUSseq according to FACS labels.” on page 36 of the main manuscript.

Minor Concerns

** This paper seems to be solving two separate problems that are not very intrinsically linked. I wonder if the theoretical discussion about chain-type and reference panel designs, along with the detailed mathematical proofs, could be better suited to a statistical journal. I admit I do not have a deep expertise in these types of proofs so I’m not able to do a thorough job verifying them. Intuitively, I think the theoretical arguments make sense, I just don’t see how they are really necessary to present the methods advance that is the most useful part for applications. My concern is that readers will think this paper is about experimental design when it is really a new method for simultaneous clustering and batch effects correction. My suggestion is the authors consider which of the two parts is the main focus of the paper and either split off the other part into a separate publication or reduce its prominence in the title/abstract so readers will not be confused.*

We sincerely thank the reviewer for the comment.

Over the last two years, researchers have actively sought to develop batch effects correction

methods for scRNA-seq data. However, existing methods do not provide conditions under which scRNA-seq experiments are valid such that batch effects can be corrected. As a result, the community has long been advocating for completely randomized experimental designs [2, 3, 7, 19]. Unfortunately, everyone is talking, but no one is listening. The problem is that a completely randomized experimental design is often too difficult to implement in reality. For example, researchers often have to enroll patients sequentially and conduct scRNA-seq experiments patient by patient. Therefore, many researchers ignore the need for experimental designs. However, such practice can be dangerous. For example, if two batches of data share only one cell type, there is no way to separate the biological variability due to cell type effects from that due to batch effects. To the best of our knowledge, “when can we analyze multiple batches of scRNA-seq data together” remains an open yet urgent question.

In this manuscript, we show mathematically that when a completely randomized experimental design is infeasible, we can still correct batch effects for scRNA-seq data under mild conditions, at least with our proposed BUSseq. Using an analogy, the previous consensus is that “ X is impossible unless A ”, with X being “batch effects correction for scRNA-seq data” and A being “a completely randomized experimental design.” Our theoretical results can be summarized as “ X is possible under B if we use method M , where $A \subset B$ ”, which implies that “ X is possible under B .” Our theoretical results demonstrate that a completely randomized experimental design is beneficial but not essential for a scRNA-seq experiment. Specifically, we prove that a reference panel experimental design and a chain-type experimental design are also legitimate for scRNA-seq experiments.

We understand the reviewer’s concern that the mathematical proofs may be overwhelming for biomedical readers. This is why we have avoided presenting any theorems in the main manuscript, and have tried to present the theoretical results as intuitively as we can in Figure 1. According to our communications with biomedical collaborators, they can understand and are very interested in learning the conditions under which we can combine multiple scRNA-seq datasets. Therefore, we hope that this manuscript not only presents our new method, BUSseq, but also elucidates when we can jointly analyze multiple scRNA-seq datasets with rigorous mathematical proofs. As there is no way to save a confounded experimental design, we feel that it is crucial to share our findings with the broad biomedical community in addition to statisticians. The question of when multiple scRNA-seq data can be analyzed together has been overlooked in the past. We hope that our answer to this question will increase researchers’ awareness of this issue and encourage them to implement valid designs for scRNA-seq experiments at the very first stage of scientific research. We solicit the reviewer’s understanding for allowing us to present both the methodological and the theoretical results and letting the readers to focus on whatever interests them.

** They should explain what are the experimental designs for the simulated and real datasets in the paper- chain-type, reference panel, etc.*

We thank the reviewer for this suggestion. All the datasets in the paper—the simulation study, the hematopoietic study, the pancreas study and the LUAD study—follow the chain-type design. We now discuss this issue on pages 8, 13, 16 and 20 of the main manuscript.

** The figures have way too many sub-panels making it very difficult to understand the main message. Please simplify and move the non-essential parts to the supplemental figures. Perhaps only show a comparison to one or two competing methods in the main figure instead of all methods.*

We thank the reviewer for the kind suggestion. After agonizing over which methods to move to the supplementary materials, in order to be fair, we hope we can still present all of the comparisons in the main manuscript with equal attention. We have provided our code for generating all of the figures on Github. If readers are interested in seeing the comparison with only one or two methods, they are more than welcome to rerun the code for only those comparisons. We solicit the reviewer for kind understandings.

** The γ_{b1} parameter is the log-odds ratio not the odds ratio.*

We are very grateful to the reviewer for pointing out this typo. The parameter γ_{b1} should be the log-odds ratio in the logistic regression for the dropout events. We have corrected the sentence on page 4 of the main manuscript.

** Fig 1a orange and gray background makes the math hard to read.*

We thank the reviewer for this comment. We lighten the background color for easier reading. We now use light red to represent the observed variables and very light grey to denote the unobserved variables.

Figure R36: The hierarchical structure of the BUSseq model. Only Y_{big} in the grey rectangle is observed.

** p. 6, second-to-last paragraph: the sentence "the latter further relaxes the assumption implied by the former", I have no idea what "the latter" and "the former" is referring to. Please make this more clear. Also, it seems like the authors are using the terms "identical" and "completely randomized" interchangeably here, whereas these terms do not mean the same thing.*

We sincerely thank the reviewer for pointing out this confusion. Following the reviewer's suggestion, we have revised the paragraph on page 6 of the main manuscript from

"The commonly advocated completely randomized experimental design falls into the "complete setting" design, whereas the latter further relaxes the assumption implied by the former that the cell-type proportions are almost the same for all batches. The identical composition of the cell population within each batch is a crucial requirement for traditional batch effects correction methods developed for bulk experiments such as ComBat [24]."

to

"The commonly advocated completely randomized experimental design is a special case of the "complete setting" design. In a completely randomized design, cells are assigned to different

batches completely at random. As a result, each batch has similar compositions of cell populations. In contrast, under the “complete setting” design, cells from different cell types can be distributed into different batches very unevenly. The requirement that batches have similar cellular compositions is crucial for traditional batch effects correction methods developed for bulk experiments, such as ComBat [24] to work well for scRNA-seq data.”

** p. 7, last paragraph: “dropout rate depends on the underlying expression levels” needs a citation. Also, the sentence containing “creating a set of functions similar to the probability generating function” is confusing and probably unnecessary. Just say something like “we prove identifiability of BUSseq in theorems XXX in the supplement”.*

We thank the reviewer for this suggestion. First, we have added a citation on page 4 of the main manuscript regarding the dependency of dropout events on the underlying expression levels:

“For scRNA-seq data, the dropout rate depends on the underlying expression levels [25].

We have also revised the aforementioned sentence on page 8 of the main manuscript following the reviewer’s suggestion: “Fortunately, however, despite the dropout events and the cell-specific size factors, we proved Theorems 1-4 (see their proofs in Supplementary Notes).”

** p. 8 simulation- how many genes were differentially expressed? were any genes just set to be pure noise across cell types with only batch effects? How many cells per true cluster?*

We thank the reviewer for this comment.

There were 500 differentially expressed genes in the simulation dataset. The remaining 2,500 genes show no biological differences between cell types, so they are pure noises with only batch effects. We detail out the distribution of cell types in all batches in **Table R21**. We have added a sentence on page 8 of the main manuscript.

The number of cells	Batch 1	Batch 2	Batch 3	Batch 4	Total
Cell Type 1	120	0	48	0	168
Cell Type 2	90	60	0	60	210
Cell Type 3	60	90	40	80	270
Cell Type 4	30	90	52	60	232
Cell Type 5	0	60	60	0	120
Total	300	300	200	200	1000

Table R21: The distribution of each cell types in the four batches. The same table is shown as **Supplementary Table 1**.

** What is the distinction between “intrinsic genes” and “marker genes”?*

We thank the reviewer for this comment. Following Huo et al. [23], we define “intrinsic genes” as genes that are differentially expressed between at least two cell types. In contrast, “marker genes” are genes that belong to certain cell types according to the literature. For example, in the pancreas study, *GCG* gene is known to be highly expressed in alpha islet cells, so this gene often serves as a marker to label alpha islet cells [27]. We now discuss this point on pages 32-33 of the main manuscript.

** How did the authors compute a BIC score from MCMC samples? Did they simply use a single draw as theta hat? Or pick a draw that had a high log-likelihood?*

We thank the reviewer for this comment. We calculate the BIC using the following formula

$$BIC = -2\log(L_o(\hat{\Theta}|\mathbf{y})) + [K(B + G) + 2B + (2B - 1)G + \sum_{b=1}^B (n_b - 1)] \cdot \log\left(\sum_{b=1}^B n_b G\right),$$

where $L_o(\hat{\Theta}|\mathbf{y}) = \prod_{b=1}^B \prod_{i=1}^{n_b} [\sum_{k=1}^K \pi_{bk} \prod_{g=1}^G Pr(Y_{big} = y_{big}|\hat{\Theta})]$ denotes the observed data likelihood function and

$$Pr(Y_{big} = y_{big}|\Theta) = \begin{cases} \sum_{x=1}^{\infty} \frac{\exp(\gamma_{b0} + \gamma_{b1}x)}{1 + \exp(\gamma_{b0} + \gamma_{b1}x)} f_{NB}(x; \exp(\alpha_g + \beta_{gk} + \nu_{bg} + \delta_{bi}), \phi_{bg}) \\ + f_{NB}(0; \exp(\alpha_g + \beta_{gk} + \nu_{bg} + \delta_{bi}), \phi_{bg}) & y_{big} = 0, \\ \frac{1}{1 + \exp(\gamma_{b0} + \gamma_{b1}y_{big})} f_{NB}(y_{big}; \exp(\alpha_g + \beta_{gk} + \nu_{bg} + \delta_{bi}), \phi_{bg}) & y_{big} > 0. \end{cases}$$

We take the posterior mean of each parameter as the estimates $\hat{\Theta} = (\hat{\alpha}, \hat{\beta}, \hat{\gamma}, \hat{\nu}, \hat{\phi}, \hat{\delta}, \hat{\pi})$. We have added an explanation on page 30 of the main manuscript.

** They should clarify whether the MCMC has to be run separately for each value of “k” used to compute the BIC for model selection. Since this is not a reversible jump or Bayesian non-parametric method I assume yes. Also, along those lines, why not use something like marginal likelihood instead of BIC?*

We thank the reviewer for the comment. We indeed ran the MCMC algorithm separately for different K s and calculated the BIC values, respectively, for model selection. Thus, we did not use any reversible jump MCMC steps or Bayesian nonparametric methods. The marginal likelihood corresponds to the observed data likelihood, which integrates over the latent variables X_{big} , Z_{big} and w_{bi} . Compared with the log marginal likelihood, the BIC incorporates a penalty regarding the number of parameters, which depends on the number of cell types K . As K increases, the marginal likelihood often increases. As a result, the model selected by the marginal likelihood often overfits the data. In contrast, the penalty in BIC helps the model selection to balance between goodness-of-fit and model complexity. We have added an explanation on page 33 of the main manuscript.

** They need to explain how they chose the perplexity parameter for tSNE, and whether PCA was run prior to tSNE as an initialization (it almost always is since tSNE cannot handle more than about 100 dimensions).*

We thank the reviewer for the comment. Following Haghverdi et al. [16], we applied the *Rtsne* function of the R package *Rtsne* with the perplexity set to 30. However, we did not run PCA prior to tSNE even if the dimension of input was large. Instead, we directly input the Euclidean distances of cells to *Rtsne* and set the option *is_distance* to TRUE. We detail out the generation of t-SNE plots in Supplementary Notes.

** Are all the results from the simulation study from a single simulated dataset? Or did they draw multiple replicates and average the results across them?*

We thank the reviewer for this comment. We have conducted extensive simulation studies and the results are similar, so in this manuscript, we only show the results for one simulation dataset for the convenience of discussion.

** Fig 3- what is the take-home message here? Please include the main conclusion in the figure legend.*

We thank the reviewer for this suggestion. We have added the following take-home message to the legend of **Figure 3** on page 11 in the revised manuscript.

“BUSseq successfully corrects the batch effects and perfectly clusters cells into different cell types in the simulation study, whose batch effects and cell type effects are of the same scale as those of the two real datasets.”

** Optionally, the authors could validate their model fit on the real data analyses by using posterior predictive checks. This would be interesting especially in comparing the zero inflated vs non-zero inflated versions of BUSseq, but would not be useful in comparing against non-Bayesian methods.*

We thank the reviewer for this great suggestion. We have conducted a posterior predictive check of the zero rates to compare the zero inflated versus non-zero inflated versions of BUSseq. For further details, please kindly see our response to your first major point.

** p. 11- “t-SNE plots confirm that BUSseq performs the best”. This is really not convincing. tSNE plots are exploratory not confirmatory, and are known to easily produce artificial clusters. The ARI and silhouette scores are sufficient to make this claim.*

We thank the reviewer for this suggestion. We have removed the sentence “t-SNE plots confirm that BUSseq performs the best.”

** Fig 4- what is the take-home message? Please include in the figure legend the main conclusion. What normalization was used prior to PCA?*

We thank the reviewer for this suggestion. We have added the following take-home message to the legend of **Figure 4** on page 14 of the main manuscript.

“BUSseq can adjust batch effects and correctly cluster cells according to their FACS cell type labels in the hematopoietic study.”

As different correction methods involve different normalization strategies and some methods (BUSseq and ZINB-WaVE) do not require prior normalization, we provide PCA plots of the raw count data without normalization in the first subgraph of the original **Figure 4(c)** of the manuscript. In case of interest, we also provide PCA plots of the uncorrected count data with the common HVGs after applying size factor normalization and gene coverage scaling between the two batches in **Figure R16**. The same figure is show as **Supplementary Figure 8**.

** Fig 5e- please use a different color scale, the white-red scale makes it hard to see the individual points. I would suggest blue-red or gray-red. Also what are the units of "expression" here?*

We thank the reviewer for this suggestion. We have changed the color of the bar from white-red to gray-red (**Figure R37**). The expression levels of each marker gene are calculated by taking the logarithm of the corrected count data plus one $\log(1 + \tilde{x}_{big})$. Thus, the unit of “expression” levels is log-scale expression level.

Figure R37: The log-scale expression levels of four marker genes, *Apoe*, *Gata2*, *Ctse* and *Igll1*, shown in the PCA plots of corrected count data by BUSseq, respectively. The digit labels denote the corresponding clusters identified by BUSseq. The same figure is updated in **Figure 6** of the main manuscript.

** p. 13- arguing that BUSseq preserves the differentiation trajectory is not very convincing given that it is essentially a clustering method. A better approach would be to use BUSseq clusters as input to a method like slingshot and compare that way. I would actually recommend just removing the whole discussion of differentiation trajectories as it seems tangential to the main arguments.*

We thank the reviewer for the kind advice.

Following the vignette of the *slingshot* R package [17], we draw the 3-dimensional slingshot based on the corrected count data output by BUSseq in the hematopoietic study (**Figure R38**). More specifically, the first three principal components of the corrected count data by BUSseq are taken as the coordinates of each cell in the 3-dimensional figure. Cells are colored by their corresponding cell types inferred by BUSseq. After naming cell clusters according to FACS labels, we can find the differentiation trajectory from long-term hematopoietic stem cells (LTHSC) or multi-potent progenitors (MPP) (green cluster in the figure) to megakaryocyte-erythrocyte progenitors (MEP, red cluster), granulocyte-monocyte progenitors (GMP, yellow cluster) and lymphoid-primed multipotent progenitors (LMPP, cyan cluster), respectively.

We agree with the reviewer that the main purpose of BUSseq is performing batch effects correction and clustering. Nevertheless, as the previous literature [16] discuss the differentiation trajectories, we wish that we can still keep the discussion just in case of readers' interests. We solicit the reviewer's kind consideration. We have mentioned **Figure R38** on page 16 of the main manuscript and elaborated the details in Supplementary Notes.

Figure R38: The slingshot of hematopoietic studies. Each point represents a cell colored by its estimated cell type label by BUSseq, and the 3-dimensional coordinates of cells correspond to the first three principal components of the corrected count data output by BUSseq. The differentiation trajectory from LTHSC or MPP (green cluster in the figure) to MEP (red cluster), GMP (yellow cluster) and LMPP (cyan cluster) can be found.

** I would like to see them compare against the cellassign method (<https://www.nature.com/articles/s41592->*

019-0529-1), which is semi-supervised and as such would be expected to outperform BUSseq since it can use more information (in the form of marker gene reference panels). Even though it's not a fair comparison, it would be very useful for the community to see the pros and cons of conducting clustering de novo (as in BUSseq) versus using reference panels to map cells to known types.

We sincerely thank the reviewer for referring us to CellAssign [48].

CellAssign [48] models the raw read count Y_{ig} of gene g in the cell i by a negative binomial distribution given the cell type indicator w_i as (Here, we reparametrize their model such that their notations are as consistent with BUSseq as possible):

$$y_{ig}|w_i = k \sim NB(\mu_{igk}, \phi_{ig}) \quad (\text{R.8})$$

$$\mu_{igk} = \exp(\alpha_g + \beta_{gk} + \delta_i + \sum_{p=1}^P \nu_{pg} x_{ip}), \quad (\text{R.9})$$

$$\phi_{igk} = \sum_{j=1}^J a_j \exp(-b_j (\mu_{ig} - x_i)^2) \quad (\text{R.10})$$

where $\alpha_g, \beta_{gk}, \delta_i$ have the same notations with BUSseq. If P is equal to the number of batches B and \mathbf{x}_i denotes the batch indicator such that if the cell i belongs to batch b , then $x_{ib} = 1$ and $x_{ic} = 0$ for all $c \neq b$, then the model for μ_{igk} turns to $\exp(\alpha_g + \beta_{gk} + \nu_{bg} + \delta_i)$. Moreover, they model ϕ_{igk} by a sum of radial basis functions and let it depend on μ_{igk} [9].

However, compared with BUSseq, CellAssign requires knowledge of the number of cell types and a set of marker genes for each cell type. CellAssign provided a vignette on how to construct marker genes for the three LUAD cell lines—HCC827, H2228, H1975—using bulk RNA-seq data generated by Holik et al. [21] with the *FindMarkers* function of the *scrn* R package. A total of 96 marker genes were identified. We matched the gene set of each protocol to the 96 maker genes and finally obtained 85 common marker genes across the three batches.

Next, following the workflow in the vignette, we extracted data for all common genes shared among these three batches and calculated the size factors over all common genes instead of only the common marker genes using the *computeSumFactors* function of *scrn* R package. Because there are three batches, we constructed a covariate matrix to indicate the corresponding batch of each cell. Then, we applied CellAssign with the batch information and set the other parameters the same as those in the example of the vignette. As CellAssign requires the input of the number of cell types, for a fair comparison, instead of selecting the number of cell types K by BIC, we applied BUSseq to the data by setting the number of cell types as three, corresponding to the three cell lines.

To evaluate the performance of clustering, CellAssign [48] calculates the accuracy and F1-score of the estimated cell types compared with the true cell type labels. We further considered the ARI between the estimated labels and true labels. We can find that BUSseq outperforms CellAssign in terms of all criteria for the LUAD dataset although BUSseq is unsupervised(**Table R22**).

We discuss about this results on page 20 of the main manuscript and explain the details into Supplementary Notes.

	F1 score	Accuracy	ARI
BUSseq	0.9971	0.9979	0.9933
CellAssign	0.9892	0.9907	0.9721

Table R22: The comparison of F1 score, Accuracy and ARI between BUSseq and CellAssign in the LUAD dataset.

** Methods p. 23- “use the mode of posterior samples of W_{bi} to infer the cell type”. Is this affected by label switching? What if there are two possible cluster assignments that have equal probability mass (ie more than one mode)? Did this ever happen in their experiments?*

We thank the reviewer for this comment. Label switching is often observed for low dimensional mixture models. However, for high-dimensional models, our experience is that samples from MCMC algorithms are likely to wander around one of the symmetric modes of the posterior distribution for a very long time as it becomes difficult to jump from one symmetric mode to another. Ideally, we could have waited for the samples to explore all of the symmetric modes and then applied label switching algorithms for adjustment and inference. In practice, we found that performing posterior inference based on samples around one of the symmetric modes of the posterior distribution gave good enough results. In neither the simulation studies nor the real data analyses did we observe label switching among the samples that we collected. Moreover, we checked the posterior probability of $Pr(W_{bi} = k | \widehat{\mathbf{X}}, \widehat{\Theta})$ for each cell based on the posterior mean estimates of the parameters $\widehat{\Theta} = (\widehat{\alpha}, \widehat{\beta}, \widehat{\gamma}, \widehat{\nu}, \widehat{\phi}, \widehat{\delta}, \widehat{\pi})$.

$$Pr(W_{bi} = k | \widehat{\mathbf{X}}, \widehat{\Theta}) \propto \widehat{\pi}_{bk} \prod_{g=1}^G \frac{\exp(\widehat{\beta}_{gk} \widehat{x}_{big})}{(\exp(\widehat{\alpha}_g + \widehat{\beta}_{gk} + \widehat{\nu}_{bg} + \widehat{\delta}_{bi}) + \widehat{\phi}_{bg})^{\widehat{x}_{big} + \widehat{\phi}_{bg}}},$$

where $\widehat{\mathbf{X}}$ denotes the imputed read counts. Recall that we assign cell i from batch b to cell type w_{bi} if $w_{bi} = \arg \max_k Pr(W_{bi} = k | \widehat{\mathbf{X}}, \widehat{\Theta})$. In **Figure R39**, we plot the distribution of $\max_k Pr(W_{bi} = k | \widehat{\mathbf{X}}, \widehat{\Theta})$ for all of the cells for the two real applications. We can see that the highest conditional probabilities are always above 0.5 and very close to one for most cells in these two studies. Thus, we do not observe the scenario under which a cell has equal probability mass

to be assigned to two clusters in our analyses. We now discuss this issue in Supplementary Notes.

Figure R39: The histogram of the highest conditional probabilities $\max_k Pr(W_{bi} = k | \widehat{\mathbf{X}}, \widehat{\Theta})$ for all cells in (a) the hematopoietic study and (b) the pancreas study. The same figure is shown as **Supplementary Figure 19**.

** Methods “batch effects corrected values”- I’m confused why the authors want to draw random samples to produce a “batch corrected” dataset. This will still have zeros and other features that make it hard to analyze with something like PCA (eg we are going to have to log transform and/or try to do variance stabilization). Why not just use (alpha+beta) as the imputed values because these are real-valued and likely have a nicer distribution than the negative binomial?*

We thank the reviewer for the comment and apologize for any potential confusion. The use of batch-effect-corrected data by ComBat [24] is very popular. For ComBat, the model is formulated as:

$$Y_{big} = \alpha_g + \mathbf{X}\beta_g + \nu_{bg} + \delta_{bi}\varepsilon_{big},$$

where α_g is the overall gene expression, \mathbf{X} is a design matrix for sample conditions, and β_g is the vector of regression coefficients corresponding to \mathbf{X} . The error terms, ε_{big} , can be assumed to follow a normal distribution with expected value zero and variance σ_g^2 . The ν_{bg} and δ_{bg} represent the additive and multiplicative batch effects of batch i for gene g , respectively.

If $y_{big}^* = \widehat{\alpha}_g + \mathbf{X}\widehat{\beta}_g$ is taken as the corrected value, then all samples under the same biological condition will have exactly the same numbers. Instead of $\widehat{\alpha}_g + \mathbf{X}\widehat{\beta}_g$, ComBat outputs

$$y_{big}^* = \frac{y_{big} - \widehat{\alpha}_g - \mathbf{X}\widehat{\beta}_g - \widehat{\nu}_{bbg}}{\widehat{\delta}_{bi}} + \widehat{\alpha}_g + \mathbf{X}\widehat{\beta}_g,$$

which removes the batch effects and preserves all other variabilities.

We wish to provide a version of the batch-effect-corrected data $\widetilde{\mathbf{X}}$ similar to that of ComBat to facilitate downstream analysis. Ideally, if x_{big} is the α^{th} percentile of $NB(\exp(\widehat{\alpha}_g + \widehat{\beta}_g\widehat{w}_{bi}) + \widehat{\nu}_{bg} + \widehat{\delta}_{bi}), \widehat{\phi}_{bg})$, we aim to take the α^{th} percentile of $NB(\exp(\widehat{\alpha}_g + \widehat{\beta}_g\widehat{w}_{bi}), \widehat{\phi}_{bg})$ as the corrected value

\tilde{x}_{big} . However, the negative binomial distribution is a discrete distribution. As a result, several \tilde{x} s can lie between the $Pr(x \leq x_{big} - 1)$ -percentile and $Pr(x \leq x_{big})$ -percentile of the distribution of \tilde{X}_{big} . For example, if $X_{big} \sim NB(exp(2), 3)$, $\tilde{X}_{big} \sim NB(exp(3), 5)$, and our observed value $x_{big} = 8$, then $Pr(x_{big} \leq 7)$ and $Pr(x_{big} \leq 8)$ correspond to the 58.67th and 65.76th percentiles of $NB(exp(2), 3)$. However, three numbers—21, 22 and 23—lie between 58.67th and 65.76th percentile of $NB(exp(3), 5)$. Thus, to avoid bias, we draw one number uniformly from 21, 22 and 23 rather than taking the maximum or the minimum to calculate \tilde{x}_{big} . We now elaborate the point on pages 34-35 of the main manuscript.

Referee Information

F. William Townes

Department of Computer Science

Princeton University

References

- [1] Tallulah S Andrews and Martin Hemberg. False signals induced by single-cell imputation. *F1000Research*, 7, 2018.
- [2] Rhonda Bacher and Christina Kendzioriski. Design and computational analysis of single-cell RNA-sequencing experiments. *Genome Biology*, 17(1):63, 2016.
- [3] J Baran-Gale, T Chandra, and K Kirschner. Experimental design for single-cell RNA sequencing. *Briefings in Functional Genomics*, 17(4), 2017.
- [4] Philip Brennecke, Simon Anders, Jong Kyoung Kim, Aleksandra A Kołodziejczyk, Xiuwei Zhang, Valentina Proserpio, Bianka Baying, Vladimir Benes, Sarah A Teichmann, John C Marioni, et al. Accounting for technical noise in single-cell RNA-seq experiments. *Nature Methods*, 10(11):1093, 2013.
- [5] Suzanne J Clark and Joe N Perry. Estimation of the negative binomial parameter κ by maximum quasi-likelihood. *Biometrics*, pages 309–316, 1989.
- [6] Hongsheng Dai, Yanchun Bao, and Mingtang Bao. Maximum likelihood estimate for the dispersion parameter of the negative binomial distribution. *Statistics & Probability Letters*, 83(1):21–27, 2013.
- [7] Molin A Dal and Camillo B Di. How to design a single-cell RNA-sequencing experiment: pitfalls, challenges and perspectives. *Briefings in Bioinformatics*, (1), 2018.
- [8] Angelo Duò, Mark D Robinson, and Charlotte Sonesson. A systematic performance evaluation of clustering methods for single-cell RNA-seq data. *F1000Research*, 7, 2018.
- [9] Nils Eling, Arianne C Richard, Sylvia Richardson, John C Marioni, and Catalina A Vallejos. Correcting the mean-variance dependency for differential variability testing using single-cell RNA sequencing data. *Cell Systems*, 7(3):284–294, 2018.

- [10] Gökçen Eraslan, Lukas M Simon, Maria Mircea, Nikola S Mueller, and Fabian J Theis. Single-cell RNA-seq denoising using a deep count autoencoder. *Nature Communications*, 10(1):1–14, 2019.
- [11] Andrew Gelman, Xiao-Li Meng, and Hal Stern. Posterior predictive assessment of model fitness via realized discrepancies. *Statistica Sinica*, pages 733–760, 1996.
- [12] Andrew Gelman, Gareth O Roberts, Walter R Gilks, et al. Efficient metropolis jumping rules. *Bayesian Statistics*, 5(599-608):42, 1996.
- [13] Andrew Gelman, John B Carlin, Hal S Stern, David B Dunson, Aki Vehtari, and Donald B Rubin. *Bayesian data analysis*. Chapman and Hall/CRC, 2013.
- [14] Wuming Gong, Il-Youp Kwak, Pruthvi Pota, Naoko Koyano-Nakagawa, and Daniel J Garry. Drimpute: imputing dropout events in single cell RNA sequencing data. *BMC Bioinformatics*, 19(1):220, 2018.
- [15] Dominic Grün, Mauro J Muraro, Jean-Charles Boisset, Kay Wiebrands, Anna Lyubimova, Gitanjali Dharmadhikari, Maaïke van den Born, Johan Van Es, Erik Jansen, Hans Clevers, et al. De novo prediction of stem cell identity using single-cell transcriptome data. *Cell Stem Cell*, 19(2):266–277, 2016.
- [16] Laleh Haghverdi, Aaron TL Lun, Michael D Morgan, and John C Marioni. Batch effects in single-cell RNA-sequencing data are corrected by matching mutual nearest neighbors. *Nature Biotechnology*, 36(5):421, 2018.
- [17] Trevor Hastie and Werner Stuetzle. Principal curves. *Journal of the American Statistical Association*, 84(406):502–516, 1989.
- [18] W Keith Hastings. Monte Carlo sampling methods using Markov chains and their applications. *Biometrika*, 57(1):97–109, 1970.
- [19] S. C. Hicks, F. W. Townes, M. Teng, and R. A. Irizarry. Missing data and technical variability in single-cell RNA-sequencing experiments. *Biostatistics*, 19(4):562–578, 2018.
- [20] Brian Hie, Bryan Bryson, and Bonnie Berger. Efficient integration of heterogeneous single-cell transcriptomes using Scanorama. *Nature Biotechnology*, 37(6):685, 2019.
- [21] Aliaksei Z Holik, Charity W Law, Ruijie Liu, Zeya Wang, Wenyi Wang, Jaeil Ahn, Marie-Liesse Asselin-Labat, Gordon K Smyth, and Matthew E Ritchie. RNA-seq mixology: designing realistic control experiments to compare protocols and analysis methods. *Nucleic Acids Research*, 45(5):e30–e30, 2017.
- [22] Mo Huang, Jingshu Wang, Eduardo Torre, Hannah Dueck, Sydney Shaffer, Roberto Bonasio, John I Murray, Arjun Raj, Mingyao Li, and Nancy R Zhang. Saver: gene expression recovery for single-cell rna sequencing. *Nature Methods*, 15(7):539, 2018.
- [23] Zhiguang Huo, Ying Ding, Silvia Liu, Steffi Oesterreich, and George Tseng. Meta-analytic framework for sparse k-means to identify disease subtypes in multiple transcriptomic studies. *Journal of the American Statistical Association*, 111(513):27–42, 2016.
- [24] W Evan Johnson, Cheng Li, and Ariel Rabinovic. Adjusting batch effects in microarray expression data using empirical Bayes methods. *Biostatistics*, 8(1):118–127, 2007.

- [25] Peter V Kharchenko, Lev Silberstein, and David T Scadden. Bayesian approach to single-cell differential expression analysis. *Nature Methods*, 11(7):740, 2014.
- [26] Jerald F Lawless. Negative binomial and mixed poisson regression. *Canadian Journal of Statistics*, 15(3):209–225, 1987.
- [27] Nathan Lawlor, Joshy George, Mohan Bolisetty, Romy Kursawe, Lili Sun, V Sivakamasundari, Ina Kycia, Paul Robson, and Michael L Stitzel. Single-cell transcriptomes identify human islet cell signatures and reveal cell-type-specific expression changes in type 2 diabetes. *Genome Research*, 27(2):208–222, 2017.
- [28] Wei Vivian Li and Jingyi Jessica Li. An accurate and robust imputation method scImpute for single-cell RNA-seq data. *Nature Communications*, 9(1):1–9, 2018.
- [29] Yingxin Lin, Shila Ghazanfar, Dario Strbenac, Andy Wang, Ellis Patrick, David M Lin, Terence Speed, Jean YH Yang, and Pengyi Yang. Evaluating stably expressed genes in single cells. *GigaScience*, 8(9):giz106, 2019.
- [30] Romain Lopez, Jeffrey Regier, Michael B Cole, Michael I Jordan, and Nir Yosef. Deep generative modeling for single-cell transcriptomics. *Nature Methods*, 15(12):1053, 2018.
- [31] Michael I Love, Wolfgang Huber, and Simon Anders. Moderated estimation of fold change and dispersion for RNA-seq data with *deseq2*. *Genome Biology*, 15(12):550, 2014.
- [32] Aaron TL Lun, Karsten Bach, and John C Marioni. Pooling across cells to normalize single-cell RNA sequencing data with many zero counts. *Genome Biology*, 17(1):75, 2016.
- [33] Aaron TL Lun, Davis J McCarthy, and John C Marioni. A step-by-step workflow for low-level analysis of single-cell RNA-seq data with Bioconductor. *F1000Research*, 5, 2016.
- [34] Sonia Nestorowa, Fiona K Hamey, Blanca Pijuan Sala, Evangelia Diamanti, Mairi Shepherd, Elisa Laurenti, Nicola K Wilson, David G Kent, and Berthold Göttgens. A single cell resolution map of mouse haematopoietic stem and progenitor cell differentiation. *Blood*, 128(8):e20–31, 2016.
- [35] Franziska Paul, Yaara Arkin, Amir Giladi, Diego Adhemar Jaitin, Ephraim Kenigsberg, Hadas Keren-Shaul, Deborah Winter, David Lara-Astiaso, Meital Gury, Assaf Weiner, et al. Transcriptional heterogeneity and lineage commitment in myeloid progenitors. *Cell*, 163(7):1663–1677, 2015.
- [36] Davide Risso, Fanny Perraudeau, Svetlana Gribkova, Sandrine Dudoit, and Jean-Philippe Vert. A general and flexible method for signal extraction from single-cell RNA-seq data. *Nature Communications*, 9(1):284, 2018.
- [37] Gareth O Roberts and Jeffrey S Rosenthal. Examples of adaptive mcmc. *Journal of Computational and Graphical Statistics*, 18(2):349–367, 2009.
- [38] Mark D Robinson and Alicia Oshlack. A scaling normalization method for differential expression analysis of RNA-seq data. *Genome Biology*, 11(3):R25, 2010.
- [39] Krishna Saha and Sudhir Paul. Bias-corrected maximum likelihood estimator of the negative binomial dispersion parameter. *Biometrics*, 61(1):179–185, 2005.
- [40] Tim Stuart, Andrew Butler, Paul Hoffman, Christoph Hafemeister, Efthymia Papalexi,

- William M Mauck III, Yuhan Hao, Marlon Stoeckius, Peter Smibert, and Rahul Satija. Comprehensive integration of single-cell data. *Cell*, 177(7):1888–1902.e21, 2019.
- [41] Luyi Tian, Xueyi Dong, Saskia Freytag, Kim-Anh Lê Cao, Shian Su, Abolfazl JalalAbadi, Daniela Amann-Zalcenstein, Tom S Weber, Azadeh Seidi, Jafar S Jabbari, et al. Benchmarking single cell RNA-sequencing analysis pipelines using mixture control experiments. *Nature Methods*, page 1, 2019.
- [42] Luke Tierney. Markov chains for exploring posterior distributions. *Annals of Statistics*, pages 1701–1728, 1994.
- [43] David Van Dijk, Roshan Sharma, Juozas Nainys, Kristina Yim, Pooja Kathail, Ambrose J Carr, Cassandra Burdziak, Kevin R Moon, Christine L Chaffer, Diwakar Pattabiraman, et al. Recovering gene interactions from single-cell data using data diffusion. *Cell*, 174(3): 716–729, 2018.
- [44] Florian Wagner, Yun Yan, and Itai Yanai. K-nearest neighbor smoothing for high-throughput single-cell RNA-seq data. *BioRxiv*, page 217737, 2017.
- [45] Joshua D Welch, Velina Kozareva, Ashley Ferreira, Charles Vanderburg, Carly Martin, and Evan Z Macosko. Single-cell multi-omic integration compares and contrasts features of brain cell identity. *Cell*, 177(7):1873–1887.e17, 2019.
- [46] LJ Willson, JL Folks, and JH Young. Complete sufficiency and maximum likelihood estimation for the two-parameter negative binomial distribution. *Metrika*, 33(1):349–362, 1986.
- [47] Luke Zappia, Belinda Phipson, and Alicia Oshlack. Splatter: simulation of single-cell RNA sequencing data. *Genome Biology*, 18(1):174, 2017.
- [48] Allen W Zhang, Ciara O’Flanagan, Elizabeth A Chavez, Jamie LP Lim, Nicholas Ceglia, Andrew McPherson, Matt Wiens, Pascale Walters, Tim Chan, Brittany Hewitson, et al. Probabilistic cell-type assignment of single-cell RNA-seq for tumor microenvironment profiling. *Nature Methods*, 16(10):1007–1015, 2019.
- [49] Parekh S Reinius B Guillaumet-Adkins A Smets M Leonhardt H Heyn H Hellmann I Enard W. Ziegenhain C, Vieth B. Comparative analysis of single-cell RNA sequencing methods. *Molecular Cell*, 4(631-643).

REVIEWERS' COMMENTS:

Reviewer #1 (Remarks to the Author):

The authors have addressed all of my comments in a very detailed and extensive revision. I do not have further comments.

Reviewer #2 (Remarks to the Author):

Thanks to the authors for addressing all of my comments from the previous revision.

F. William Townes

Below, we display reviewers' comments in *blue italics*.

Reviewer 1 (Remarks to the Author):

The authors have addressed all of my comments in a very detailed and extensive revision. I do not have further comments.

We sincerely thank the reviewer for his or her thorough review and constructive suggestions.

Reviewer 2 (Remarks to the Author):

*Thanks to the authors for addressing all of my comments from the previous revision.
F. William Townes.*

We sincerely thank Dr. Townes for his thorough review and constructive suggestions.